# Stable Minima of ReLU Neural Networks Suffer from the Curse of Dimensionality:
# The Neural Shattering Phenomenon

**Tongtong Liang**
UC San Diego
ttliang@ucsd.edu

**Dan Qiao**
UC San Diego
d2qiao@ucsd.edu

**Yu-Xiang Wang**
UC San Diego
yuxiangw@ucsd.edu

**Rahul Parhi**
UC San Diego
rahul@ucsd.edu

## Abstract

We study the implicit bias of flatness / low (loss) curvature and its effects on generalization in two-layer overparameterized ReLU networks with multivariate inputs—a problem well motivated by the minima stability and edge-of-stability phenomena in gradient-descent training. Existing work either requires interpolation or focuses only on univariate inputs. This paper presents new and somewhat surprising theoretical results for multivariate inputs. On two natural settings (1) generalization gap for flat solutions, and (2) mean-squared error (MSE) in nonparametric function estimation by stable minima, we prove upper and lower bounds, which establish that while flatness does imply generalization, the resulting rates of convergence necessarily deteriorate exponentially as the input dimension grows. This gives an exponential separation between the flat solutions compared to low-norm solutions (i.e., weight decay), which are known not to suffer from the curse of dimensionality. In particular, our minimax lower bound construction, based on a novel packing argument with boundary-localized ReLU neurons, reveals how flat solutions can exploit a kind of "neural shattering" where neurons rarely activate, but with high weight magnitudes. This leads to poor performance in high dimensions. We corroborate these theoretical findings with extensive numerical simulations. To the best of our knowledge, our analysis provides the first systematic explanation for why flat minima may fail to generalize in high dimensions.

## 1   Introduction

Modern deep learning is inherently overparameterized. In this regime, there are typically infinitely many global (i.e., zero-loss or interpolating) minima to the training objective, yet gradient-descent (GD) training seems to successfully avoid "bad" minima, finding those that generalize well. Understanding this phenomenon boils down to understanding the *implicit biases* of training algorithms [Zhang et al., 2021]. A large body of work has focused on understanding this phenomenon in the interpolation regime [Du et al., 2018, Liu et al., 2022], and the related concept of "benign overfitting" [Belkin et al., 2019, Bartlett et al., 2020, Frei et al., 2022].

While these directions have been fruitful, there is increasing evidence that rectified linear unit (ReLU) neural networks do not benignly overfit [Mallinar et al., 2022, Haas et al., 2023], particularly in the case of learning problems with noisy data [Joshi et al., 2023, Qiao et al., 2024]. Furthermore, for noisy labels, it takes many iterations of GD to actually interpolate the labels [Zhang et al., 2021]. This discounts theories based on interpolation to explain the generalization performance of *practical* neural networks, which would have entered the so-called *edge-of-stability* regime [Cohen et al., 2020], or stopped long before interpolating the training data.

39th Conference on Neural Information Processing Systems (NeurIPS 2025).

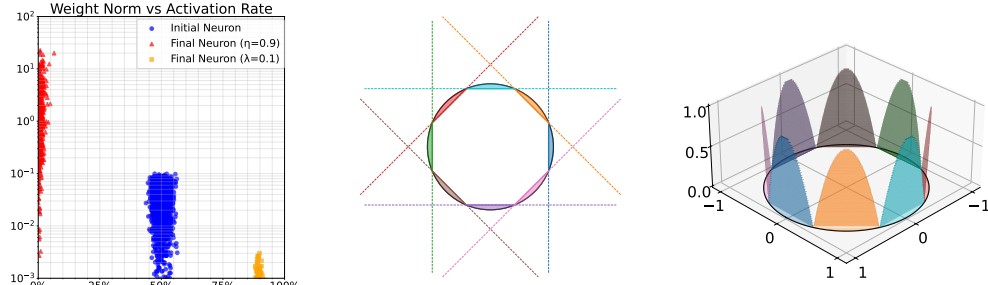

Figure 1: The "neural shattering" phenomenon: From empirical observations to its geometric origin and theoretical consequences. **Left panel:** Training with a large learning rate and gradient descent empirically results in "neural shattering": Neurons develop large weights despite activating on very few inputs, leading to a high MSE of $\approx 1.105$ (red points). In contrast, explicit $\ell^2$-regularization prevents this, achieving a much lower MSE of $\approx 0.055$ (orange points). **Middle panel:** The number of distinct directions, or "caps", on a high-dimensional sphere grows exponentially. Consequently, the data sites are spread thinly across these caps. This makes it trivial for a ReLU neuron to find a direction that isolates only a few data points. This sparse activation pattern allows neurons to use large weight magnitudes for this local fitting without impacting the global loss curvature, thus "tricking" the flatness criterion. **Right panel:** Visualization of "hard-to-learn" function from our minimax lower bound construction, built from localized ReLU neurons described in the middle panel.

To that end, it has been observed that an important factor that affects/characterizes the implicit bias of GD training is the notion of *dynamical stability* [Wu et al., 2018]. Intuitively, the (dynamical) stability of a particular minimum refers to the ability of the training algorithm to "stably converge" to that minimum. The stability of a minimum is intimately related to the flatness of the loss landscape about the minima [Mulayoff et al., 2021]. A number of recent works have focused on understanding *linear stability*, i.e., the stability of an algorithm's linearized dynamics about a minimum, in order to characterize the implicit biases of training algorithms [Wu et al., 2018, Nar and Sastry, 2018, Mulayoff et al., 2021, Ma and Ying, 2021, Nacson et al., 2023]. Minima that exhibit linear stability are often referred to as stable minima. In particular, Mulayoff et al. [2021] and Nacson et al. [2023] focus on the interpolation regime of two-layer overparameterized ReLU neural networks in the univariate input and multivariate input settings, respectively. Roughly speaking, the main takeaway from their work is that stability / flatness in parameter space implies a bounded-variation-type of smoothness in function space.

Moving beyond the interpolation regime, Qiao et al. [2024] extend the framework of Mulayoff et al. [2021] and provide generalization and risk bounds for stable minima in the non-interpolation regime for univariate inputs. They show that for univariate nonparametric regression, the functions realized by stable minima cannot overfit in the sense that the generalization gap vanishes as the number of training examples grows. Furthermore, they show that the learned functions achieve near-optimal estimation error rates for functions of second-order bounded variation on an interval strictly inside the data support. While this work is a good start, it begs the questions of (i) what happens in the multivariate / high-dimensional case and (ii) what happens off of this interval (i.e., how does the network *extrapolate*). Indeed, these are key to understanding the implicit bias of GD trained neural networks, especially since learning high dimensions seems to always amount to extrapolation [Balestriero et al., 2021]. These two questions motivate the present paper in which we provide a precise answer to the following fundamental question.

*How well do stable minima of two-layer overparameterized ReLU neural networks perform in the high-dimensional and non-interpolation regime?*

We provide several new theoretical results for stable minima in this scenario, which are corroborated by numerical simulations. Some of our findings are surprising given the current state of understanding of stable minima. Notably, we show that, while flatness does imply generalization, the resulting sample complexity grows exponentially with the input dimension. This gives an exponential separation between flat solutions and low-norm solutions (weight decay) which are known not to suffer from the curse of dimensionality [Bach, 2017, Parhi and Nowak, 2023b].

## 1.1 Contributions

In this paper, we provide new theoretical results for stable minima of two-layer ReLU neural networks, particularly in the high-dimensional and non-interpolation regime. Our primary contributions lie in the rigorous analysis of the generalization and statistical properties of stable minima and the resulting insights into their high-dimensional behavior. In particular, our contributions include the following.

1. We establish that the functions realized by stable minima are regular in the sense of a weighted variation norm (Theorem 3.2 and Corollary 3.3). This norm defines a *data-dependent* function class that captures the inductive bias of stable minima.[1] Furthermore, this regularity admits an analytic description as a form of weighted total variation in the domain of the Radon transform. These results synthesize and extend previous work [cf., Nacson et al., 2023, Qiao et al., 2024] by removing interpolation assumptions and generalizing them to multivariate inputs.

2. We analyze the generalization properties of stable minima in both a statistical learning setting and a nonparametric regression setting defined using the smoothness class above.

   - We establish that stable minima provably cannot overfit in the sense that their generalization gap (i.e., a uniform convergence bound) tends to 0 as the number of training examples $n \to \infty$ at a rate $n^{-\frac{1}{2d+4}}$ up to logarithmic factors (Theorem 3.5).
   - For high-dimensional ($d > 1$) nonparametric regression, we show that stable minima (up to logarithmic factors) achieve an estimation error rate, in mean-squared error (MSE), upper bounded by $n^{-\frac{1}{2d+4}}$ (Theorem 3.6).
   - We prove a minimax lower bound of rate $n^{-\frac{2}{d+1}}$ up to a constant (Theorem 3.7) on both the MSE and the generalization gap, which certifies that stable minima are not immune to the curse of dimensionality. This gives an exponential separation between flat solutions and low-norm solutions (weight decay) [Bach, 2017, Parhi and Nowak, 2023b].
   - By specializing the MSE upper bound to the univariate case ($d = 1$), we show that stable minima (up to logarithmic factors) achieve an upper bound of $n^{-\frac{1}{6}}$. Furthermore, by a construction specific to the univariate case, we have a sharper lower bound of $n^{-\frac{1}{2}}$ when $d = 1$. These results should be contrasted to those of Qiao et al. [2024], who derive matching upper and lower bounds of $n^{-\frac{4}{5}}$ on an interval strictly inside the data support. Note that our results hold over the full domain, therefore capturing how the networks extrapolate. Thus, our results provide a more realistic characterization of the statistical properties of stable minima in the univariate case than in prior work.

3. In Section 4, we corroborate our theoretical results with extensive numerical simulations. As a by-product, we uncover and characterize a phenomenon we refer to as "neural shattering" that is inherent to stable minima in high dimensions. This refers to the observation that each neuron in a flat solution has very few activated data points, which means that the activation boundaries of the ReLU neurons in the solutions shatter the data set into small pieces. This leads to poor performance in high dimensions. We also highlight that this observation exactly matches the construction of "hard-to-learn" functions for our minimax lower bound. Thus, our empirical validation combined with our theoretical analysis offers fresh insights into how high-dimensionality impacts neural network optimization and generalization. Indeed, our results reveal a subtle mechanism that leads to poor performance specifically in high dimensions.

These results are based on two novel technical innovations in the analysis of minima stability in comparison to prior works, which we summarize below.

**Statistical bounds on the full input domain.** The data-dependent nature of the stable minima function class implies that there are regions of the input domain where neuron activations are sparse for stable minima. This is because the functions in this class have local smoothness that can become arbitrarily irregular near the boundary of the data support. This makes it challenging to study the statistical performance of stable minima in the irregular regions. This was bypassed in the univariate case by Qiao et al. [2024] by restricting their attention to an interval strictly inside the data support, completely ignoring these hard-to-handle regions. Our analysis overcomes this via a novel technique

---

[1]More specifically, this quantity defines a *seminorm* which correspondingly defines a kind of Banach space of functions called a *weighted variation space* [DeVore et al., 2025].

that balances the error strictly inside the data support with the error close to the boundary. This allows us to establish meaningful statistical bounds on the full input domain.

**ReLU-specific minimax lower bound construction.** We develop a novel minimax lower bound construction (see proof of Theorem 3.7) using functions built from sums of ReLU neurons. These neurons are strategically chosen to have activation regions near the boundary of the input domain. This exploits the "on/off" nature of ReLUs and high-dimensional geometry to create "hard-to-learn" functions. The data-dependent weighting allows these sparsely active, high-magnitude neurons to exist within the stable minima function class. This construction is fundamentally different from classical nonparametric techniques and is tightly linked to our experimental findings on neural shattering (see Figure 1).

## 1.2 Related Work

**Stable minima and function spaces.** Many works have investigated characterizations of the implicit bias of GD training from the perspective of dynamical stability [Wu et al., 2018, Nar and Sastry, 2018, Mulayoff et al., 2021, Ma and Ying, 2021, Nacson et al., 2023, Wang et al., 2022, Qiao et al., 2024]. In particular, Mulayoff et al. [2021] characterized the function-space implicit bias of minima stability for two-layer overparameterized univariate ReLU networks in the interpolation regime. This was extended to the multivariate case by Nacson et al. [2023] and, in the univariate case, this was extended to the non-interpolation regime by Qiao et al. [2024] with the addition of generalization guarantees. In this paper, we extend these works to the high-dimensional and non-interpolation regime and characterize the generalization and statistical properties of stable minima.

**Nonparametric function estimation with neural networks.** It is well known that neural networks are minimax optimal estimators for a wide variety of functions [Suzuki, 2018, Schmidt-Hieber, 2020, Kohler and Langer, 2021, Parhi and Nowak, 2023b, Zhang and Wang, 2023, Yang and Zhou, 2024, Qiao et al., 2024]. Outside of the univariate work of Qiao et al. [2024], all prior works construct their estimators via empirical risk minimization problems. Thus, they do not incorporate the training dynamics that arise when training neural networks in practice. Thus, the results of this paper provide more practically relevant results on nonparametric function estimation, providing estimation error rates achieved by local minima that GD training can stably converge to.

**Loss curvature and generalization.** A long-standing theory to explain why overparameterized neural networks generalize well is that the flat minima found by GD training generalize well [Hochreiter and Schmidhuber, 1997, Keskar et al., 2017]. Although there is increasing theoretical evidence for this phenomenon in various settings [Ding et al., 2024, Qiao et al., 2024], there is also evidence that sharp minima can also generalize [Dinh et al., 2017]. Thus, this paper adds complementary results to this list where we establish that, while flatness does imply generalization for two-layer ReLU networks, the resulting sample complexity grows exponentially with the input dimension.

## 2 Preliminaries, Notation, and Problem Setup

We investigate learning with two-layer ReLU neural networks. Our focus is on understanding the generalization and statistical performance of solutions obtained through GD training, particularly those that are stable.

**Neural networks.** We consider two-layer ReLU neural networks with $K$ neurons. Such a network implements a function $f_{\boldsymbol{\theta}} : \mathbb{R}^d \to \mathbb{R}$ of the form

$$f_{\boldsymbol{\theta}}(\boldsymbol{x}) = \sum_{k=1}^{K} v_k \phi(\boldsymbol{w}_k^{\mathsf{T}} \boldsymbol{x} - b_k) + \beta, \tag{1}$$

where $\boldsymbol{\theta} = \{K\} \cup \{v_k, \boldsymbol{w}_k, b_k\}_{k=1}^{K} \cup \{\beta\}$ denotes the collection of all neural network parameters, including the width $K \in \mathbb{N}$. Here, $v_k \in \mathbb{R}$ denotes the output-layer weights, $\boldsymbol{w}_k \in \mathbb{R}^d$ denotes the input-layer weights, $b_k \in \mathbb{R}$ denotes the input-layer biases, and $\beta \in \mathbb{R}$ denotes the output-layer bias.

**Data fitting and loss function.** We consider the problem of fitting the data $\mathcal{D} = \{(\boldsymbol{x}_i, y_i)\}_{i=1}^{n}$, where $\boldsymbol{x}_i \in \mathbb{R}^d$ and $y_i \in \mathbb{R}$. We consider the empirical risk minimization problem with squared-error loss $\mathcal{L}(\boldsymbol{\theta}) = \frac{1}{2n} \sum_{i=1}^{n} (y_i - f_{\boldsymbol{\theta}}(\boldsymbol{x}_i))^2$.

**Gradient descent and minima stability.** We aim to minimize $\mathcal{L}(\cdot)$ via GD training, i.e., we consider the iteration $\boldsymbol{\theta}_{t+1} = \boldsymbol{\theta}_t - \eta \nabla_{\boldsymbol{\theta}} \mathcal{L}(\boldsymbol{\theta}_t)$, for $t = 0, 1, 2, \ldots$, where $\eta > 0$ is the step size / learning rate, $\nabla_{\boldsymbol{\theta}}$ denotes the gradient operator with respect to $\boldsymbol{\theta}$, $\nabla_{\boldsymbol{\theta}}^2$ denotes the Hessian operator with respect to $\boldsymbol{\theta}$, and the iteration is initialized with some initial condition $\boldsymbol{\theta}_0$. The analysis of these dynamics in generality is intractable in most cases. Thus, following the work of Wu et al. [2018], many works [e.g., Nar and Sastry, 2018, Mulayoff et al., 2021, Ma and Ying, 2021, Wang et al., 2022, Nacson et al., 2023, Qiao et al., 2024] have considered the behavior of this iteration using *linearized dynamics* about a minimum. Following Mulayoff et al. [2021], we consider the Taylor series expansion of the loss function about a minimum $\boldsymbol{\theta}^\star$.[2] That is,

$$\mathcal{L}(\boldsymbol{\theta}) \approx \mathcal{L}(\boldsymbol{\theta}^\star) + (\boldsymbol{\theta} - \boldsymbol{\theta}^\star)^\mathsf{T} \nabla_{\boldsymbol{\theta}} \mathcal{L}(\boldsymbol{\theta}^\star) + \frac{1}{2}(\boldsymbol{\theta} - \boldsymbol{\theta}^\star)^\mathsf{T} \nabla_{\boldsymbol{\theta}}^2 \mathcal{L}(\boldsymbol{\theta}^\star)(\boldsymbol{\theta} - \boldsymbol{\theta}^\star). \tag{2}$$

As the GD iteration approaches a minimum $\boldsymbol{\theta}^\star$, it is well approximated by the *linearized dynamics*

$$\boldsymbol{\theta}_{t+1} = \boldsymbol{\theta}_t - \eta \Big[ \nabla_{\boldsymbol{\theta}} \mathcal{L}(\boldsymbol{\theta}^\star) + \nabla_{\boldsymbol{\theta}}^2 \mathcal{L}(\boldsymbol{\theta}^\star)(\boldsymbol{\theta}_t - \boldsymbol{\theta}^\star) \Big], \quad t = 0, 1, 2, \ldots. \tag{3}$$

A minimum is said to be *linearly stable* if the GD iterates are "trapped" once they enter a neighborhood of the minimum. See Wu et al. [2018], Ma and Ying [2021], or Chemnitz and Engel [2025] for various rigorous definitions of linear stability that have appeared in the literature. It turns out that stability is tightly connected to the flatness of the minimum. Indeed, many equivalences have been proven, e.g., Mulayoff et al. [2021, Lemma 1], Qiao et al. [2024, Lemma 2.2], or Chemnitz and Engel [2025, Section 2.3]. We have the following proposition from Chemnitz and Engel [2025, p. 7].

**Proposition 2.1.** *Suppose that $\eta < 2$. A minimum $\boldsymbol{\theta}^\star$ is* linearly stable[3] *if and only if*

$$\lambda_{\max}(\nabla_{\boldsymbol{\theta}}^2 \mathcal{L}(\boldsymbol{\theta}^\star)) \leq \frac{2}{\eta}. \tag{4}$$

Thus, we see that the stability of a minimum is equivalent to the flatness of the minimum under the assumption that the step size $\eta$ satisfies $\eta < 2$. Thus, we make this assumption in the remainder of this paper. Given a data set $\mathcal{D}$, we refer to the class of neural network parameters

$$\Theta_{\mathrm{flat}}(\eta; \mathcal{D}) := \left\{ \boldsymbol{\theta} : \ \lambda_{\max}(\nabla_{\boldsymbol{\theta}}^2 \mathcal{L}(\boldsymbol{\theta})) \leq \frac{2}{\eta} \right\}, \tag{5}$$

as the collection of flat / stable minima or flat / stable solutions. This parameter class is further motivated by empirical observations that GD often operates in the *edge-of-stability regime*, where $\lambda_{\max}(\nabla_{\boldsymbol{\theta}}^2 \mathcal{L}(\boldsymbol{\theta}_t))$ hovers around $2/\eta$ [Cohen et al., 2020, Damian et al., 2024].

## 3 Main Results

In this section, we characterize the implicit bias of stable solutions. It turns out that every function $f_{\boldsymbol{\theta}}$, with $\boldsymbol{\theta} \in \Theta_{\mathrm{flat}}(\eta; \mathcal{D})$, is regular in the sense of a weighted variation norm. In particular, the weight function is a data-dependent quantity. This weight function reveals that neural networks can learn features that are intrinsic within the structure of the training data. To that end, given a data set $\mathcal{D} = \{(\boldsymbol{x}_i, y_i)\}_{i=1}^n \subset \mathbb{R}^d \times \mathbb{R}$, we consider a weight function $g : \mathbb{S}^{d-1} \times \mathbb{R} \to \mathbb{R}$, where $\mathbb{S}^{d-1} := \{\boldsymbol{u} \in \mathbb{R}^d : \|\boldsymbol{u}\| = 1\}$ denotes the unit sphere. This weight is defined by $g(\boldsymbol{u}, t) := \min\{\tilde{g}(\boldsymbol{u}, t), \tilde{g}(-\boldsymbol{u}, -t)\}$, where

$$\tilde{g}(\boldsymbol{u}, t) := \mathbb{P}(\boldsymbol{X}^\mathsf{T} \boldsymbol{u} > t)^2 \cdot \mathbb{E}[\boldsymbol{X}^\mathsf{T} \boldsymbol{u} - t \mid \boldsymbol{X}^\mathsf{T} \boldsymbol{u} > t] \cdot \sqrt{1 + \|\mathbb{E}[\boldsymbol{X} \mid \boldsymbol{X}^\mathsf{T} \boldsymbol{u} > t]\|^2}. \tag{6}$$

---

[2]Technically, we require that the loss is twice differentiable at $\boldsymbol{\theta}^\star$. Due to the ReLU activation, there is a measure 0 set in the parameter space where this is not true. However, if we randomly initialize the weights with a density and use gradient descent with non-vanishing learning rate, then with probability 1 the GD iterations do not visit such non-differentiable points. For the interest of generalization bounds, the behaviors of non-differentiable points are identical to their infinitesimally perturbed neighbor, which is differentiable. For these reasons, this assumption will be implicitly assumed at each candidate $\boldsymbol{\theta}$ in the remainder of the paper.

[3]In particular, this holds for the definition of linear stability where $\mu(\boldsymbol{\theta}^\star) \leq 0$ in the notation of Chemnitz and Engel [2025, p. 7], which is a strictly weaker notion of linear stability than that of Wu et al. [2018] and Ma and Ying [2021] [see the discussion in Chemnitz and Engel, 2025, Appendix A].

Here, $\boldsymbol{X}$ is a random vector drawn uniformly at random from the training examples $\{\boldsymbol{x}_i\}_{i=1}^n$. Note that the distribution $\mathbb{P}_X$ from which $\{\boldsymbol{x}_i\}_{i=1}^n$ are drawn i.i.d. controls the regularity of $g$.

With this weight function in hand, we define a (semi)norm on functions of the form

$$f_{\nu,\boldsymbol{c},c_0}(\boldsymbol{x}) = \int_{\mathbb{S}^{d-1}\times[-R,R]} \phi(\boldsymbol{u}^\mathsf{T}\boldsymbol{x} - t)\,\mathrm{d}\nu(\boldsymbol{u},t) + \boldsymbol{c}^\mathsf{T}\boldsymbol{x} + c_0, \quad \boldsymbol{x}\in\mathbb{R}^d, \tag{7}$$

where $R > 0$, $\boldsymbol{c}\in\mathbb{R}^d$, and $c_0\in\mathbb{R}$. Functions of this form are "infinite-width" neural networks. We define the *weighted variation (semi)norm* as

$$|f|_{\mathrm{V}_g} := \inf_{\substack{\nu\in\mathcal{M}(\mathbb{S}^{d-1}\times[-R,R])\\ \boldsymbol{c}\in\mathbb{R}^d, c_0\in\mathbb{R}}} \|g\cdot\nu\|_{\mathcal{M}} \quad \text{s.t.} \quad f = f_{\nu,\boldsymbol{c},c_0}, \tag{8}$$

where, if there does not exist a representation of $f$ in the form of (7), then the seminorm[4] is understood to take the value $+\infty$. Here, $\mathcal{M}(\mathbb{S}^{d-1}\times[-R,R])$ denotes the Banach space of (Radon) measures and, for $\mu\in\mathcal{M}(\mathbb{S}^{d-1}\times[-R,R])$, $\|\mu\|_{\mathcal{M}} := \int_{\mathbb{S}^{d-1}\times[-R,R]}\mathrm{d}|\mu|(\boldsymbol{u},t)$ is the measure-theoretic total-variation norm.

With this seminorm, we define the Banach space of functions $\mathrm{V}_g(\mathbb{B}_R^d)$ on the ball $\mathbb{B}_R^d := \{\boldsymbol{x}\in\mathbb{R}^d : \|\boldsymbol{x}\|_2 \leq R\}$ as the set of all functions $f$ such that $|f|_{\mathrm{V}_g}$ is finite. When $g\equiv 1$, $|\cdot|_{\mathrm{V}_g}$ and $\mathrm{V}_g(\mathbb{B}_R^d)$ coincide with the variation (semi)norm and variation norm space of Bach [2017].

**Example 3.1.** *Since we are interested in functions defined on $\mathbb{B}_R^d$, for a finite-width neural network $f_{\boldsymbol{\theta}}(\boldsymbol{x}) = \sum_{k=1}^K v_k\phi(\boldsymbol{w}_k^\mathsf{T}\boldsymbol{x} - b_k) + \beta$, we observe that it has the equivalent implementation as $f_{\boldsymbol{\theta}}(\boldsymbol{x}) = \sum_{j=1}^J a_j\phi(\boldsymbol{u}_j^\mathsf{T}\boldsymbol{x} - t_j) + \boldsymbol{c}^\mathsf{T}\boldsymbol{x} + c_0$, where $a_j\in\mathbb{R}$, $\boldsymbol{u}_j\in\mathbb{S}^{d-1}$, $t_j\in\mathbb{R}$, $\boldsymbol{c}\in\mathbb{R}^d$, and $c_0\in\mathbb{R}$. Indeed, this is due to the fact that the ReLU is homogeneous, which allows us to absorb the magnitude of the input weights into the output weights (i.e., each $a_j = |v_{k_j}|\|\boldsymbol{w}_{k_j}\|_2$ for some $k_j\in\{1,\ldots,K\}$). Furthermore, any ReLUs in the original parameterization whose activation threshold[5] is outside $\mathbb{B}_R^d$ can be implemented by an affine function on $\mathbb{B}_R^d$, which gives rise to the $\boldsymbol{c}^\mathsf{T}\boldsymbol{x} + c_0$ term in the implementation. If this new implementation is in "reduced form", i.e., the collection $\{(\boldsymbol{u}_j, t_j)\}_{j=1}^J$ are distinct, then we have that $|f_{\boldsymbol{\theta}}|_{\mathrm{V}_g} = \sum_{j=1}^J |a_j| g(\boldsymbol{u}_j, t_j)$.*

This example reveals that this seminorm is a weighted path norm of a neural network and, in fact, coincides with the path norm when $g\equiv 1$ [Neyshabur et al., 2015]. It also turns out that the data-dependent regularity induced by this seminorm is tightly linked to the flatness of a neural network minimum. We summarize this fact in the next theorem.

**Theorem 3.2.** *Suppose that $f_{\boldsymbol{\theta}}$ is a two-layer neural network such that the loss $\mathcal{L}(\cdot)$ is twice differentiable at $\boldsymbol{\theta}$. Then, $|f_{\boldsymbol{\theta}}|_{\mathrm{V}_g} \leq \frac{\lambda_{\max}(\nabla_{\boldsymbol{\theta}}^2\mathcal{L}(\boldsymbol{\theta}))}{2} - \frac{1}{2} + (R+1)\sqrt{2\mathcal{L}(\boldsymbol{\theta})}$.*

The proof of this theorem appears in Appendix C. This theorem reveals that flatness implies regularity in the sense of the variation space $\mathrm{V}_g(\mathbb{B}_R^d)$. In particular, we also have an immediate corollary for stable minima thanks to Proposition 2.1.

**Corollary 3.3.** *For any $\boldsymbol{\theta}\in\Theta_{\mathrm{flat}}(\eta; \mathcal{D})$, $|f_{\boldsymbol{\theta}}|_{\mathrm{V}_g} \leq \frac{1}{\eta} - \frac{1}{2} + (R+1)\sqrt{2\mathcal{L}(\boldsymbol{\theta})}$.*

The main takeaway messages from Theorem 3.2 and Corollary 3.3 are that flat / stable solutions are smooth in the sense of $\mathrm{V}_g(\mathbb{B}_R^d)$. In particular, we see that the Banach space $\mathrm{V}_g(\mathbb{B}_R^d)$ is the natural function space to study stable minima. This framework provides the mathematical foundation and sets the stage to investigate the generalization and statistical performance of stable minima.

We also note that, from Corollary 3.3 and Example 3.1, for stable solutions $f_{\boldsymbol{\theta}}$, as the step size $\eta$ grows, the function $f_{\boldsymbol{\theta}}$ becomes smoother, eventually approaching an affine function as $\eta\to\infty$. This can be viewed as an example of the *simplicity bias* phenomenon of GD training [Arpit et al., 2017, Kalimeris et al., 2019, Valle-Perez et al., 2019].

---

[4] We use the notation $|\cdot|$ instead of $\|\cdot\|$ to highlight that this quantity is a seminorm. This quantity is a seminorm since affine functions are in its null space. See Kůrková and Sanguineti [2001, 2002], Mhaskar [2004], Bach [2017], Siegel and Xu [2023], Shenouda et al. [2024] for more details about variation spaces.

[5] The activation threshold of a neuron $\phi(\boldsymbol{w}^\mathsf{T}\boldsymbol{x} - b)$ is the hyperplane $\{\boldsymbol{x}\in\mathbb{R}^d : \boldsymbol{w}^\mathsf{T}\boldsymbol{x} = b\}$.

Finally, we note that the function-space regularity induced by $V_g(\mathbb{B}_R^d)$ has an equivalent analytic description via a weighted norm in the domain of the Radon transform of the function. This analytic description is based on the $\mathscr{R}$-(semi)norm/second-order Radon-domain total variation inductive bias of infinite-width two-layer neural networks [Ongie et al., 2020, Parhi and Nowak, 2021, Bartolucci et al., 2023]. Before stating our theorem, we first recall the definition of the Radon transform. The Radon transform of a function $f \in L^1(\mathbb{R}^d)$ is given by

$$\mathscr{R}\{f\}(\boldsymbol{u}, t) = \int_{\boldsymbol{u}^\mathsf{T}\boldsymbol{x}=t} f(\boldsymbol{x})\,\mathrm{d}\boldsymbol{x}, \quad (\boldsymbol{u}, t) \in \mathbb{S}^{d-1} \times \mathbb{R}, \tag{9}$$

where the integration is against the $(d-1)$-dimensional Lebesgue measure on the hyperplane $\boldsymbol{u}^\mathsf{T}\boldsymbol{x} = t$. Thus, we see that the Radon transform integrates functions along hyperplanes.

**Theorem 3.4.** *For every* $f \in V_g(\mathbb{B}_R^d)$, *consider the canonical extension*[6] $f_{\mathrm{ext}} : \mathbb{R}^d \to \mathbb{R}$ *via its integral representation* (7). *It holds that* $|f|_{V_g} = \|g \cdot \mathscr{R}(-\Delta)^{\frac{d+1}{2}} f_{\mathrm{ext}}\|_{\mathcal{M}}$, *where fractional powers of the Laplacian are understood via the Fourier transform.*

The proof of this theorem appears in Appendix D. We remark that the operators that appear in the theorem must be understood in the distributional sense (i.e., by duality). We refer the reader to Parhi and Unser [2024] for rigorous details about the distributional extension of the Radon transform. We also remark that a version of this theorem also appeared in Nacson et al. [2023, Theorem 1], but we note that their problem setting was the implicit bias of minima stability in the interpolation regime.

### 3.1 Stable Solutions Generalize But Suffer the Curse of Dimensionality

In the remainder of this paper, we focus on the scenario where inputs $\{\boldsymbol{x}_i\}_{i=1}^n$ are drawn i.i.d. uniformly from the unit ball $\mathbb{B}_1^d$ (i.e., $R = 1$). Under this assumption, the *population version* of the weight function, which we denote as $g_P$, has a well-defined asymptotic behavior. As detailed in Appendix E, for $|t| \geq 1$, $g_P(\boldsymbol{u}, t) = 0$, and as $|t| \to 1^-$, $g_P(\boldsymbol{u}, t) \asymp (1 - |t|)^{d+2}$. While the actual weight function $g$ in our analysis remains the empirical one derived from the data, this population behavior serves as a crucial analytical guide. Our proofs will show that the empirical $g$ concentrates around this population version (Appendix E.2). For clarity in expressing our main results and their dependence on dimensionality, we will characterize the function space of stable minima with respect to a canonical weight function $g(\boldsymbol{u}, t) := (1 - |t|)^{d+2}$, which captures this essential asymptotic property.

With this in hand, we can now characterize the *generalization gap* of stable minima, which is defined to be the absolute difference between the training loss and the population risk. We are able to characterize the generalization gap under mild conditions on the joint distribution of the training examples and the labels.

**Theorem 3.5.** *Let* $\mathcal{P}$ *denote the joint distribution of* $(\boldsymbol{x}, y)$. *Assume that* $\mathcal{P}$ *is supported on* $\mathbb{B}_1^d \times [-D, D]$ *for some* $D > 0$ *and that the marginal distribution of* $\boldsymbol{x}$ *is* $\mathrm{Uniform}(\mathbb{B}_1^d)$. *Fix a data set* $\mathcal{D} = \{(\boldsymbol{x}_i, y_i)\}_{i=1}^n$, *where each* $(\boldsymbol{x}_i, y_i)$ *is drawn i.i.d. from* $\mathcal{P}$. *Then, with probability* $\geq 1 - \delta$ *we have that for the plug-in risk estimator* $\hat{R}(f) := \frac{1}{n} \sum_{i=1}^n (f(\boldsymbol{x}_i) - y_i)^2$

$$\sup_{\boldsymbol{\theta} \in \Theta_{\mathrm{flat}}(\eta; \mathcal{D})} \mathrm{GeneralizationGap}(f_{\boldsymbol{\theta}}; \hat{R}) := \left| \mathbb{E}_{(\boldsymbol{x},y)\sim\mathcal{P}} \left[ (f_{\boldsymbol{\theta}}(\boldsymbol{x}) - y)^2 \right] - \hat{R}(f_{\boldsymbol{\theta}}) \right|$$

$$\lesssim_d \left( \frac{1}{\eta} - \frac{1}{2} + 4M \right)^{\frac{d}{d^2+4d+3}} M^2\, n^{-\frac{1}{2d+4}}, \tag{10}$$

*where* $M := \max\{D, \|f_{\boldsymbol{\theta}}\|_{L^\infty(\mathbb{B}_1^d)}, 1\}$ *and* $\lesssim_d$ *hides constants (which could depend on* $d$) *and logarithmic factors in* $n$ *and* $(1/\delta)$. *Furthermore, for any* $L \geq D$, *it holds that*

$$\inf_{\tilde{R}} \sup_{\mathcal{P}} \mathbb{E}_{\mathcal{D}\sim\mathcal{P}^{\otimes n}} \left[ \sup_{\substack{\boldsymbol{\theta} \in \Theta_{\mathrm{flat}}(\eta; \mathcal{D}) \\ \|f_{\boldsymbol{\theta}}\|_{L^\infty(\mathbb{B}_1^d)} \leq L}} \mathrm{GeneralizationGap}(f_{\boldsymbol{\theta}}; \tilde{R}) \right] \gtrsim_d L^2 n^{-\frac{2}{d+1}}, \tag{11}$$

---

[6] Since functions in $V_g(\mathbb{B}_R^d)$ are only defined on $\mathbb{B}_R^d$, we must consider their extension to $\mathbb{R}^d$ when working with the Radon transform. See Parhi and Nowak [2023b, Section IV] for more details.

*where the* inf *is over all risk estimators,* $\gtrsim_d$ *hides constants (which could depend on d), and the* sup *is over all distributions that satisfy the above hypotheses.*

The proof of this theorem appears in Appendix F. While this theorem does show that as $n \to \infty$, the generalization gap vanishes, it reveals that the sample complexity grows exponentially with the input dimension (as seen by the $n^{-\frac{1}{2d+4}}$ term in the upper bound and the $n^{-\frac{2}{d+1}}$ term in the lower bound). This suggests that the curse-of-dimensionality is intrinsic to the stable minima set $\Theta_{\mathrm{flat}}(\eta; \mathcal{D})$—not an artifact of our mathematical analysis nor the naive plug-in empirical risk estimator being suboptimal. On the other hand, for low-norm solutions (in the sense that they minimize the weight-decay objective), it can be shown that the generalization gap decays at a rate of $\widetilde{O}(n^{-\frac{1}{4}})$, where $\widetilde{O}(\cdot)$ hides logarithmic factors [cf., Bach, 2017, Parhi and Nowak, 2023b]. This uncovers an exponential gap between flat and low-norm solutions, and, in particular, that stable solutions suffer the curse of dimensionality. When $d = 1$, this result also provides a strict generalization of Qiao et al. [2024, Theorem 4.3], as they measure the error strictly inside the input domain, rather than on the full input domain. Thus, our result also characterizes how stable solutions *extrapolate*.

## 3.2 Nonparametric Function Estimation With Stable Minima

We now turn to the problem of nonparametric function estimation. As we have seen that $V_g(\mathbb{B}_1^d)$ is a natural model class for stable minima, this raises two fundamental questions: (i) How well do stable minima estimate functions in $V_g(\mathbb{B}_1^d)$ from noisy data? and (ii) What is the best performance any estimation method could hope to achieve for functions in $V_g(\mathbb{B}_1^d)$. In this section we provide answers to both these questions by deriving a mean-squared error upper bound for stable minima and a minimax lower bound for this function class.

**Theorem 3.6.** *Fix a step size $\eta > 0$ and noise level $\sigma > 0$. Given a ground truth function $f_0 \in V_g(\mathbb{B}_1^d)$ such that $\|f_0\|_{L^\infty} \leq B$ and $|f_0|_{V_g} \leq \widetilde{O}\left(\frac{1}{\eta} - \frac{1}{2} + 2\sigma\right)$, suppose that we are given a data set $y_i = f_0(\boldsymbol{x}_i) + \varepsilon_i$, where $\boldsymbol{x}_i$ are i.i.d. $\mathrm{Uniform}(\mathbb{B}_1^d)$ and $\varepsilon_i$ are i.i.d. $\mathcal{N}(0, \sigma^2)$. Then, with probability $\geq 1 - \delta$, we have that*

$$\frac{1}{n} \sum_{i=1}^{n} (f_{\boldsymbol{\theta}}(\boldsymbol{x}_i) - f_0(\boldsymbol{x}_i))^2 \; \lesssim_d \; \left(\frac{1}{\eta} - \frac{1}{2} + 2\sigma\right)^{\frac{d}{(2d^2+6d+3)(d+2)}} B^2 \left(\frac{\sigma^2}{n}\right)^{\frac{1}{2d+4}}, \qquad (12)$$

*for any $\boldsymbol{\theta} \in \Theta_{\mathrm{flat}}(\eta; \mathcal{D})$ that is optimized, i.e., $(f_{\boldsymbol{\theta}}(\boldsymbol{x}_i) - y_i)^2 \leq (f_0(\boldsymbol{x}_i) - y_i)^2$, for $i = 1, \ldots, n$.*

The proof of this theorem appears in Appendix G. This theorem shows that *optimized* stable minima incur an estimation error rate that decays as $\widetilde{O}(n^{-\frac{1}{2d+4}})$, which suffers the curse of dimensionality. The optimized assumption is mild as it only asks that the error for each data point is smaller than the label noise $\sigma^2$, which is easy to achieve in practice with GD training, especially in the overparameterized regime. The next theorem shows that the curse of dimensionality is actually necessary for this function class.

**Theorem 3.7.** *Consider the same data-generating process as in Theorem 3.6. We have the following minimax lower bounds.*

$$\inf_{\hat{f}} \sup_{\substack{f \in V_g(\mathbb{B}_1^d) \\ \|f\|_{L^\infty} \leq B, |f|_{V_g} \leq C}} \mathbb{E}\|\hat{f} - f\|_{L^2}^2 \gtrsim_d \begin{cases} \min(B, C)^2 \left(\frac{\sigma^2}{n}\right)^{\frac{2}{d+1}}, & d > 1, \\ \min(B, C)^2 \left(\frac{\sigma^2}{n}\right)^{\frac{1}{2}}, & d = 1. \end{cases} \qquad (13)$$

*where $\gtrsim_d$ hides constants (that could depend on d).*

The proof of this theorem appears in Appendix H. Our proof relies on two high-dimensional constructions. The first construction is to pack the unit sphere $\mathbb{S}^{d-1}$ with $M = \exp(\Omega(d))$ pairwise-disjoint spherical caps, each specified by a unit vector $\boldsymbol{u}_i$ as its center. Then, for every center $\boldsymbol{u}_i$ the ReLU neuron $\varphi_i(\boldsymbol{x}) = c\phi(\boldsymbol{u}_i^\mathsf{T} \boldsymbol{x} - t)$ is active only on its outward-facing cap, and attains its peak value $\min\{B, C\}$ by choosing a suitable $t$. The second construction is to observe that since the weight function $g(\boldsymbol{u}, t)$ decreases quickly as $|t| \to 1$ (see Appendix E), the regularity constraint $|\cdot|_{V_g} \leq C$ allows us to combine an exponential number of such atoms to construct a family of "hard-to-learn" functions. Traditional lower-bound constructions satisfy regularity by shrinking bump amplitudes

(vertical changes), whereas our approach fundamentally differs by shifting and resizing bump supports (horizontal changes). Our experiments reveal that stable minima actually favor these kinds of hard-to-learn functions and we refer to this as the *neural shattering* phenomenon.

## 4 Experiments

In this section, we empirically validate our claims that (i) stable minima are not immune to the curse of dimensionality and (ii) the "neural shattering" phenomenon occurs. All synthetic data points are generated by uniformly sampling $x$ from $\mathbb{B}_1^d$ and $y_i = f_0(x_i) + \mathcal{N}(0, \sigma^2)$, where the ground-truth function $f_0(x) = w^\mathsf{T}x$ for some fixed vector $w$ with $\|w\| = 1$. All the models are two-layer ReLU neural networks with width four times the training data size. The networks are randomly initialized by the standard Kaiming initialization [He et al., 2015]. We also use gradient clipping with threshold 50 to avoid divergence for large learning rates.[7]

**Curse of dimensionality.** In this experiment, we train neural networks with GD and vary the data set sizes in $\{32, 64, 128, 256, 512\}$ and dimensions in $\{1, 2, 3, 4, 5\}$, with noise level $\sigma = 1$. For each data set size, dimension, and training parameters ($\eta = 0.2$ without weight decay and $\eta = 0.01$ with weight-decay $0.1$), we conduct 5 experiments and take the median. The log-log curves are displayed in Figure 2.

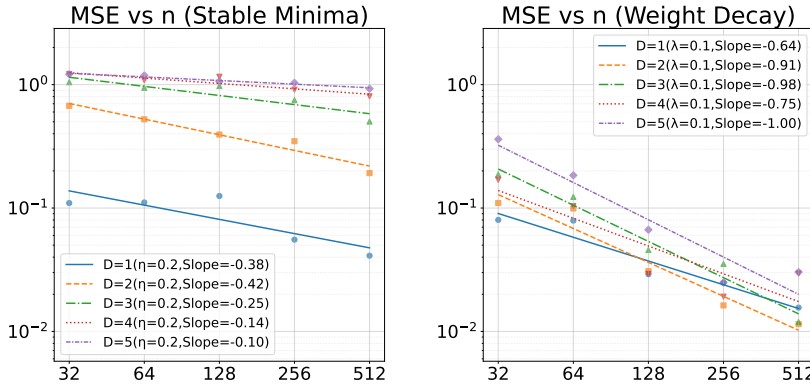

Figure 2: Empirical validation of the curse of dimensionality. **Left panel:** The slope of $\log \mathrm{MSE}$ versus $\log n$ for training with vanilla gradient descent rapidly decreases with dimension, falling to about 0.1 at $d = 5$. **Right panel:** Training with $\ell^2$ (weight decay) results in slopes above 0.5 in the log–log scale.

**Neural shattering.** As briefly illustrated in the right panel of Figure 1, Figure 3 presents more detailed experiments. We train a two-layer ReLU network of width 2048 on 512 noisy samples ($\sigma = 1$) of a 10-dimensional linear target. Under a large step size $\eta = 0.9$ (no weight decay), gradient descent enters a flat / stable minimum ($\lambda_{\max}(\nabla_\theta^2 \mathcal{L}(\theta))$ oscillates around $2/\eta \approx 2.2$, signaling edge-of-stability dynamics). This drastically reduces each neuron's data-activation rate to $\leq 10\%$, rather than reducing their weight norms. The network overfits (train MSE $\approx 1.105$, matching the noise level). In contrast, with $\eta = 0.01$ plus $\ell^2$-weight-decay $\lambda = 0.1$, all neurons remain active and weight norms stay tightly bounded, so the model avoids overfitting (train MSE $\approx 0.055$).

## 5 Discussion and Conclusion

This paper presents a nuanced conclusion on the link between minima stability and generalization: Stable solutions do generalize, but when data is distributed uniformly on a ball, this generalization ability is severely weakened by the Curse of Dimensionality (CoD). Our analysis pinpoints the mechanism behind this failure. The implicit regularization from GD is not uniform across the input

---

[7]We monitor clipping during the training, and the clipping only occurs in the first 10 epochs. Gradient clipping does not prevent the training dynamics from entering edge-of-stability regime.

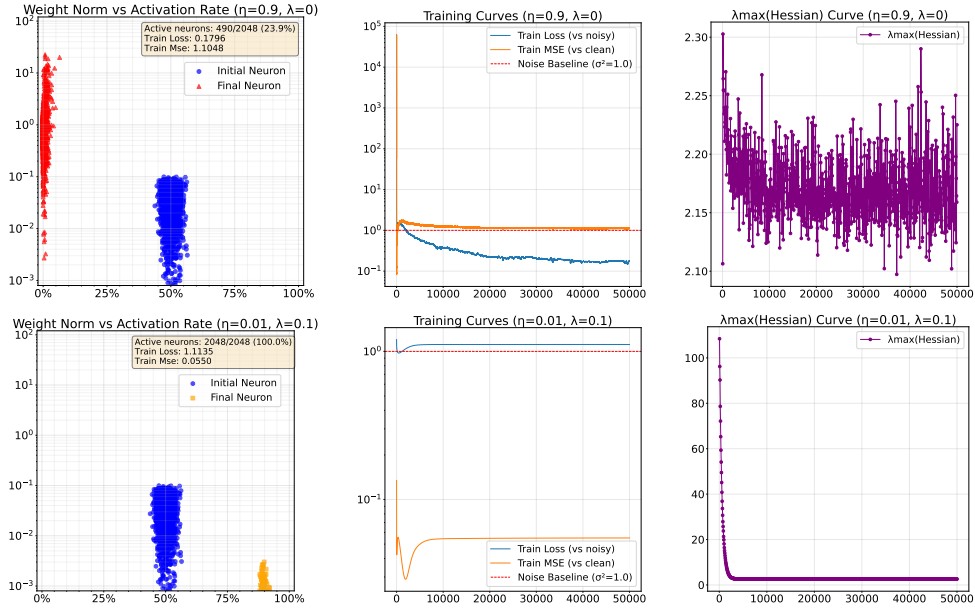

Figure 3: The top-left plot illustrates the *neural shattering* phenomenon: after large-step training each ReLU neuron (orange) is active on only a tiny fraction of the data (small horizontal support) yet its weight norm remains large, exactly as in our sphere-packing lower-bound construction where each outward-facing ReLU atom fires on very few inputs but retains full peak amplitude.

domain. While it imposes strong regularity in the strict interior of the data support, this guarantee collapses at the boundary. This localized failure of regularization is precisely what enables "neural shattering", a phenomenon where neurons satisfy the stability condition not by shrinking their weights, but by minimizing their activation frequency. This causes the CoD: The intrinsic geometry of a high-dimensional ball provides an exponential increase in available directions for shattering to occur, while the boundary regularization simultaneously weakens exponentially as the input dimension $d$ grows. This mechanism, confirmed by both our lower bounds and experiments, explains why stable solutions exhibit poor generalization in high dimensions.

Several simplifications limit the scope of these results. The theory treats only two-layer ReLU networks and relies on the idealized assumption that samples are drawn uniformly from the unit ball. For more general distributions, the induced weight function $g$ inherits the full geometry of the data and becomes harder to describe and interpret. Understanding this effect, together with extending the analysis to deeper architectures and adaptive algorithms, will take substantial effort, which we leave as future work.

# 6   Acknowledgments

The research was partially supported by NSF Award # 2134214. The authors acknowledge early discussion with Peter Bartlett at the Simons Foundation that motivated us to consider the problem. Tongtong Liang thanks Zihan Shao for providing helpful suggestions on the implementation of the experiments.

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

## Supplementary Materials

# A  Additional Experiments

## A.1  Empirical Evidence: High Dimensionality Yields Neural Shattering

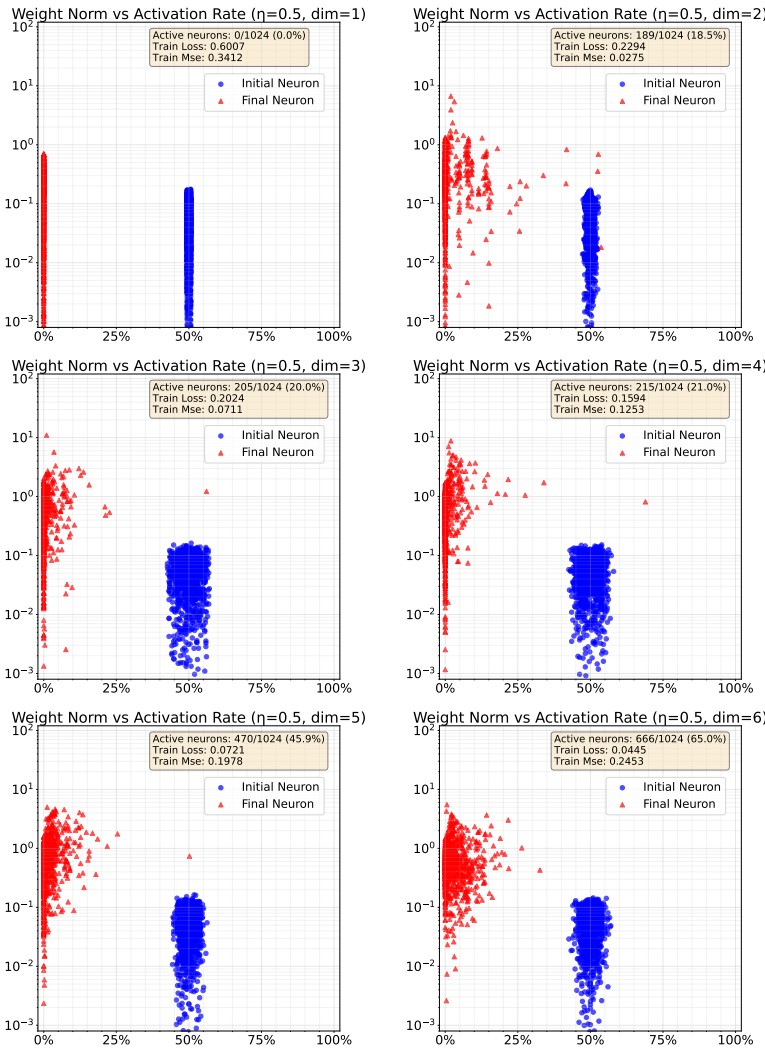

Figure 4: Comparison across input dimension $d$ for a two-layer ReLU network of width 1024 trained on 512 samples for 20000 epochs with learning rate $\eta = 0.5$. At $d = 1$, all neurons extrapolate (0% active), while as $d$ increases the fraction of neurons surviving training rises dramatically (up to 65% at $d = 6$). Simultaneously, the training loss monotonically decreases whereas the training MSE increases with $d$, demonstrating that neural shattering under large learning rates may be the key driver of the curse of dimensionality in stable minima.

## A.2 Neural Shattering and Learning Rate (dim=5)

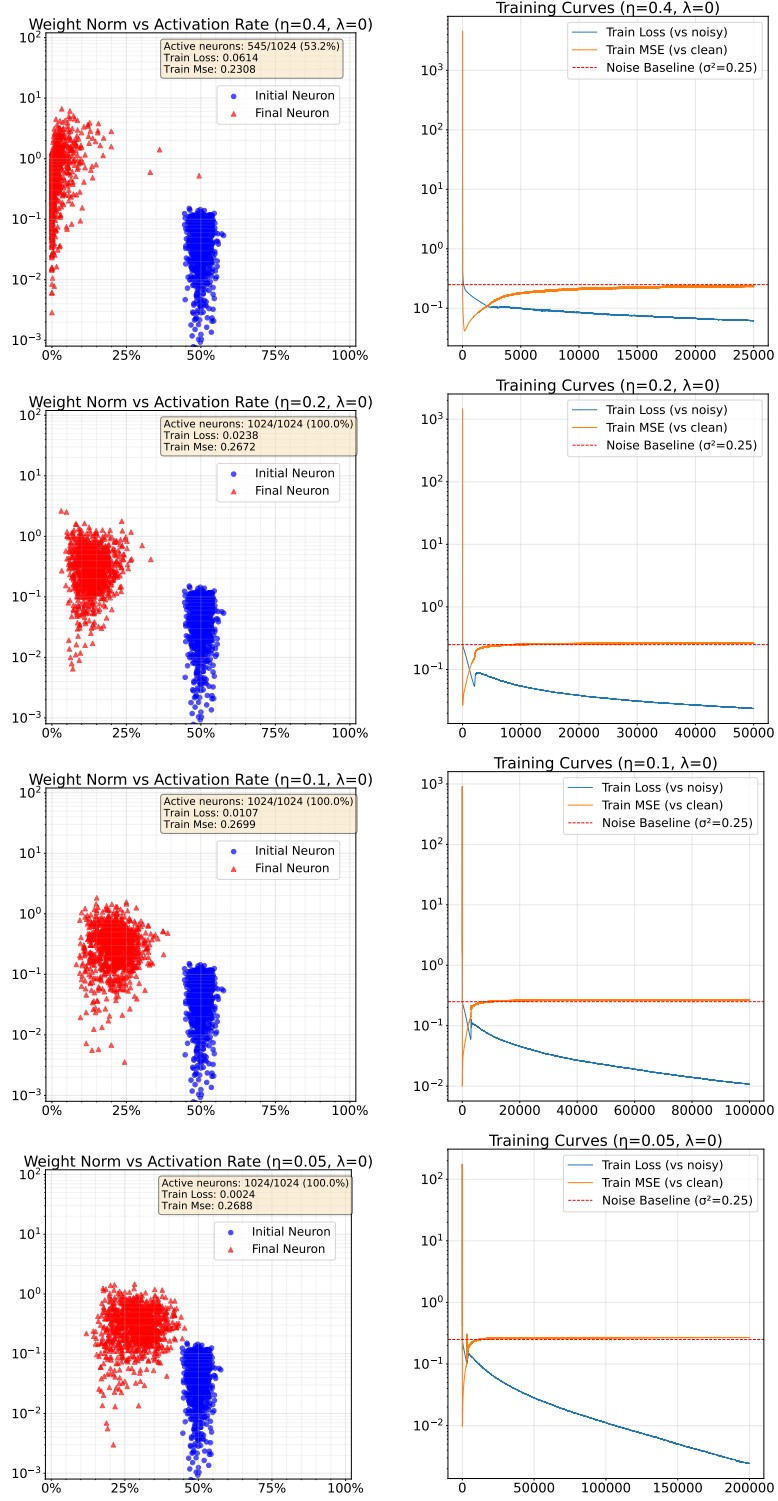

Figure 5: Effect of increasing learning rate $\eta$ on shattering ($\eta \times \text{epochs} = 10000$): as $\eta$ grows, the stability/flatness constraint forces an ever larger fraction of neurons to activate only on a small subset of the data (neural shattering). To further decrease the training loss, gradient descent correspondingly increases the weight norms of the remaining active neurons.

## A.3  Neural Shattering in the Underparametrized Regime

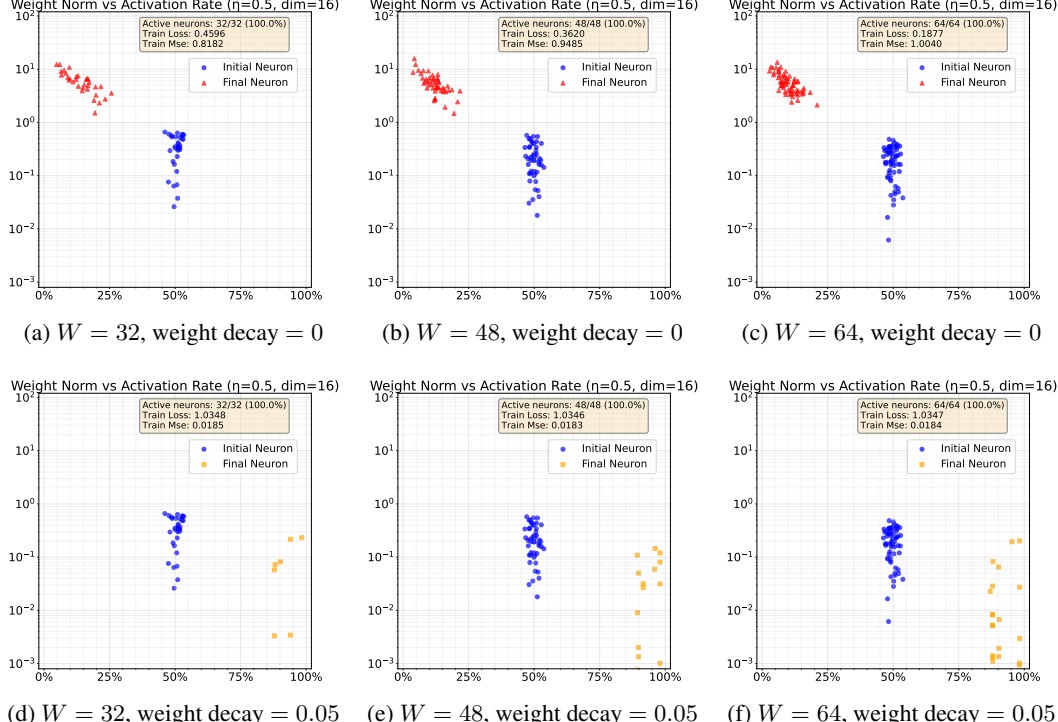

(a) $W = 32$, weight decay $= 0$     (b) $W = 48$, weight decay $= 0$     (c) $W = 64$, weight decay $= 0$

(d) $W = 32$, weight decay $= 0.05$     (e) $W = 48$, weight decay $= 0.05$     (f) $W = 64$, weight decay $= 0.05$

Figure 6: Persistence of neural shattering across width in the underparametrized regime. Each panel plots per-neuron activation rate versus weight norm after training a two-layer ReLU network on $n = 1024$ samples with learning rate $\eta = 0.5$. Columns correspond to widths $W \in \{32, 48, 64\}$; the top row uses no weight decay, and the bottom row uses mild decay ($\lambda = 0.05$). The parameter count is $Wd + W + 1$, so $W = 64$ is mildly overparameterized while $W = 32, 48$ are strictly underparameterized. Across all widths, we observe clear neural shattering: neurons with smaller activation fractions carry larger weight norms. This monotone trend is especially visible in the stable-minima panels ($\lambda = 0$), exactly as predicted by our theory. The weight-decay panels serve only as a high-activation baseline to calibrate what "few activations" means, underscoring how exceptionally low the activation rates are at stable minima.

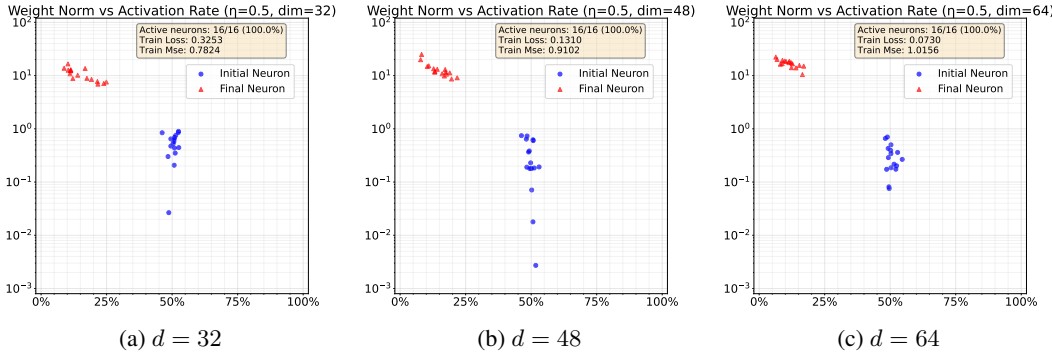

(a) $d = 32$            (b) $d = 48$            (c) $d = 64$

Figure 7: Each panel shows the relation between neuron activation rate and weight norm after training a two-layer ReLU network of width $W = 16$ on $n = 1024$ samples drawn from a linear target with learning rate $\eta = 0.5$ for $d \in \{64, 48, 32\}$. This observation indicates that neural shattering is a generic feature of stable minima, robust even when in small network.

## A.4 Neural Shattering for GELU Networks

This set of stable minima serves as a prism through which we can understand the emergent behaviors of the learning process. The data-dependent weight function $g(\boldsymbol{u}, t)$, which is central to our analysis, arises directly from the structure of the loss Hessian and provides a static characterization of the implicit bias of gradient dynamics. A smaller value of $g(\boldsymbol{u}, t)$ for a neuron $(\boldsymbol{u}, t)$ implies that the stability condition imposes a weaker regularization, allowing for larger weight magnitudes for that neuron.

This static view is intimately connected to the underlying learning dynamics. In high-dimensional spaces, a neuron's activation boundary can easily drift to a region where it activates on only a small fraction of the data points. For such a neuron, the gradients it receives are sparse and localized. If the few data points it activates on are already well-fitted, the local gradient signal can vanish, causing the neuron's parameters to become effectively "stuck" or stable. The small value of $g(\boldsymbol{u}, t)$ in these boundary regions creates "space" within the class of stable functions for these trapped, high-magnitude, yet sparsely-activating neurons to exist, a possibility our lower bound construction then formalizes and exploits.

The ReLU activation function is analytically convenient for this analysis because its hard-sparsity property: a strictly zero gradient for non-activating inputs. This leads to a sparse loss Hessian, allowing for a clean derivation of the weight function $g(\boldsymbol{u}, t)$. However, the underlying "stuck neuron" dynamic is not necessarily unique to ReLU. Activations like GELU provide a non-zero gradient for negative inputs, but this signal is weak and decays quickly away from the activation boundary. It is therefore plausible that this weak gradient is insufficient to pull a "stuck" neuron back from the data boundary once it has drifted there and its activation rate has diminished. This suggests that the fundamental mechanism enabling "neural shattering" may persist. This hypothesis motivates an empirical investigation into whether the same phenomena of neural shattering also manifests in networks trained with GELU activations.

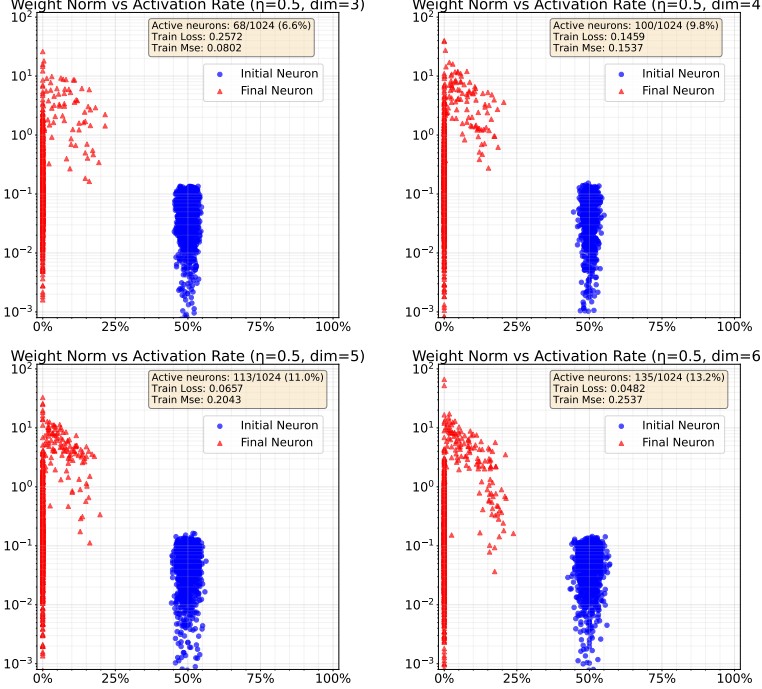

Figure 8: Comparison across input dimension $d$ for a two-layer **GELU** network of width 1024 trained on 512 samples for 20000 epochs with learning rate $\eta = 0.5$. The neural shattering behavior observed for ReLU networks in Figure 4 also appears here with GeLU activations. In particular, we can see the trend more clearly: neurons with lower activation rates tend to develop larger weight norms, highlighting that the neural shattering mechanism extends beyond piecewise-linear activations.

## A.5 Empirical Analysis of the Curse of Dimensionality (I)

We conduct the following experiments in the setting where the ground-truth function is linear with Gaussian noise $\sigma^2 = 1$. The width of neural network is 4 times of the number of samples.

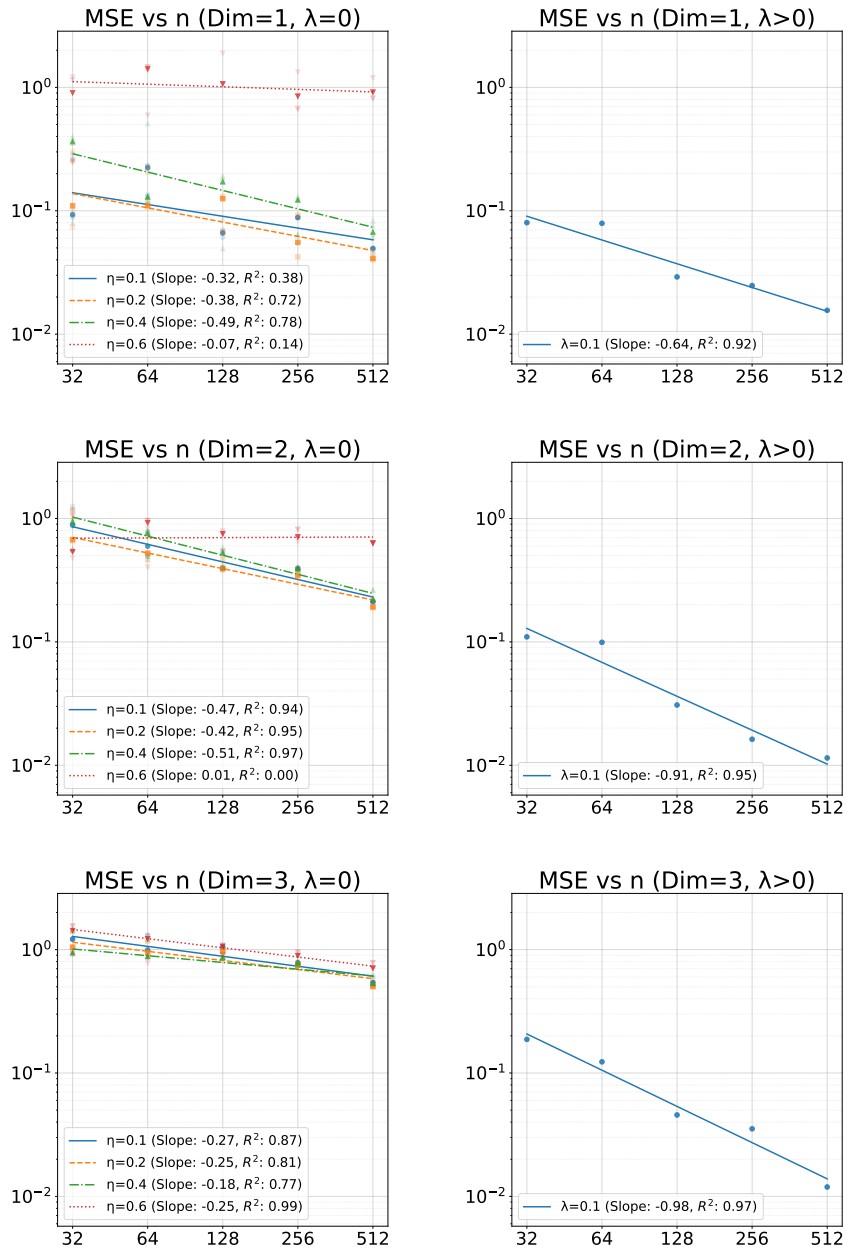

Figure 9: Log–log plots of the mean squared error (MSE) versus sample size $n$ (Part I). Each curve is regressed by the median result over five random initializations (lighter markers), while the shallow markers denote the other runs. As the input dimension increases, the slope of the fitted regression line becomes progressively shallower, indicating slower error decay.

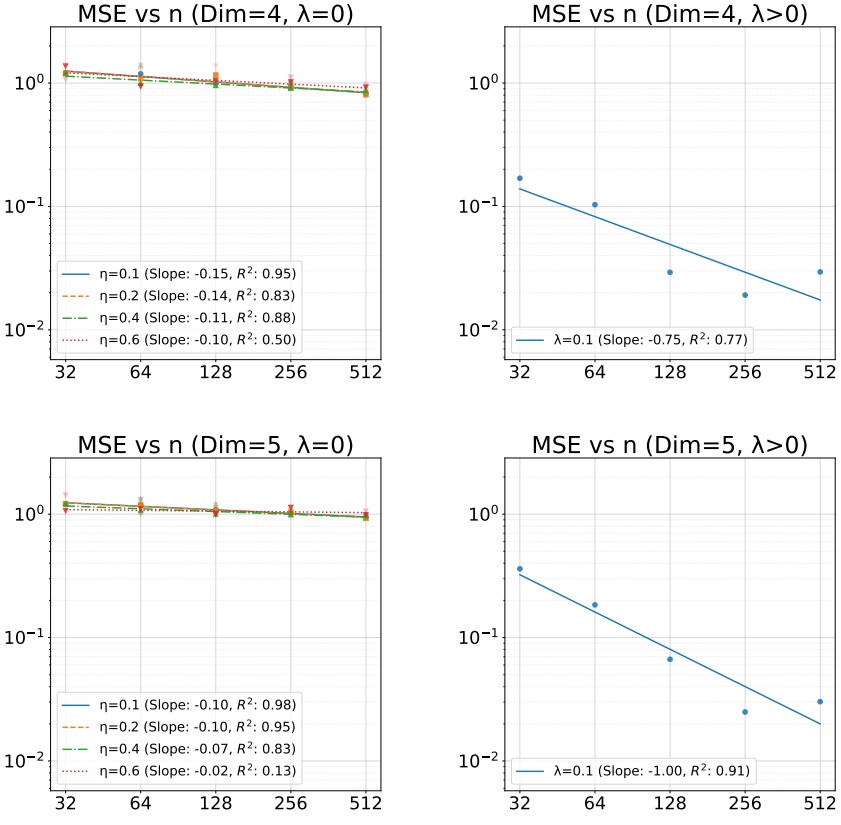

Figure 10: Log–log plots of the mean squared error (MSE) versus sample size $n$, illustrating the curse of dimensionality in stable minima (Part II). We can see in dimension 5, the slope is almost flat and even the large-step size cannot save the results (even worse than small step-size).

## A.6 Empirical Analysis of the Curse of Dimensionality (II)

We conduct the following experiments in the setting where the ground-truth function is linear with Gaussian noise $\sigma^2 = 0.25$. The width of neural network is 2 times of the number of samples.

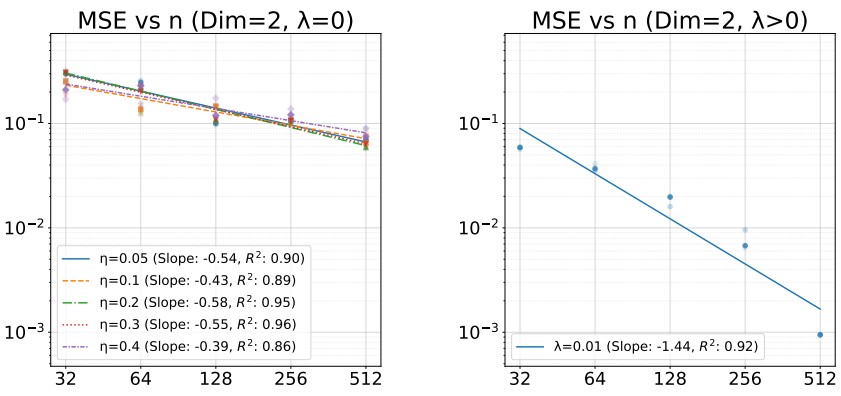

Figure 11: Log–log plots of the mean squared error (MSE) versus sample size $n$ (Part III). Compared to the previous experiments, this setup reduces the noise level to $\sigma = 0.5$, applies weight decay $\lambda = 0.01$, and constrains the model width to $2n$.

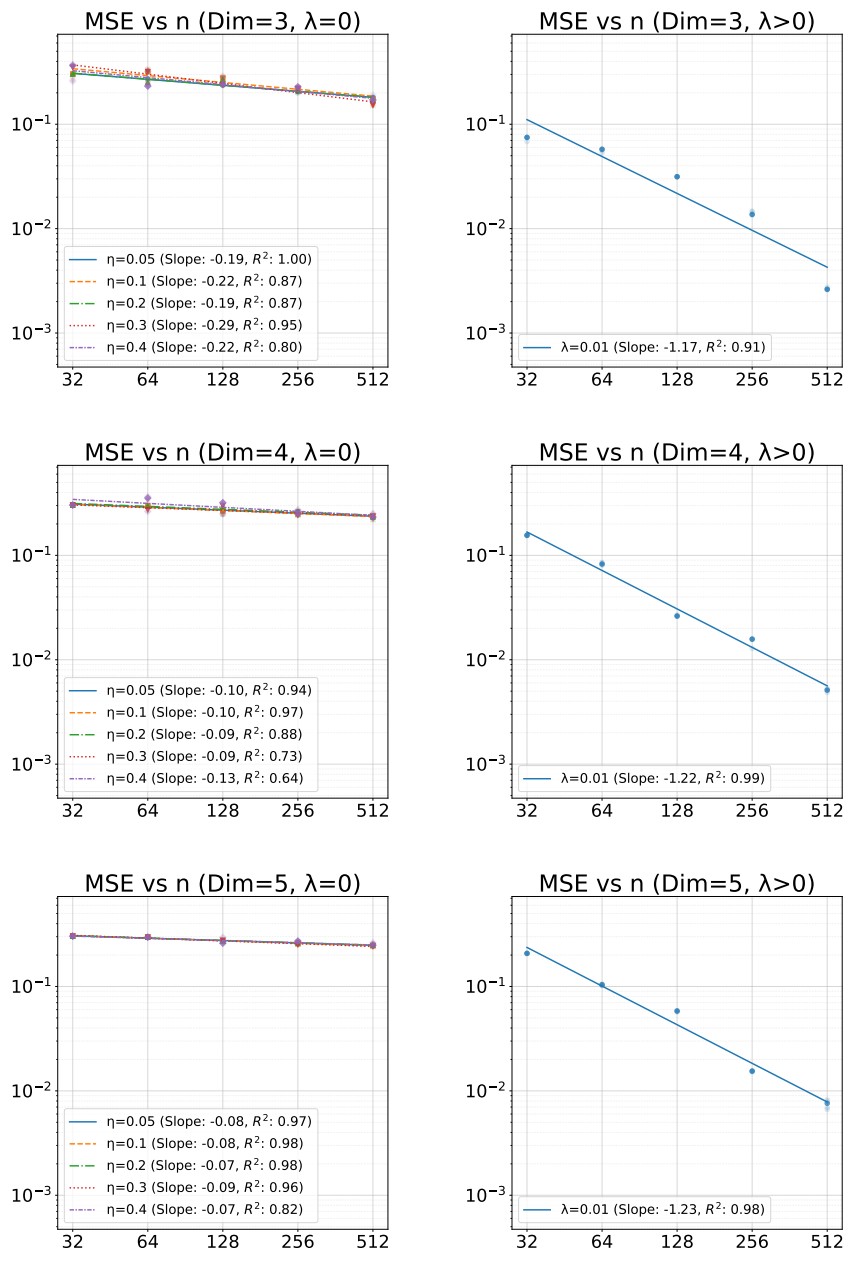

Figure 12: Log–log plots of the mean squared error (MSE) versus sample size $n$ (Part IV). The log–log MSE vs. $n$ curves still exhibit progressively flattening slopes as the input dimension grows, demonstrating the enduring curse of dimensionality in stable minima.

## A.7 Empirical Analysis of the Curse of Dimensionality (III)

We conduct the following experiments in the setting where the ground-truth function is Hölder(1/2) $f(\boldsymbol{x}) = \frac{1}{d} \sum_{i=1}^{d} |\boldsymbol{u}_j^\mathsf{T} \boldsymbol{x}|^{1/2} + 1$ with Gaussian noise $\sigma^2 = 0.25$, where $\boldsymbol{u}_j$ is uniformly sampled from $\mathbb{S}^{d-1}$. The width of neural network is 2 times of the number of samples.

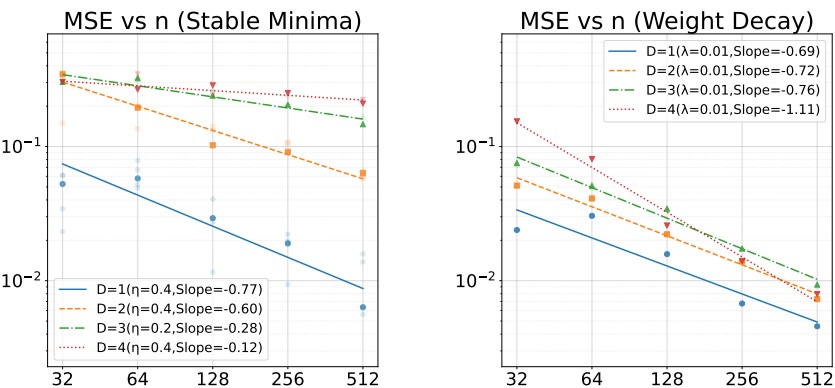

Figure 13: Log–log plots of the mean squared error (MSE) versus sample size $n$ (Part V). We can see the generalization slopes of stable minima degrades as dimension increase from 1 to 4.

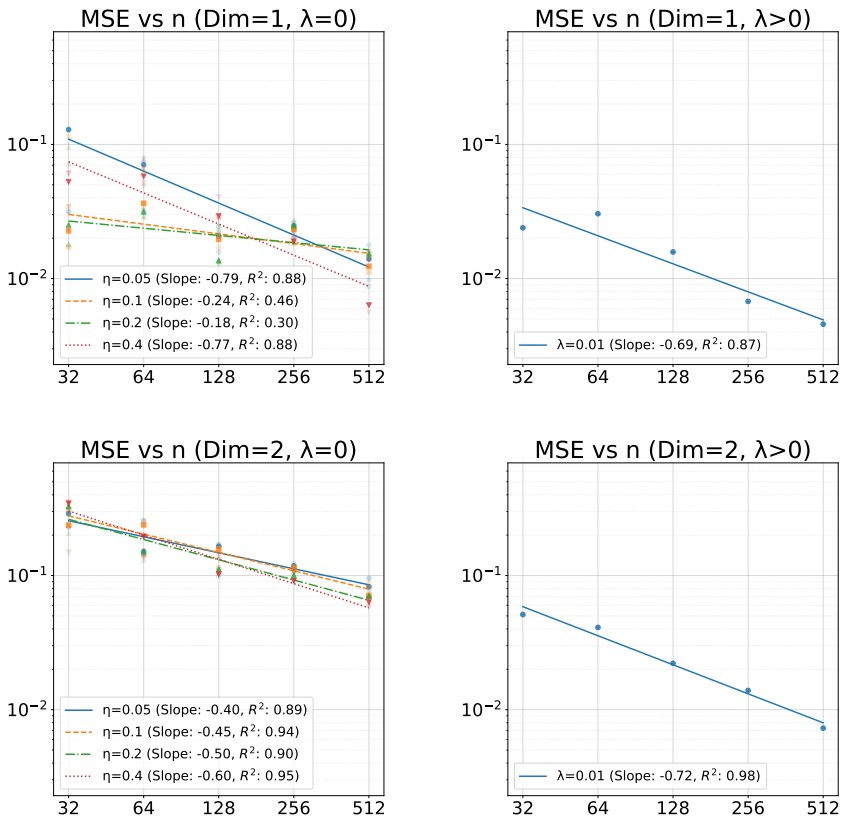

Figure 14: Log–log plots of the mean squared error (MSE) versus sample size $n$ (Part VI). The panels on the left are the log-log plots for stable minima trained in $\eta \in \{0.05, 0.1, 0.2, 0.4\}$, while the panels on the left are the log-log plots for low-norm solutions trained in weight decay $\lambda = 0.01$.

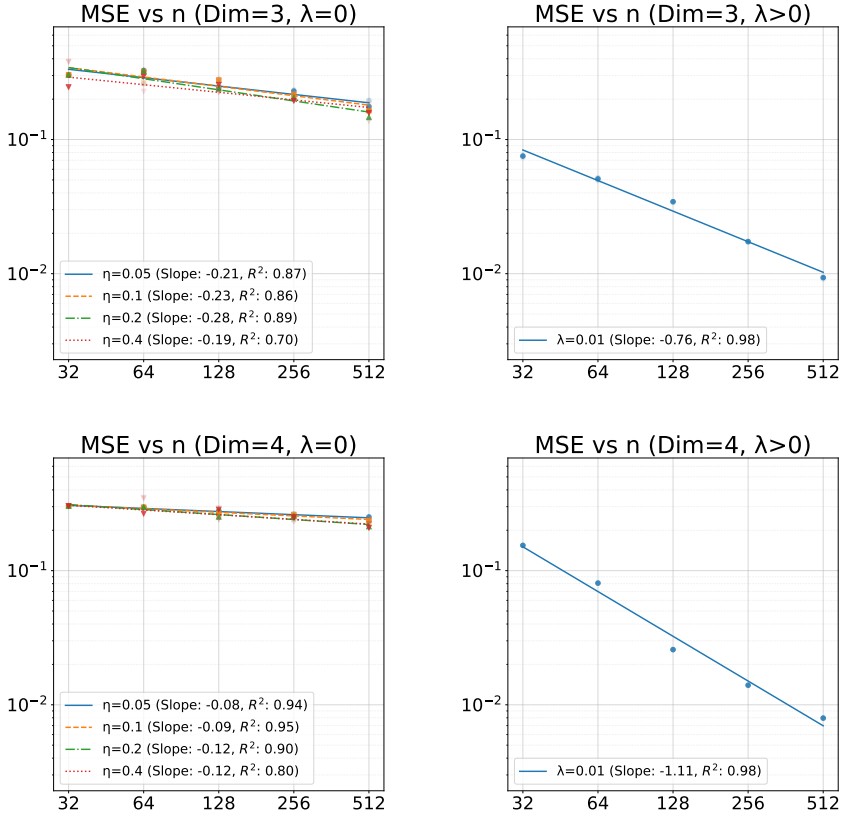

Figure 15: Log–log plots of the mean squared error (MSE) versus sample size $n$ (Part VII). The panels on the left are the log-log plots for stable minima trained in different learning rate $\eta \in \{0.05, 0.1, 0.2, 0.4\}$, while The panels on the left are the log-log plots for low-norm solutions trained in weight decay $\lambda = 0.01$.

# B   Overview of the Proofs

In this section, we provide an overview of the proofs of the claims in the paper. The full proofs are deferred to later appendices. We introduce the following notations we use in our proofs and their overviews.

- Let $\varphi(\varepsilon)$ and $\psi(\varepsilon)$ be two functions in variable of $\varepsilon$. For constants $a, b \in \mathbb{R}$ (independent of $\varepsilon$), the notation

$$\varphi(\varepsilon) \overset{a}{\underset{b}{\asymp}} \psi(\varepsilon)$$

  means that $\varphi(\varepsilon) \leq a\,\psi(\varepsilon)$ and $b\,\psi(\varepsilon) \leq \varphi(\varepsilon)$. We may directly use the notation $\asymp$ if the constants are hidden (we may use the simplified version when the constants are justified).

- $f(x) = O(g(x))$ means there exist constants $c > 0$ and $x_0 > 0$ such that

$$0 \leq f(x) \leq c\,g(x), \quad \forall\, x \geq x_0.$$

  Intuitively, for sufficiently large $x$, $f(x)$ grows at most as fast as $g(x)$, up to a constant factor. We may also use $f(x) \lesssim g(x)$.

- $f(x) = \Omega(g(x))$ means there exist constants $c' > 0$ and $x_1 > 0$ such that

$$0 \leq c'\,g(x) \leq f(x), \quad \forall\, x \geq x_1.$$

  Intuitively, for sufficiently large $x$, $f(x)$ grows at least as fast as $g(x)$, up to a constant factor.

- $f(x) = \Theta(g(x))$ means there exist constants $c_1, c_2 > 0$ and $x_2 > 0$ such that

$$0 \leq c_1\,g(x) \leq f(x) \leq c_2\,g(x), \quad \forall\, x \geq x_2.$$

  Equivalently,

$$f(x) = \Theta(g(x)) \iff [f(x) = O(g(x))] \text{ and } [f(x) = \Omega(g(x))].$$

  Intuitively, for sufficiently large $x$, $f(x)$ grows at the same rate as $g(x)$, up to constant factors.

## B.1   Proof Overview of Theorem 3.2.

We consider the neural network of the form:

$$f_{\boldsymbol{\theta}}(\boldsymbol{x}) = \sum_{k=1}^{K} v_k\, \phi(\boldsymbol{w}_k^{\mathsf{T}} \boldsymbol{x} - b_k) + \beta. \tag{14}$$

The Hessian matrix of the loss function, obtained through direct computation, is expressed as:

$$\nabla_{\boldsymbol{\theta}}^2 \mathcal{L}(\boldsymbol{\theta}) = \frac{1}{n} \sum_{i=1}^{n} (\nabla_{\boldsymbol{\theta}} f_{\boldsymbol{\theta}}(\boldsymbol{x}_i))\, (\nabla_{\boldsymbol{\theta}} f_{\boldsymbol{\theta}}(\boldsymbol{x}_i))^{\mathsf{T}} + \frac{1}{n} \sum_{i=1}^{n} (f_{\boldsymbol{\theta}}(\boldsymbol{x}_i) - y_i)\, \nabla_{\boldsymbol{\theta}}^2 f_{\boldsymbol{\theta}}(\boldsymbol{x}_i). \tag{15}$$

Consider $\boldsymbol{v}$ to be the unit eigenvector (i.e., $\|\boldsymbol{v}\|_2 = 1$) corresponding to the largest eigenvalue of the matrix $\frac{1}{n} \sum_{i=1}^{n} (\nabla_{\boldsymbol{\theta}} f_{\boldsymbol{\theta}}(\boldsymbol{x}_i))(\nabla_{\boldsymbol{\theta}} f_{\boldsymbol{\theta}}(\boldsymbol{x}_i))^{\mathsf{T}}$. Consequently, the maximum eigenvalue of the Hessian of the loss can be lower-bounded as follows:

$$\lambda_{\max}(\nabla_{\boldsymbol{\theta}}^2 \mathcal{L}(\boldsymbol{\theta})) \geq \boldsymbol{v}^{\mathsf{T}} \nabla_{\boldsymbol{\theta}}^2 \mathcal{L}(\boldsymbol{\theta}) \boldsymbol{v}$$

$$= \underbrace{\lambda_{\max}\left(\frac{1}{n} \sum_{i=1}^{n} (\nabla_{\boldsymbol{\theta}} f_{\boldsymbol{\theta}}(\boldsymbol{x}_i))(\nabla_{\boldsymbol{\theta}} f_{\boldsymbol{\theta}}(\boldsymbol{x}_i))^{\mathsf{T}}\right)}_{\text{(Term A)}}$$

$$+ \underbrace{\frac{1}{n} \sum_{i=1}^{n} (f_{\boldsymbol{\theta}}(\boldsymbol{x}_i) - y_i)\boldsymbol{v}^{\mathsf{T}} \nabla_{\boldsymbol{\theta}}^2 f_{\boldsymbol{\theta}}(\boldsymbol{x}_i) \boldsymbol{v}}_{\text{(Term B)}}. \tag{16}$$

Regarding (Term A), its maximum eigenvalue at a given $\boldsymbol{\theta}$ can be related to the $V_g$ seminorm of the associated function $f = f_{\boldsymbol{\theta}}$. Letting $\Omega = \mathbb{B}^d(\boldsymbol{0}, R)$, Nacson et al. [2023, Appendix F.2] demonstrate that:

$$\text{(Term A)} = \lambda_{\max}\left(\frac{1}{n}\sum_{i=1}^{n}(\nabla_{\boldsymbol{\theta}}f_{\boldsymbol{\theta}}(\boldsymbol{x}_i))(\nabla_{\boldsymbol{\theta}}f_{\boldsymbol{\theta}}(\boldsymbol{x}_i))^{\mathsf{T}}\right) \geq 1 + 2\sum_{k=1}^{K}|v_k|\|\boldsymbol{w}_k\|_2\,\tilde{g}(\bar{\boldsymbol{w}}_k, \bar{b}_k), \quad (17)$$

where $\bar{\boldsymbol{w}}_k = \boldsymbol{w}_k/\|\boldsymbol{w}_k\|_2 \in \mathbb{S}^{d-1}$, $\bar{b}_k = b_k/\|\boldsymbol{w}_k\|_2$ and

$$\tilde{g}(\bar{\boldsymbol{w}}, \bar{b}) = \mathbb{P}(\boldsymbol{X}^{\mathsf{T}}\bar{\boldsymbol{w}} > \bar{b})^2 \cdot \mathbb{E}[\boldsymbol{X}^{\mathsf{T}}\bar{\boldsymbol{w}} - \bar{b} \mid \boldsymbol{X}^{\mathsf{T}}\bar{\boldsymbol{w}} > \bar{b}] \cdot \sqrt{1 + \|\mathbb{E}[\boldsymbol{X} \mid \boldsymbol{X}^{\mathsf{T}}\bar{\boldsymbol{w}} > \bar{b}]\|^2}. \quad (18)$$

For (Term B), an upper bound can be established using the training loss $\mathcal{L}(\boldsymbol{\theta})$ via the Cauchy-Schwarz inequality. This also employs a notable uniform upper bound for $\boldsymbol{v}^{\mathsf{T}}\nabla_{\boldsymbol{\theta}}^2 f_{\boldsymbol{\theta}}(\boldsymbol{x}_n)\boldsymbol{v}$, as detailed in Lemma C.1:

$$|\text{(Term B)}| \leq \sqrt{\frac{1}{n}\sum_{i=1}^{n}(f_{\boldsymbol{\theta}}(\boldsymbol{x}_i) - y_i)^2} \cdot \sqrt{\frac{1}{n}\sum_{i=1}^{n}(\boldsymbol{v}^{\mathsf{T}}\nabla_{\boldsymbol{\theta}}^2 f_{\boldsymbol{\theta}}(\boldsymbol{x}_i)\boldsymbol{v})^2} \leq 2(R+1)\sqrt{2\mathcal{L}(\boldsymbol{\theta})}. \quad (19)$$

## B.2 Proof Overview of Theorem 3.5

This proof establishes an upper bound on the generalization gap for stable minima in $\Theta_{\text{flat}}(\eta; \mathcal{D})$. The strategy leverages the structural properties of these solutions, which are captured by a data-dependent weighted variation norm.

First, we recall from Corollary 3.3 that any stable solution $f_{\boldsymbol{\theta}}$ has a bounded norm, $|f_{\boldsymbol{\theta}}|_{V_g} \leq A$. The weight function $g$ is determined by the training data. This data-dependent nature is central to our analysis. To bound the generalization gap, we must translate the constraint on the weighted norm $|f|_{V_g}$ into a bound on the standard, unweighted norm $|f|_V$. This is possible only in regions where the weight function $g$ is bounded away from zero. This naturally suggests a decomposition of the input domain into two parts: a "well-behaved" region where $g$ has a positive lower bound, and the remaining region where $g$ may be arbitrarily close to zero.

To facilitate a tractable analysis, we introduce the deterministic population weight function $g_P$ as a reference. We then bridge the two using empirical process theory. As established in Appendix E.2 (Theorem E.5), the uniform deviation between $g$ and its population counterpart is bounded by a statistical error term $\epsilon_n = \tilde{O}(\sqrt{d/n})$ with high probability. This allows us to leverage the well-behaved properties of $g_P$ to characterize the behavior of the empirical function $g$.

For inputs from $\text{Uniform}(\mathbb{B}_1^d)$, this population function behaves like $(1 - |t|)^{d+2}$ (see Appendix E), where $|t|$ is the distance of a neuron's activation boundary from the origin. This behavior motivates a specific geometric decomposition: an inner core $\mathbb{B}_{1-\varepsilon}^d$ (where $|t| < 1 - \varepsilon$) and an outer annulus $\mathbb{A}_{\varepsilon}^d$.

For the inner core $\mathbb{B}_{1-\varepsilon}^d$, the key step is to translate the bound on $|f|_{V_g}$ to a bound on the standard (unweighted) norm $|f|_V$. To do this, we need a lower bound on the empirical weight $g$ within the core. Using $g_P$ as our analytical proxy, we establish that $g_{\min} \geq g_{P,\min} - \varepsilon_n \approx \varepsilon^{d+2} - \varepsilon_n$. This step requires a validity condition: $\varepsilon$ must be large enough such that the geometric term $\varepsilon^{d+2}$ dominates the statistical error $\varepsilon_n$. With the unweighted norm now bounded, we utilize metric entropy arguments (e.g., Proposition F.4 and results from Parhi and Nowak [2023b]) to bound the generalization error in the core that scales with $O\left(\varepsilon^{-\frac{d(d+2)}{2d+3}} n^{-\frac{d+3}{4d+6}}\right)$. In the annulus $\mathbb{A}_{\varepsilon}^d$, the contribution is small, scaling with $O(\varepsilon)$.

## B.3 Proof Overview of Theorem 3.6

The proof for Theorem 3.6 establishes an upper bound on the mean squared error (MSE) for estimating a true function $f_0$ using a stable minimum $f_{\boldsymbol{\theta}}$. The overall strategy shares similarities with the proof of the generalization gap, particularly in its treatment of the data-dependent function class.

The argument begins by leveraging the property that a stable minimum $\boldsymbol{\theta} \in \Theta_{\text{flat}}(\eta; \mathcal{D})$ corresponds to a neural network $f_{\boldsymbol{\theta}}$ with a bounded weighted variation norm $|f_{\boldsymbol{\theta}}|_{V_g}$, where $g$ is the empirical

weight function. The theorem also assumes the ground truth $f_0$ lies in a similar space. A key condition is that $f_{\boldsymbol{\theta}}$ is "optimized" such that its empirical loss is no worse than that of $f_0$. This is crucial as it allows us to bound the empirical MSE primarily by an empirical process term involving the noise terms $\varepsilon_i$.

To bound this empirical process, the proof again decomposes the input domain $\mathbb{B}_1^d$ into a strict interior ball $\mathbb{B}_{1-\varepsilon}^d$ and an annulus $\mathbb{A}_{\varepsilon}^d$. In the outer shell, the contribution to the MSE is controlled by the function's $L^\infty$ bound. For the strict interior $\mathbb{B}_{1-\varepsilon}^d$, we analyze the difference function $f_\Delta = f_{\boldsymbol{\theta}} - f_0$. Consistent with our generalization analysis, we use the results from Appendix E.2 to ensure the empirical weight function $g$ can be reliably analyzed via its population counterpart. This allows us to bound the unweighted variation norm of $f_\Delta$ over the core, which is then used to bound the empirical process via local Gaussian complexities (as detailed in Appendix G).

The bounds from the annulus and the core are then summed. The resulting expression for the total MSE is minimized by choosing an optimal $\varepsilon$. This balancing yields the final estimation error rate presented in Theorem 3.6, connecting the stability-induced regularity and the "optimized" nature of the solution to its statistical performance.

### B.4 Proof Overview of Theorem 3.7

The proof establishes the minimax lower bound by constructing a packing set of functions within the specified function class $V_g(\mathbb{B}_1^d)$ and then applying Fano's Lemma. The construction differs for multivariate ($d > 1$) and univariate ($d = 1$) cases.

**Multivariate Case** ($d > 1$)  The core idea is to use highly localized ReLU atoms that have a small $V_g$ norm due to the weighting $g(\boldsymbol{u}, t)$ vanishing near the boundary ($|t| \to 1$), yet can be combined to form a sufficiently rich and separated set of functions.

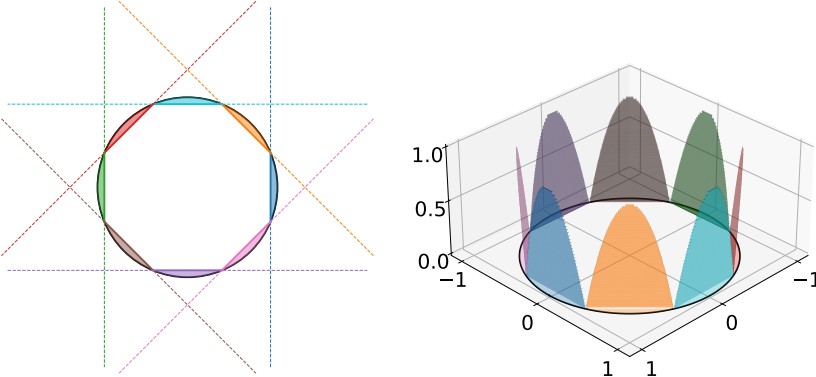

Figure 16: The ReLU atoms only activate on the localized spherical cap and with $L^\infty(\mathbb{B}_1^d)$-norm equal to 1. As dimension increases, more data points will concentrate on the boundary region and the choice of directions increase exponentially.

1. **Atom Construction:** We utilize ReLU atoms $\Phi_{\boldsymbol{u}, \varepsilon^2}(\boldsymbol{x}) = \varepsilon^{-2}\phi(\boldsymbol{u}^\mathsf{T}\boldsymbol{x} - (1 - \varepsilon^2))$ as defined in Construction H.4 (see Eq. (105) for the unnormalized version). These atoms are $L^\infty$-normalized, have an $L^2(\mathbb{B}_1^d)$-norm $\|\Phi_{\boldsymbol{u}, \varepsilon^2}\|_{L^2} \asymp \varepsilon^{\frac{d+1}{2}}$ (Lemma H.1), and a weighted variation norm $|\Phi_{\boldsymbol{u}, \varepsilon^2}|_{V_g} = \varepsilon^{2d+2}$ (Lemma H.2, Eq. (116)). The small $V_g$ norm is crucial.

2. **Packing Set:** Using a packing of $K \asymp \varepsilon^{-(d-1)}$ disjoint spherical caps on $\mathbb{S}^{d-1}$ (Lemma H.3), we construct a family of functions $f_\xi(\boldsymbol{x}) = \sum_{i=1}^{K} \xi_i \Phi_{\boldsymbol{u}_i, \varepsilon^2}(\boldsymbol{x})$ for $\xi \in \{-1, 1\}^K$. By Varshamov-Gilbert lemma (Lemma J.2), we can find a subset $\Xi \subset \{-1, 1\}^K$ such that $\log|\Xi| \asymp K \asymp \varepsilon^{-(d-1)}$ and for any distinct $f_\xi, f_{\xi'} \in \{f_\zeta\}_{\zeta \in \Xi}$, their $L^2$-distance is $\|f_\xi - f_{\xi'}\|_{L^2} \gtrsim \varepsilon$. The total variation norm $|f_\xi|_{V_g} \le K\varepsilon^{2d+2} \asymp \varepsilon^{d+3}$, which is significantly smaller than 1 when $\varepsilon < 1$.

3. **Leveraging Fano's Lemma:** (Proposition H.5) The KL divergence between distributions induced by $f_\xi$ and $f_{\xi'}$ is $\mathrm{KL}(P_\xi \| P_{\xi'}) \asymp n\varepsilon^2/\sigma^2$. To apply Fano's Lemma (see Lemma J.1), we need to satisfy the condition (141) that $n\varepsilon^2/\sigma^2 \lesssim \log|\Xi| \asymp \varepsilon^{-(d-1)}$, which implies $\varepsilon \asymp (\sigma^2/n)^{\frac{1}{d+1}}$ and the minimax risk is then given by $\mathbb{E}\|\hat{f} - f\|_{L^2}^2 \gtrsim \varepsilon^2 \asymp (\sigma^2/n)^{\frac{2}{d+1}}$.

**Univariate Case ($d = 1$)** The high-dimensional spherical cap packing is not applicable. Instead, we use scaled bump functions and exploit the simplified 1D $\mathrm{V}_g$ norm.

1. **Function Class:** For $d = 1$, if we assume $f$ is smooth, then $|f|_{\mathrm{V}_g} = \|f'' \cdot g\|_{\mathcal{M}} = \int_{-1}^1 |f''(x)|(1-|x|)^3 \, \mathrm{d}x$ (from Theorem 3.4 and leading to the class in Eq. (121)).

2. **Atom Construction:** We construct functions $\Phi_k(x)$ as smooth bump functions, each supported on a distinct interval of width $\varepsilon^2$ near the boundary (e.g., $x \in [1-\varepsilon+(k-1)\,\varepsilon^2, 1-\varepsilon+k\varepsilon^2]$). These are scaled such that $\|\Phi_k\|_{L^2} \asymp \varepsilon$. Due to the $(1-|x|)^3 \lesssim \varepsilon^3$ factor in the $\mathrm{V}_g$ norm and $\int_{-1}^1 |\Phi_k''(x)| \, \mathrm{d}x \asymp 1/\varepsilon^2$, the weighted variation is $|\Phi_k|_{\mathrm{V}_g} \asymp \varepsilon^3 \cdot (1/\varepsilon^2) = \varepsilon$.

3. **Packing Set:** A family $f_\xi(x) = \sum_{k=1}^K \xi_k \Phi_k(x)$ is formed with $K \asymp 1/\varepsilon$ terms. Using Varshamov-Gilbert (Lemma J.2), we find a subset $\Xi$ with $\log|\Xi| \asymp K \asymp 1/\varepsilon$ such that for distinct $f_\xi, f_{\xi'}$, the $L^2$-distance is $\|f_\xi - f_{\xi'}\|_{L^2} \gtrsim \sqrt{K}\varepsilon \asymp \sqrt{1/\varepsilon} \cdot \varepsilon = \sqrt{\varepsilon}$.

4. **Leveraging Fano's Lemma:** The KL divergence is $\mathrm{KL}(P_\xi \| P_{\xi'}) \asymp n(\sqrt{\varepsilon})^2/\sigma^2 = n\varepsilon/\sigma^2$. Fano's condition (141) $n\varepsilon/\sigma^2 \lesssim \log|\Xi| \asymp 1/\varepsilon$ implies $\varepsilon \asymp (\sigma^2/n)^{1/2}$. The minimax risk is then $\mathbb{E}\|\hat{f} - f\|_{L^2}^2 \gtrsim (\sqrt{\varepsilon})^2 = \varepsilon \asymp (\sigma^2/n)^{1/2}$.

## B.5 Discussion of the Proofs

A notable feature in the proofs for the generalization gap upper bound (Theorem 3.5) and the MSE upper bound (Theorem 3.6) is the strategy of decomposing the domain $\mathbb{B}_1^d$ into an inner core $\mathbb{B}_{1-\varepsilon}^d$ and an annulus $\mathbb{A}_\varepsilon^d$. This decomposition, involving a trade-off by treating the boundary region differently, is not merely a technical convenience but is fundamentally motivated by the characteristics of the function class $\mathrm{V}_g(\mathbb{B}_1^d)$ and the nature of "hard-to-learn" functions within it.

The necessity for this approach is starkly illustrated by our minimax lower bound construction in Theorem 3.7 (see Appendix H for construction details) and Proposition I.1. The hard-to-learn functions used to establish this lower bound are specifically constructed using ReLU neurons that activate *only* near the boundary of the unit ball (i.e., for $\boldsymbol{x}$ such that $\boldsymbol{u}^\top \boldsymbol{x} \approx 1$). The crucial insight here is the behavior of the weight function $g(\boldsymbol{u}, t) \asymp (1 - |t|)^{d+2}$ (see Appendix E). For these boundary-activating neurons, $|t|$ is close to 1, making $g(\boldsymbol{u}, t)$ exceptionally small. This allows for functions that are potentially complex or have large unweighted magnitudes near the boundary (the annulus) to still possess a small weighted variation norm $|f|_{\mathrm{V}_g}$, thus qualifying them as members of the function class under consideration. Our lower bound construction focuses almost exclusively on these boundary phenomena, as they represent the primary source of difficulty for estimation within this specific weighted variation space.

The upper bound proofs implicitly acknowledge this. By isolating the annulus $\mathbb{A}_\varepsilon^d$, the analysis effectively concedes that this region might harbor complex behavior. The error contribution from this annulus is typically bounded by simpler means, often proportional to its small volume (controlled by $\varepsilon$) and the $L^\infty$ norm of the functions. The more sophisticated analysis, involving metric entropy or Gaussian complexity arguments (which depend on an *unweighted* variation norm that becomes large as $|f|_{\mathrm{V}_g}/\varepsilon^{d+2}$ when restricted to the strict interior $\mathbb{B}_{1-\varepsilon}^d$), is applied to the "better-behaved" interior region. The parameter $\varepsilon$ is then chosen optimally to balance the error from the boundary (which increases with $\varepsilon$) against the error from the interior (where the complexity term effectively increases as $\varepsilon$ shrinks).

This methodological alignment between our upper and lower bounds underscores a self-consistency in our analysis. Both components of the argument effectively exploit the geometric properties stemming from the uniform data distribution on a sphere and the specific decay characteristics of the data-dependent weight function $g$ near the boundary. The strategy of "sacrificing the boundary" in the upper bounds is thus a direct and necessary consequence of where the challenging functions identified by the lower bound constructions.

# C  Proof of Theorem 3.2: Stable Minima Regularity

In this section, we prove the regularity constraint of stable minima. We begin by upper bounding the operator norm of the Hessian matrix. In other words, we upper bound $|v^\top \nabla_\theta^2 f_\theta(x)v|$ under the constraint that $\|v\|_2 = 1$.

**Lemma C.1.** *Assume $f_\theta(x) = \sum_{k=1}^{K} v_k \phi(w_k^\top x + b_k) + \beta$ is a two-layer ReLU network with input $x \in \mathbb{R}^d$ such that $\|x\|_2 \leq R$. Let $\theta$ represent all parameters $\{w_k, b_k, v_k, \beta\}_{k=1}^K$. Assume $f_\theta(x)$ is twice differentiable with respect to $\theta$ at $x$. Then for any vector $v$ corresponding to a perturbation in $\theta$ such that $\|v\|_2 = 1$, it holds that:*

$$|v^\top \nabla_\theta^2 f_\theta(x)v| \leq 2(R+1). \tag{20}$$

*Proof.* Let the parameters be $\theta = (w_1^\top, ..., w_K^\top, b_1, ..., b_K, v_1, ..., v_K, \beta)^\top$. The total number of parameters is $N = K \times d + K + K + 1 = K(d+2) + 1$. Let the corresponding perturbation vector be $v \in \mathbb{R}^N$, structured as: $v = (\alpha_1^\top, ..., \alpha_K^\top, \delta_1, ..., \delta_K, \gamma_1, ..., \gamma_K, \iota)^\top$, where $\alpha_k \in \mathbb{R}^d$ corresponds to $w_k$, $\delta_k \in \mathbb{R}$ corresponds to $b_k$, $\gamma_k \in \mathbb{R}$ corresponds to $v_k$, and $\iota \in \mathbb{R}$ corresponds to $\beta$. The normalization constraint is

$$\|v\|_2^2 = \sum_{k=1}^K \|\alpha_k\|_2^2 + \sum_{k=1}^K \delta_k^2 + \sum_{k=1}^K \gamma_k^2 + \iota^2 = 1 \tag{21}$$

We need to compute the Hessian matrix $\nabla_\theta^2 f_\theta(x)$. Let $z_k = w_k^\top x + b_k$ and $1_k = 1(z_k > 0)$. Since we assume twice differentiability, $z_k \neq 0$ for all $k$, the Hessian $\nabla_\theta^2 f_\theta(x)$ is block diagonal, with $K$ blocks corresponding to each neuron. The $k$-th block, $\nabla_{(\theta_k)}^2 f_\theta(x)$, involves derivatives with respect to $\theta_k = (w_k^\top, b_k, v_k)^\top$. The relevant non-zero second partial derivatives defining this block are:

- $\frac{\partial^2 f_\theta}{\partial w_k \partial v_k} = \frac{\partial}{\partial v_k}(\nabla_{w_k} f_\theta) = \frac{\partial}{\partial v_k}(v_k \phi'(z_k)x) = \phi'(z_k)x = 1_k x$

- $\frac{\partial^2 f_\theta}{\partial b_k \partial v_k} = \frac{\partial}{\partial v_k}\left(\frac{\partial f_\theta}{\partial b_k}\right) = \frac{\partial}{\partial v_k}(v_k \phi'(z_k)) = \phi'(z_k) = 1_k$

All other second derivatives within the block are zero, as are derivatives between different blocks or involving $\beta$. The $k$-th block of the Hessian is thus:

$$\nabla_{(\theta_k)}^2 f_\theta(x) = \begin{pmatrix} \frac{\partial^2 f_\theta}{(\partial w_k)^2} & \frac{\partial^2 f_\theta}{\partial w_k \partial b_k} & \frac{\partial^2 f_\theta}{\partial w_k \partial v_k} \\ \frac{\partial^2 f_\theta}{\partial b_k \partial w_k} & \frac{\partial^2 f_\theta}{(\partial b_k)^2} & \frac{\partial^2 f_\theta}{\partial b_k \partial v_k} \\ \frac{\partial^2 f_\theta}{\partial v_k \partial w_k} & \frac{\partial^2 f_\theta}{\partial v_k \partial b_k} & \frac{\partial^2 f_\theta}{(\partial v_k)^2} \end{pmatrix} = \begin{pmatrix} \mathbf{0}_{d \times d} & \mathbf{0}_d & 1_k x \\ \mathbf{0}_d^\top & 0 & 1_k \\ 1_k x^\top & 1_k & 0 \end{pmatrix} \tag{22}$$

where $\mathbf{0}_{d \times d}$ is the $d \times d$ zero matrix and $\mathbf{0}_d$ is the $d$-dimensional zero vector.

The quadratic form $v^\top \nabla_\theta^2 f_\theta(x)v$ becomes:

$$\begin{aligned} v^\top \nabla_\theta^2 f_\theta(x)v &= \sum_{k=1}^K \begin{pmatrix} \alpha_k^\top & \delta_k & \gamma_k \end{pmatrix} \begin{pmatrix} \mathbf{0}_{d \times d} & \mathbf{0}_d & 1_k x \\ \mathbf{0}_d^\top & 0 & 1_k \\ 1_k x^\top & 1_k & 0 \end{pmatrix} \begin{pmatrix} \alpha_k \\ \delta_k \\ \gamma_k \end{pmatrix} \\ &= \sum_{k=1}^K \left(\alpha_k^\top(1_k x)\gamma_k + \delta_k(1_k)\gamma_k + \gamma_k(1_k x^\top)\alpha_k + \gamma_k(1_k)\delta_k\right) \\ &= \sum_{k=1}^K 2 \cdot 1_k \cdot \gamma_k(\alpha_k^\top x + \delta_k) \end{aligned} \tag{23}$$

Now, we bound the absolute value:

$$
\begin{aligned}
|\boldsymbol{v}^{\mathsf{T}}\nabla_{\boldsymbol{\theta}}^2 f_{\boldsymbol{\theta}}(\boldsymbol{x})\boldsymbol{v}| = \left|\sum_{k=1}^{K} 2 \cdot 1_k \cdot \gamma_k(\boldsymbol{\alpha}_k^{\mathsf{T}}\boldsymbol{x} + \delta_k)\right| &\leq \sum_{k=1}^{K} 2 \cdot 1_k \cdot |\gamma_k||\boldsymbol{\alpha}_k^{\mathsf{T}}\boldsymbol{x} + \delta_k| \\
&\leq \sum_{k=1}^{K} 2|\gamma_k|(|\boldsymbol{\alpha}_k^{\mathsf{T}}\boldsymbol{x}| + |\delta_k|) \quad \leq \sum_{k=1}^{K} 2|\gamma_k|(\|\boldsymbol{\alpha}_k\|_2\|\boldsymbol{x}\|_2 + |\delta_k|) \quad \text{(Cauchy-Schwarz)} \\
&= 2R\sum_{k=1}^{K}|\gamma_k|\|\boldsymbol{\alpha}_k\|_2 + 2\sum_{k=1}^{K}|\gamma_k||\delta_k| \\
&\leq 2R\sqrt{\sum_{k=1}^{K}\gamma_k^2}\sqrt{\sum_{k=1}^{K}\|\boldsymbol{\alpha}_k\|_2^2} + 2\sqrt{\sum_{k=1}^{K}\gamma_k^2}\sqrt{\sum_{k=1}^{K}\delta_k^2} \quad \text{(Cauchy-Schwarz on sums)} \\
&\leq 2R\sqrt{\sum\gamma_k^2}\cdot\sqrt{1} + 2\sqrt{\sum\gamma_k^2}\cdot\sqrt{1} \\
&= 2(R+1)\sqrt{\sum\gamma_k^2} \quad \leq 2(R+1).
\end{aligned}
$$

$\square$

Now we are ready to prove Theorem 3.2.

*Proof of Theorem 3.2.* Without loss of generality, we consider neural networks of the following form:

$$
f_{\boldsymbol{\theta}}(\boldsymbol{x}) = \sum_{k=1}^{K} v_k\,\phi(\boldsymbol{w}_k^{\mathsf{T}}\boldsymbol{x} - b_k) + \beta. \tag{24}
$$

The Hessian matrix of the loss function, obtained through direct computation, is expressed as:

$$
\nabla_{\boldsymbol{\theta}}^2 \mathcal{L}(\boldsymbol{\theta}) = \frac{1}{n}\sum_{i=1}^{n}(\nabla_{\boldsymbol{\theta}}f_{\boldsymbol{\theta}}(\boldsymbol{x}_i))(\nabla_{\boldsymbol{\theta}}f_{\boldsymbol{\theta}}(\boldsymbol{x}_i))^{\mathsf{T}} + \frac{1}{n}\sum_{i=1}^{n}(f_{\boldsymbol{\theta}}(\boldsymbol{x}_i) - y_i)\,\nabla_{\boldsymbol{\theta}}^2 f_{\boldsymbol{\theta}}(\boldsymbol{x}_i). \tag{25}
$$

Let $\boldsymbol{v}$ be the unit eigenvector (i.e., $\|\boldsymbol{v}\|_2 = 1$) corresponding to the largest eigenvalue of the matrix $\frac{1}{n}\sum_{i=1}^{n}(\nabla_{\boldsymbol{\theta}}f_{\boldsymbol{\theta}}(\boldsymbol{x}_i))(\nabla_{\boldsymbol{\theta}}f_{\boldsymbol{\theta}}(\boldsymbol{x}_i))^{\mathsf{T}}$, the maximum eigenvalue of the Hessian matrix of the loss can be lower-bounded as follows:

$$
\begin{aligned}
\lambda_{\max}(\nabla_{\boldsymbol{\theta}}^2 \mathcal{L}(\boldsymbol{\theta})) \geq &\boldsymbol{v}^{\mathsf{T}}\nabla_{\boldsymbol{\theta}}^2 \mathcal{L}(\boldsymbol{\theta})\boldsymbol{v} \\
= &\underbrace{\lambda_{\max}\left(\frac{1}{n}\sum_{i=1}^{n}(\nabla_{\boldsymbol{\theta}}f_{\boldsymbol{\theta}}(\boldsymbol{x}_i))(\nabla_{\boldsymbol{\theta}}f_{\boldsymbol{\theta}}(\boldsymbol{x}_i))^{\mathsf{T}}\right)}_{\text{(Term A)}} \\
&+ \underbrace{\frac{1}{n}\sum_{i=1}^{n}(f_{\boldsymbol{\theta}}(\boldsymbol{x}_i) - y_i)\boldsymbol{v}^{\mathsf{T}}\nabla_{\boldsymbol{\theta}}^2 f_{\boldsymbol{\theta}}(\boldsymbol{x}_i)\boldsymbol{v}}_{\text{(Term B)}}.
\end{aligned} \tag{26}
$$

Regarding (Term A), its maximum eigenvalue at a given $\boldsymbol{\theta}$ can be related to the $\mathrm{V}_g$ norm of the associated function $f = f_{\boldsymbol{\theta}}$. Considering the domain $\mathbb{B}_R^d$, Nacson et al. [2023, Appendix F.2] demonstrate that:

$$
\text{(Term A)} = \lambda_{\max}\left(\frac{1}{n}\sum_{i=1}^{n}(\nabla_{\boldsymbol{\theta}}f_{\boldsymbol{\theta}}(\boldsymbol{x}_i))(\nabla_{\boldsymbol{\theta}}f_{\boldsymbol{\theta}}(\boldsymbol{x}_i))^{\mathsf{T}}\right) \geq 1 + 2\sum_{k=1}^{K}|v_k|\|\boldsymbol{w}_k\|_2\,\tilde{g}(\bar{\boldsymbol{w}}_k, \bar{b}_k), \tag{27}
$$

where $\bar{\boldsymbol{w}}_k = \boldsymbol{w}_k/\|\boldsymbol{w}_k\|_2 \in \mathbb{S}^{d-1}$, $\bar{b}_k = b_k/\|\boldsymbol{w}_k\|_2$ and

$$
\tilde{g}(\bar{\boldsymbol{w}}, \bar{b}) = \mathbb{P}(\boldsymbol{X}^{\mathsf{T}}\bar{\boldsymbol{w}} > \bar{b})^2 \cdot \mathbb{E}[\boldsymbol{X}^{\mathsf{T}}\bar{\boldsymbol{w}} - \bar{b} \mid \boldsymbol{X}^{\mathsf{T}}\bar{\boldsymbol{w}} > \bar{b}] \cdot \sqrt{1 + \|\mathbb{E}[\boldsymbol{X} \mid \boldsymbol{X}^{\mathsf{T}}\bar{\boldsymbol{w}} > \bar{b}]\|^2}. \tag{28}
$$

For (Term B), an upper bound can be established using the training loss $\mathcal{L}(\boldsymbol{\theta})$ via the Cauchy-Schwarz inequality. This also employs a notable uniform upper bound for $|\boldsymbol{v}^\mathsf{T}\nabla_{\boldsymbol{\theta}}^2 f_{\boldsymbol{\theta}}(\boldsymbol{x}_n)\boldsymbol{v}|$, as detailed in Lemma C.1:

$$|(\text{Term B})| \leq \sqrt{\frac{1}{n}\sum_{i=1}^n (f_{\boldsymbol{\theta}}(\boldsymbol{x}_i) - y_i)^2} \cdot \sqrt{\frac{1}{n}\sum_{i=1}^n (\boldsymbol{v}^\mathsf{T}\nabla_{\boldsymbol{\theta}}^2 f_{\boldsymbol{\theta}}(\boldsymbol{x}_i)\boldsymbol{v})^2} \leq 2(R+1)\sqrt{2\mathcal{L}(\boldsymbol{\theta})}. \quad (29)$$

Finally, the proof of Theorem 3.2 is complete by plugging (27) and (29) into (26). $\qquad\square$

# D Proof of Theorem 3.4: Radon-Domain Characterization of Stable Minima

In this part, we prove Theorem 3.4 by extending the unweighted case to the weighted one.

*Proof of Theorem 3.4.* In the unweighted scenario, i.e., $g \equiv 1$, it was established by Parhi and Nowak [2023b, Lemma 2] that if $f \in \mathrm{V}(\mathbb{B}_R^d) := \mathrm{V}_1(\mathbb{B}_R^d)$ with integral representation

$$f(\boldsymbol{x}) = \int_{\mathbb{S}^{d-1}\times[-R,R]} \phi(\boldsymbol{u}^\mathsf{T}\boldsymbol{x} - t)\,\mathrm{d}\nu(\boldsymbol{u},t) + \boldsymbol{c}^\mathsf{T}\boldsymbol{x} + c_0, \quad \boldsymbol{x} \in \mathbb{B}_R^d, \quad (30)$$

where $\nu$, $\boldsymbol{c}$, and $c_0$ solve (8) (with $g \equiv 1$) that

$$\nu = \mathscr{R}(-\Delta)^{\frac{d+1}{2}}f_{\text{ext}}, \quad (31)$$

where we recall that $f_{\text{ext}}$ is the canonical extension of $f$ from $\mathbb{B}_R^d$ to $\mathbb{R}^d$ via the formula (30) and $\nu \in \mathcal{M}(\mathbb{S}^{d-1}\times\mathbb{R})$ with $\operatorname{supp}\nu = \mathbb{S}^{d-1}\times[-R,R]$ (i.e., we can identify $\nu$ with a measure in $\mathcal{M}(\mathbb{S}^{d-1}\times[-R,R])$). Since the weighted variation seminorm $|\cdot|_{\mathrm{V}_g}$ is simply (cf., (8))

$$|f|_{\mathrm{V}_g} = \inf_{\substack{\nu\in\mathcal{M}(\mathbb{S}^{d-1}\times[-R,R])\\ \boldsymbol{c}\in\mathbb{R}^d, c_0\in\mathbb{R}}} \|g\cdot\nu\|_{\mathcal{M}} \quad \text{s.t.} \quad f = f_{\nu,\boldsymbol{c},c_0}, \quad (32)$$

we readily see that $|f|_{\mathrm{V}_g} = \|g\cdot\mathscr{R}(-\Delta)^{\frac{d+1}{2}}f_{\text{ext}}\|_{\mathcal{M}}$. $\qquad\square$

**Remark D.1.** The unweighted variation seminorm exactly corresponds to the second-order Radon-domain total variation of the function [Ongie et al., 2020, Parhi and Nowak, 2021, 2022, 2023b,a]. Thus, the weighted variation seminorm is a weighted variant of the second-order Radon-domain total variation.

# E Characterization of the Weight Function for the Uniform Distribution

Recall that, given a data set $\mathcal{D} = \{(\boldsymbol{x}_i, y_i)\}_{i=1}^n \subset \mathbb{R}^d \times \mathbb{R}$, we consider a weight function $g : \mathbb{S}^{d-1}\times\mathbb{R} \to \mathbb{R}$, where $\mathbb{S}^{d-1} := \{\boldsymbol{u} \in \mathbb{R}^d : \|\boldsymbol{u}\| = 1\}$ denotes the unit sphere. This weight is defined by $g(\boldsymbol{u},t) := \min\{\tilde{g}(\boldsymbol{u},t), \tilde{g}(-\boldsymbol{u},-t)\}$, where

$$\tilde{g}(\boldsymbol{u},t) := \mathbb{P}(\boldsymbol{X}^\mathsf{T}\boldsymbol{u} > t)^2 \cdot \mathbb{E}[\boldsymbol{X}^\mathsf{T}\boldsymbol{u} - t \mid \boldsymbol{X}^\mathsf{T}\boldsymbol{u} > t] \cdot \sqrt{1 + \|\mathbb{E}[\boldsymbol{X} \mid \boldsymbol{X}^\mathsf{T}\boldsymbol{u} > t]\|^2}. \quad (33)$$

Here, $\boldsymbol{X}$ is a random vector drawn uniformly at random from the training examples $\{\boldsymbol{x}_i\}_{i=1}^n$. Note that the distribution $\mathbb{P}_X$ for which the $\{\boldsymbol{x}_i\}_{i=1}^n$ are drawn i.i.d. from controls the regularity of $g$.

In this section, We first analyze the properties of the population version $g_P$ by assuming that the random vector $\boldsymbol{X}$ is uniformly sampled from the $d$-dimensional unit ball $\mathbb{B}_1^d = \{\boldsymbol{x} \in \mathbb{R}^d : \|\boldsymbol{x}\|_2 \leq 1\}$. Then we analyze the gap between the empirical $g$ and the population $g_P$ using the empricial process.

## E.1 The Computation of the Population Weight Function

We focus on the marginal distribution of a single coordinate and related conditional expectations. Let $X_1$ be the first coordinate of $\boldsymbol{X}$. Due to symmetry, all coordinates have the same marginal distribution.

The following proposition calculates the marginal probability density function of the first coordinate (and also other coordinates) of the random vector $X$.

**Proposition E.1** (Marginal PDF of a Coordinate). *Let $X$ follow the uniform distribution in $\mathbb{B}_1^d$. The probability density function (PDF) of its first coordinate $X_1$ is given by:*

$$f_{X_1}(t) = c_1(d)\,(1-t^2)^\alpha, \quad t \in [-1, 1] \tag{34}$$

*where $\alpha = \frac{d-1}{2}$ and the normalization constant is*

$$c_1(d) = \frac{\Gamma\left(\frac{d}{2}+1\right)}{\sqrt{\pi}\,\Gamma\left(\frac{d+1}{2}\right)}. \tag{35}$$

*Proof.* The volume of the unit ball is $V_d = \frac{\pi^{d/2}}{\Gamma(d/2+1)}$. The uniform density is $f_X(x) = 1/V_d$ for $x \in \mathbb{B}_1^d$. The marginal PDF is found by integrating out the other coordinates:

$$f_{X_1}(t) = \int_{\{x' \in \mathbb{R}^{d-1} : \|x'\|^2 \le 1 - t^2\}} \frac{1}{V_d}\,dx' = \frac{\mathrm{Vol}_{d-1}(\sqrt{1-t^2})}{V_d}$$

where $\mathrm{Vol}_{d-1}(R)$ is the volume of a $(d-1)$-ball of radius $R$. Using $V_{d-1} = \frac{\pi^{(d-1)/2}}{\Gamma((d-1)/2+1)} = \frac{\pi^{(d-1)/2}}{\Gamma((d+1)/2)}$, we get

$$f_{X_1}(t) = \frac{V_{d-1}(1-t^2)^{(d-1)/2}}{V_d} = \frac{\pi^{(d-1)/2}}{\Gamma((d+1)/2)}\frac{\Gamma(d/2+1)}{\pi^{d/2}}(1-t^2)^{(d-1)/2}$$

which simplifies to the stated result. For $d = 2$, $\alpha = 1/2$, $c_1(2) = \frac{\Gamma(2)}{\sqrt{\pi}\Gamma(3/2)} = \frac{1}{\sqrt{\pi}(\sqrt{\pi}/2)} = \frac{2}{\pi}$. For $d = 3$, $\alpha = 1$, $c_1(3) = \frac{\Gamma(5/2)}{\sqrt{\pi}\Gamma(2)} = \frac{3\sqrt{\pi}/4}{\sqrt{\pi}} = \frac{3}{4}$. $\square$

Given the marginal probability density function, the tail probability follows from direct calculation.

**Proposition E.2** (Tail Probability). *Let $X$ be a random vector uniformly distributed in the $d$-dimensional unit ball $\mathbb{B}_1^d = \{x \in \mathbb{R}^d : \|x\|_2 \le 1\}$. Let $X_1$ be its first coordinate whose tail probability is defined as $Q(x) = \mathbb{P}(X_1 > x)$ for $x \in [-1, 1]$. Then there exists a fixed $x_0 \in [0, 1)$ (specifically, we choose $x_0 = 3/4$, which implies $(1 - x) \in (0, 1/4]$) such that for all $x \in [x_0, 1)$:*

$$Q(x) \underset{c_2(d)}{\overset{c_3(d)}{\asymp}} (1-x)^{\frac{d+1}{2}}.$$

*Or equivalently,*

$$c_2(d)(1-x)^{\frac{d+1}{2}} \le Q(x) \le c_3(d)(1-x)^{\frac{d+1}{2}},$$

*where the constants $c_2(d)$ and $c_3(d)$ are given by:*

$$c_2(d) = \frac{c_1(d)}{d+1}\left(\frac{7}{4}\right)^{\frac{d+1}{2}}$$

$$c_3(d) = \frac{c_1(d)}{d+1}2^{\frac{d+2}{2}}$$

*and $c_1(d) = \frac{\Gamma\left(\frac{d}{2}+1\right)}{\sqrt{\pi}\,\Gamma\left(\frac{d+1}{2}\right)}$ is the normalization constant from the marginal PDF of $X_1$.*

*Proof.* The tail probability $Q(x)$ is given by the integral of the marginal PDF $f_{X_1}(t) = c_1(d)(1-t^2)^\alpha$ for $t \in [-1, 1]$, where $\alpha = \frac{d-1}{2}$.

$$Q(x) = \int_x^1 c_1(d)(1-t^2)^\alpha\,dt$$

Let $s = t^2$, so $dt = ds/(2\sqrt{s})$. The limits of integration for $s$ become $x^2$ to 1.

$$Q(x) = c_1(d)\int_{x^2}^1 (1-s)^\alpha \frac{ds}{2\sqrt{s}}$$

Now, let $u = 1 - s$. Then $s = 1 - u$ and $ds = -du$. Let $\delta_s = 1 - x^2$. The limits for $u$ become $\delta_s$ to 0.

$$Q(x) = \frac{c_1(d)}{2} \int_0^{\delta_s} (1 - u)^{-1/2} u^\alpha \, du$$

For $u \in [0, \delta_s]$, we have $1 \leq (1 - u)^{-1/2} \leq (1 - \delta_s)^{-1/2}$, because $(1 - u)$ is decreasing and non-negative. The integral $\int_0^{\delta_s} u^\alpha \, du = \frac{\delta_s^{\alpha+1}}{\alpha+1}$. Substituting these bounds for the term $(1 - u)^{-1/2}$:

$$\frac{c_1(d)}{2(\alpha + 1)} \delta_s^{\alpha+1} \leq Q(x) \leq \frac{c_1(d)}{2(\alpha + 1)} (1 - \delta_s)^{-1/2} \delta_s^{\alpha+1}$$

We use $\alpha = \frac{d-1}{2}$, so $2(\alpha + 1) = 2(\frac{d-1}{2} + 1) = 2(\frac{d+1}{2}) = d + 1$. The exponent $\alpha + 1 = \frac{d+1}{2}$. Thus,

$$\frac{c_1(d)}{d + 1} \delta_s^{\frac{d+1}{2}} \leq Q(x) \leq \frac{c_1(d)}{d + 1} (1 - \delta_s)^{-1/2} \delta_s^{\frac{d+1}{2}}$$

We choose $x_0 = 3/4$, so we consider $x \in [3/4, 1)$, which means $(1 - x) \in (0, 1/4]$. For $x \in [3/4, 1)$, we have $1 + x \in [7/4, 2)$. The term $\delta_s = 1 - x^2 = (1 - x)(1 + x)$. Given the range for $1 + x$, for $(1 - x) \in (0, 1/4]$:

$$\frac{7}{4}(1 - x) \leq \delta_s < 2(1 - x)$$

- Lower Bound for $Q(x)$: Using $\delta_s \geq \frac{7}{4}(1 - x)$ from the above range:

$$Q(x) \geq \frac{c_1(d)}{d + 1} \left(\frac{7}{4}(1 - x)\right)^{\frac{d+1}{2}} = \frac{c_1(d)}{d + 1} \left(\frac{7}{4}\right)^{\frac{d+1}{2}} (1 - x)^{\frac{d+1}{2}}$$

  This establishes the lower bound with $c_2(d) = \frac{c_1(d)}{d+1} \left(\frac{7}{4}\right)^{\frac{d+1}{2}}$.

- Upper Bound for $Q(x)$: For the term $\delta_s^{\frac{d+1}{2}}$, we use $\delta_s < 2(1-x)$, so $\delta_s^{\frac{d+1}{2}} < (2(1-x))^{\frac{d+1}{2}}$. For the term $(1 - \delta_s)^{-1/2}$: Since $(1 - x) \in (0, 1/4]$, $\delta_s < 2(1 - x) \leq 2(1/4) = 1/2$. So, $1 - \delta_s > 1 - 1/2 = 1/2$. This implies $(1 - \delta_s)^{-1/2} < (1/2)^{-1/2} = \sqrt{2}$. Combining these for the upper bound of $Q(x)$:

$$Q(x) \leq \frac{c_1(d)}{d + 1} 2^{1/2} \cdot 2^{\frac{d+1}{2}} (1 - x)^{\frac{d+1}{2}} = \frac{c_1(d)}{d + 1} 2^{\frac{d+2}{2}} (1 - x)^{\frac{d+1}{2}}$$

  This establishes the upper bound with $c_3(d) = \frac{c_1(d)}{d+1} 2^{\frac{d+2}{2}}$.

Thus, for $x \in [3/4, 1)$ (i.e., $1 - x \in (0, 1/4]$):

$$c_2(d)(1 - x)^{\frac{d+1}{2}} \leq Q(x) \leq c_3(d)(1 - x)^{\frac{d+1}{2}}$$

This corresponds to $Q(x) \asymp_{c_2(d)}^{c_3(d)} (1 - x)^{\frac{d+1}{2}}$. The constants $c_2(d)$ and $c_3(d)$ depend only on the dimension $d$ (via $c_1(d)$ and the exponents derived from $d$) and are valid for the specified range of $x$. $\qquad \square$

Based on the tail probability, we calculate the expectation conditional on the tail events.

**Proposition E.3** (Conditional Expectation). *For $x \in [3/4, 1)$, the conditional expectation $\mathbb{E}[X_1 \mid X_1 > x]$ is bounded by*

$$1 - c_5(d)(1 - x) \leq \mathbb{E}[X_1 \mid X_1 > x] \leq 1 - c_4(d)(1 - x), \tag{36}$$

*where the constants $c_4(d)$ and $c_5(d)$ are given by:*

$$c_4(d) = \frac{2(d + 1)}{d + 3} \frac{(7/4)^{(d-1)/2}}{2^{(d+2)/2}},$$

$$c_5(d) = \frac{2(d + 1)}{d + 3} \frac{2^{(d-1)/2}}{(7/4)^{(d+1)/2}}.$$

*Proof.* We consider $1 - \mathbb{E}[X_1 \mid X_1 > x] = \mathbb{E}[1 - X_1 \mid X_1 > x]$.

$$\mathbb{E}[1 - X_1 \mid X_1 > x] = \frac{1}{Q(x)} \int_x^1 (1 - t) f_{X_1}(t) \, dt$$

$$= \frac{c_1(d)}{Q(x)} \int_x^1 (1 - t)(1 - t^2)^\alpha dt$$

$$= \frac{c_1(d)}{Q(x)} \int_x^1 (1 - t)^{\alpha+1}(1 + t)^\alpha \, dt$$

Let $I_1(x) = \int_x^1 (1 - t)^{\alpha+1}(1 + t)^\alpha dt$. We consider $x \in [3/4, 1)$. For $t \in [x, 1]$, we have $1 + t \in [1 + x, 2]$. Since $x \geq 3/4$, $1 + x \geq 7/4$. Thus, $(7/4)^\alpha \leq (1 + t)^\alpha \leq 2^\alpha$ for $t \in [x, 1]$ (assuming $\alpha \geq 0$, which holds for $d \geq 1$). The integral $\int_x^1 (1 - t)^{\alpha+1} \, dt = \left[-\frac{(1-t)^{\alpha+2}}{\alpha+2}\right]_x^1 = \frac{(1-x)^{\alpha+2}}{\alpha+2}$. So, $I_1(x)$ is bounded by:

$$(7/4)^\alpha \frac{(1 - x)^{\alpha+2}}{\alpha + 2} \leq I_1(x) \leq 2^\alpha \frac{(1 - x)^{\alpha+2}}{\alpha + 2}$$

Let $N_1(x) = c_1(d) I_1(x)$. Then, using $\alpha + 2 = (d + 3)/2$:

$$\frac{2 c_1(d)(7/4)^{(d-1)/2}}{d + 3}(1 - x)^{\frac{d+3}{2}} \leq N_1(x) \leq \frac{2 c_1(d) 2^{(d-1)/2}}{d + 3}(1 - x)^{\frac{d+3}{2}}$$

From Proposition E.2 , for $x \in [3/4, 1)$, $Q(x)$ is bounded by:

$$c_2(d)(1 - x)^{\frac{d+1}{2}} \leq Q(x) \leq c_3(d)(1 - x)^{\frac{d+1}{2}}$$

where $c_2(d) = \frac{c_1(d)}{d+1}\left(\frac{7}{4}\right)^{\frac{d+1}{2}}$ and $c_3(d) = \frac{c_1(d)}{d+1} 2^{\frac{d+2}{2}}$. Therefore, $\mathbb{E}[1 - X_1 \mid X_1 > x] = \frac{N_1(x)}{Q(x)}$ is bounded by:

- Lower bound:

$$\frac{\frac{2 c_1(d)(7/4)^{(d-1)/2}}{d+3}(1 - x)^{\frac{d+3}{2}}}{c_3(d)(1 - x)^{\frac{d+1}{2}}} = \frac{2 c_1(d)(7/4)^{(d-1)/2}/(d + 3)}{\frac{c_1(d)}{d+1} 2^{\frac{d+2}{2}}}(1 - x)$$

$$= \frac{2(d + 1)}{d + 3} \frac{(7/4)^{(d-1)/2}}{2^{(d+2)/2}}(1 - x) = c_4(d)(1 - x)$$

- Upper bound:

$$\frac{\frac{2 c_1(d) 2^{(d-1)/2}}{d+3}(1 - x)^{\frac{d+3}{2}}}{c_2(d)(1 - x)^{\frac{d+1}{2}}} = \frac{2 c_1(d) 2^{(d-1)/2}/(d + 3)}{\frac{c_1(d)}{d+1}\left(\frac{7}{4}\right)^{\frac{d+1}{2}}}(1 - x)$$

$$= \frac{2(d + 1)}{d + 3} \frac{2^{(d-1)/2}}{(7/4)^{(d+1)/2}}(1 - x) = c_5(d)(1 - x)$$

So, for $x \in [3/4, 1)$:

$$c_4(d)(1 - x) \leq \mathbb{E}[1 - X_1 \mid X_1 > x] \leq c_5(d)(1 - x)$$

This implies:

$$1 - c_5(d)(1 - x) \leq \mathbb{E}[X_1 \mid X_1 > x] \leq 1 - c_4(d)(1 - x)$$

This completes the proof. $\qquad\square$

Finally, we combine the results and characterize the asymptotic behavior of the weight function $g$.

**Proposition E.4** (Asymptotic Behavior of $g^+(x)$). *Let the function $g^+(x)$ be defined as: for $x \in (-1, 1)$,*

$$g^+(x) = \mathbb{P}(X_1 > x)^2 \cdot \mathbb{E}[X_1 - x \mid X_1 > x] \cdot \sqrt{1 + (\mathbb{E}[X_1 \mid X_1 > x])^2}. \qquad (37)$$

*Then for $x \in [3/4, 1)$, we have:*

$$c_L^{(g)}(d)(1 - x)^{d+2} \leq g^+(x) \leq c_U^{(g)}(d)(1 - x)^{d+2}, \qquad (38)$$

*where $c_L^{(g)}(d)$ and $c_U^{(g)}(d)$ are positive constants depending on dimension d, defined in the proof* (39).

*Proof.* Let $Q(x) = \mathbb{P}(X_1 > x)$ and $E(x) = \mathbb{E}[X_1 \mid X_1 > x]$. The function is $g^+(x) = Q(x)^2 \cdot (E(x) - x) \cdot \sqrt{1 + E(x)^2}$. Now, we establish precise bounds for $x \in [3/4, 1)$. Let $(1 - x)$ be the variable.

1. Bounds for $Q(x)^2$: From Propostion E.2, $c_2(d)(1-x)^{\frac{d+1}{2}} \le Q(x) \le c_3(d)(1-x)^{\frac{d+1}{2}}$. So, $A_L(d)(1-x)^{d+1} \le Q(x)^2 \le A_U(d)(1-x)^{d+1}$, where

$$A_L(d) = (c_2(d))^2 = \left( \frac{c_1(d)}{d+1} \left( \frac{7}{4} \right)^{\frac{d+1}{2}} \right)^2,$$

$$A_U(d) = (c_3(d))^2 = \left( \frac{c_1(d)}{d+1} 2^{\frac{d+2}{2}} \right)^2.$$

2. Bounds for $E(x) - x = \mathbb{E}[X_1 - x \mid X_1 > x]$: From Propostion E.3, we have $(1-x) - c_5(d)(1-x) \le E(x) - x \le (1-x) - c_4(d)(1-x)$. So, $B_L(d)(1-x) \le E(x) - x \le B_U(d)(1-x)$, where

$$B_L(d) = 1 - c_5(d) = 1 - \frac{2(d+1)}{d+3} \frac{2^{(d-1)/2}}{(7/4)^{(d+1)/2}},$$

$$B_U(d) = 1 - c_4(d) = 1 - \frac{2(d+1)}{d+3} \frac{(7/4)^{(d-1)/2}}{2^{(d+2)/2}}.$$

Since $E(x) - x = \mathbb{E}[X_1 - x \mid X_1 > x]$ must be positive (as $X_1 > x$), we take $B_L(d) = \max(0, 1 - c_5(d))$.

3. Bounds for $\sqrt{1 + E(x)^2}$: We know $1 - c_5(d)(1-x) \le E(x) \le 1 - c_4(d)(1-x)$. For $x \in [3/4, 1)$, $(1-x) \in (0, 1/4]$ and the upper bound of $E(x)$ is given by

$$E(x) \le 1 - c_4(d)(1-x) < 1.$$

The lower bound is also given in this way

$$E(x) = \mathbb{E}[X_1 \mid X_1 > x] \ge x \ge 3/4.$$

Therefore, we deduce that

$$C_L(d) \le \sqrt{1 + E(x)^2} \le C_U(d),$$

where

$$C_L(d) = \frac{5}{4}, C_U(d) \quad = \sqrt{2}.$$

Combining these bounds, for $x \in [3/4, 1)$:

$$g^+(x) \ge A_L(d)B_L(d)C_L(d)(1-x)^{d+1}(1-x) = c_L^{(g)}(d)(1-x)^{d+2},$$

$$g^+(x) \le A_U(d)B_U(d)C_U(d)(1-x)^{d+1}(1-x) = c_U^{(g)}(d)(1-x)^{d+2}.$$

The constants are:

$$\begin{aligned} c_L^{(g)}(d) &= (c_2(d))^2 \cdot (1 - c_5(d)) \cdot 5/4, \\ c_U^{(g)}(d) &= (c_3(d))^2 \cdot (1 - c_4(d)) \cdot \sqrt{2}. \end{aligned} \tag{39}$$

$\square$

## E.2 Empirical Process for the Weight Function

In this section, we discuss the empirical process of $g_P$. Now we may relax the assumption by just assuming that $X$ is random variable with $\mathrm{supp}(X) \subseteq \mathbb{B}_1^d$. Fix dimension $d \in \mathbb{N}$, sample size $n \in \mathbb{N}$, and let $X_1, \ldots, X_n$ be i.i.d. copies of $X$. We use the notation $\hat{g}_n$ to denote empirical weight function $g$ as we defined previously.

For $u \in \mathbb{S}^{d-1}$ and $t \in [-1, 1]$, define

$$p(u, t) := \mathbb{P}(X^\top u > t), \qquad s(u, t) := \mathbb{E}[(X^\top u - t)_+],$$

and recall the population weight $g_P(\boldsymbol{u}, t)$. By the bounds $0 \le (\boldsymbol{X}^\mathsf{T}\boldsymbol{u} - t)_+ \le 2$ and $\|\mathbb{E}[\boldsymbol{X} \mid \boldsymbol{X}^\mathsf{T}\boldsymbol{u} > t]\| \le 1$ (valid on $\mathbb{B}_1^d$), we have the pointwise comparison

$$g_P(\boldsymbol{u}, t) \asymp p(\boldsymbol{u}, t)\, s(\boldsymbol{u}, t) \quad \text{with absolute constants,} \tag{40}$$

i.e., there exist universal $c, C \in (0, \infty)$ such that $c\, p\, s \le g_P \le C\, p\, s$ for all $(\boldsymbol{u}, t) \in \mathbb{S}^{d-1} \times [-1, 1]$. Consider the empirical plug-ins

$$\hat{p}_n(\boldsymbol{u}, t) := \frac{1}{n}\sum_{i=1}^n \mathbb{1}\{\boldsymbol{X}_i^\mathsf{T}\boldsymbol{u} > t\}, \quad \hat{s}_n(\boldsymbol{u}, t) := \frac{1}{n}\sum_{i=1}^n (\boldsymbol{X}_i^\mathsf{T}\boldsymbol{u} - t)_+, \quad \hat{g}_n(\boldsymbol{u}, t) := \hat{p}_n(\boldsymbol{u}, t)\, \hat{s}_n(\boldsymbol{u}, t).$$

Note that $\hat{g}_n$ involves no division by $\hat{p}_n$, hence avoids any small-mass instability. We now give a self-contained proof of a sharp, distribution-free uniform deviation bound.

**Theorem E.5** (Distribution-free uniform deviation for $\hat{g}_n$)**.** *There exists a universal constant $C > 0$ such that, for every $\delta \in (0, 1)$,*

$$\mathbb{P}\left(\sup_{\boldsymbol{u} \in \mathbb{S}^{d-1},\, t \in [-1, 1]} \left|\hat{g}_n(\boldsymbol{u}, t) - g_P(\boldsymbol{u}, t)\right| > C\sqrt{\frac{d + \log(2/\delta)}{n}}\right) \le \delta.$$

*Proof.* Using (40), it is enough (up to absolute constants) to control $\left|\hat{p}_n(\boldsymbol{u}, t)\hat{s}_n(\boldsymbol{u}, t) - p(\boldsymbol{u}, t)s(\boldsymbol{u}, t)\right|$ uniformly over $(\boldsymbol{u}, t) \in \mathbb{S}^{d-1} \times [-1, 1]$. Observe that $0 \le s, \hat{s}_n \le 2$ and $0 \le p, \hat{p}_n \le 1$, so

$$\left|\hat{p}_n\hat{s}_n - p\, s\right| \le |\hat{p}_n - p|\, s + |\hat{s}_n - s|\, \hat{p}_n \le 2\,|\hat{p}_n - p| + |\hat{s}_n - s|. \tag{41}$$

We thus seek uniform bounds for the two empirical processes appearing on the right-hand side. The argument proceeds in two steps:

- **Halfspaces.** The class $\{x \mapsto \mathbb{1}\{x^\mathsf{T}\boldsymbol{u} > t\} : \boldsymbol{u} \in \mathbb{S}^{d-1},\, t \in \mathbb{R}\}$ has VC-dimension $d + 1$. Hence, by the VC uniform convergence inequality for $\{0, 1\}$-valued classes (e.g., [Vapnik, 1998]), there exists a universal constant $C_1 > 0$ such that, for all $\delta \in (0, 1)$,

$$\mathbb{P}\left(\sup_{\boldsymbol{u} \in \mathbb{S}^{d-1},\, t \in [-1, 1]} \left|\hat{p}_n(\boldsymbol{u}, t) - p(\boldsymbol{u}, t)\right| > C_1\sqrt{\frac{d + \log(1/\delta)}{n}}\right) \le \delta. \tag{42}$$

- **ReLU class.** Let $\mathcal{F} := \{f_{\boldsymbol{u}, t}(\boldsymbol{x}) = (\boldsymbol{u}^\mathsf{T}\boldsymbol{x} - t)_+ : \boldsymbol{u} \in \mathbb{S}^{d-1},\, t \in [-1, 1]\}$. Since $\|\boldsymbol{x}\| \le 1$ and $t \in [-1, 1]$, we have $f \in [0, 2]$. Consider the subgraph family

$$\mathsf{subG}(\mathcal{F}) = \{(\boldsymbol{x}, y) \in \mathbb{R}^d \times \mathbb{R} : y \le (\boldsymbol{u}^\mathsf{T}\boldsymbol{x} - t)_+\}.$$

For $y \le 0$ membership is automatic; for $y > 0$ it is equivalent to the affine halfspace condition $\boldsymbol{u}^\mathsf{T}\boldsymbol{x} - t - y \ge 0$ in $\mathbb{R}^{d+1}$. Thus $\mathrm{VCdim}(\mathsf{subG}(\mathcal{F})) \le d + 2$, whence $\mathrm{Pdim}(\mathcal{F}) \le d + 2$. Standard pseudo-dimension bounds (see [Haussler, 1992, Thm. 3, 6, 7]) give a universal $C_2 > 0$ with

$$\mathbb{P}\left(\sup_{\boldsymbol{u} \in \mathbb{S}^{d-1},\, t \in [-1, 1]} \left|\hat{s}_n(\boldsymbol{u}, t) - s(\boldsymbol{u}, t)\right| > C_2\sqrt{\frac{d + \log(1/\delta)}{n}}\right) \le \delta. \tag{43}$$

Combining Step (I) and Step (II) with a union bound and the previous inequality yields

$$\mathbb{P}\left(\sup_{\boldsymbol{u}, t} \left|\hat{p}_n\hat{s}_n - p\, s\right| > C'\sqrt{\frac{d + \log(2/\delta)}{n}}\right) \le \delta \tag{44}$$

for an absolute constant $C' > 0$. Finally, the equivalence (40) transfers this bound to $\sup_{\boldsymbol{u}, t} \left|\hat{g}_n - g_P\right|$, up to a universal multiplicative factor and the same failure probability. $\qquad\square$

# F  Proof of Theorem 3.5: Generalization Gap of Stable Minima

Let $\mathcal{P}$ denote the joint distribution of $(\boldsymbol{x}, y)$. Assume that $\mathcal{P}$ is supported on $\mathbb{B}_1^d \times [-D, D]$ for some $D > 0$. Let $f$ be a function. The *population risk* or *expected risk* of $f$ is defined to be

$$R(f) = \mathbb{E}_{(\boldsymbol{x}, y) \sim \mathcal{P}} \left[ (f(\boldsymbol{x}) - y)^2 \right] \tag{45}$$

Let $\mathcal{D} = \{(\boldsymbol{x}_i, y_i)\}_{i=1}^n$ be a data set where each $(\boldsymbol{x}_i, y_i)$ is drawn i.i.d. from $\mathcal{P}$. Then the *empirical risk* is defined to be

$$\hat{R}(f) = \frac{1}{n} \sum_{i=1}^n (f(\boldsymbol{x}_i) - y_i)^2 \tag{46}$$

The *generalization gap* is defined to be

$$\mathrm{GeneralizationGap}(f; \hat{R}) := |R(f) - \hat{R}(f)|. \tag{47}$$

The generalization gap measures the difference between the train loss and the expected testing error. The smaller the generalization gap, the less likely the model overfits.

## F.1  Definition of the Variation Space of ReLU Neural Networks

Recall the notion in Section 3, the *weighted variation (semi)norm* is defined to be

$$|f|_{\mathrm{V}_g} := \inf_{\substack{\nu \in \mathcal{M}(\mathbb{S}^{d-1} \times [-R, R]) \\ \boldsymbol{c} \in \mathbb{R}^d, c_0 \in \mathbb{R}}} \|g \cdot \nu\|_{\mathcal{M}} \quad \text{s.t.} \quad f = f_{\nu, \boldsymbol{c}, c_0}, \tag{48}$$

and now we define the *unweighted variation norm* or simply *variation norm* to be

$$|f|_{\mathrm{V}} := \inf_{\substack{\nu \in \mathcal{M}(\mathbb{S}^{d-1} \times [-R, R]) \\ \boldsymbol{c} \in \mathbb{R}^d, c_0 \in \mathbb{R}}} \|\nu\|_{\mathcal{M}} \quad \text{s.t.} \quad f = f_{\nu, \boldsymbol{c}, c_0}. \tag{49}$$

This definition is identical to the one in [Parhi and Nowak, 2023b, Section V.B]. The following example for unweighted variation norm is similar to Example 3.1.

**Example F.1.** *Since we are interested in functions defined on $\mathbb{B}_R^d$, for a finite-width neural network $f_{\boldsymbol{\theta}}(\boldsymbol{x}) = \sum_{k=1}^K v_k \phi(\boldsymbol{w}_k^{\mathsf{T}} \boldsymbol{x} - b_k) + \beta$, we observe that it has the equivalent implementation as $f_{\boldsymbol{\theta}}(\boldsymbol{x}) = \sum_{j=1}^J a_j \phi(\boldsymbol{u}_j^{\mathsf{T}} \boldsymbol{x} - t_j) + \boldsymbol{c}^{\mathsf{T}} \boldsymbol{x} + c_0$, where $a_j \in \mathbb{R}$, $\boldsymbol{u}_j \in \mathbb{S}^{d-1}$, $t_j \in \mathbb{R}$, $\boldsymbol{c} \in \mathbb{R}^d$, and $c_0 \in \mathbb{R}$. Indeed, this is due to the fact that the ReLU is homogeneous, which allows us to absorb the magnitude of the input weights into the output weights (i.e., each $a_j = |v_{k_j}| \|\boldsymbol{w}_{k_j}\|_2$ for some $k_j \in \{1, \ldots, K\}$). Furthermore, any ReLUs in the original parameterization whose activation threshold[8] is outside $\mathbb{B}_R^d$ can be implemented by an affine function on $\mathbb{B}_R^d$, which gives rise to the $\boldsymbol{c}^{\mathsf{T}} \boldsymbol{x} + c_0$ term in the implementation. If this new implementation is in "reduced form", i.e., the collection $\{(\boldsymbol{u}_j, t_j)\}_{j=1}^J$ are distinct, then we have that $|f_{\boldsymbol{\theta}}|_{\mathrm{V}} = \sum_{j=1}^J |a_j|$.*

The bounded variation function class is defined w.r.t. the unweighted variation norm.

**Definition F.2.** For the compact region $\Omega = \mathbb{B}_R^d$, we define the bounded variation function class as

$$\mathrm{V}_C(\Omega) := \left\{ f : \Omega \to \mathbb{R} \; \middle| \; f = \int_{\mathbb{S}^{d-1} \times [-R, R]} \phi(\boldsymbol{u}^{\mathsf{T}} \boldsymbol{x} - t) \, \mathrm{d}\nu(\boldsymbol{u}, t) + \boldsymbol{c}^{\mathsf{T}} \boldsymbol{x} + b, \; |f|_{\mathrm{V}} \leq C \right\}. \tag{50}$$

## F.2  Metric Entropy and Variation Spaces

Metric entropy quantifies the compactness of a set $A$ in a metric space $(X, \rho_X)$. Below we introduce the definition of covering numbers and metric entropy.

---

[8] The activation threshold of a neuron $\phi(\boldsymbol{w}^{\mathsf{T}} \boldsymbol{x} - b)$ is the hyperplane $\{\boldsymbol{x} \in \mathbb{R}^d : \boldsymbol{w}^{\mathsf{T}} \boldsymbol{x} = b\}$.

**Definition F.3** (Covering Number and Entropy). Let $A$ be a compact subset of a metric space $(X, \rho_X)$. For $t > 0$, the *covering number* $N(A, t, \rho_X)$ is the minimum number of closed balls of radius $t$ needed to cover $A$:

$$N(t, A, \rho_X) := \min \left\{ N \in \mathbb{N} : \exists\, x_1, \dots, x_N \in X \text{ s.t. } A \subset \bigcup_{i=1}^{N} \mathbb{B}(x_i, t) \right\}, \tag{51}$$

where $\mathbb{B}(x_i, t) = \{y \in X : \rho_X(y, x_i) \le t\}$. The *metric entropy* of $A$ at scale $t$ is defined as:

$$H_t(A)_X := \log N(t, A, \rho_X). \tag{52}$$

The metric entropy of the bounded variation function class has been studied in previous works. More specifically, we will directly use the one below in future analysis.

**Proposition F.4** (Parhi and Nowak 2023b, Appendix D). *The metric entropy of* $\mathrm{V}_C(\mathbb{B}_R^d)$ *(see Definition F.2) with respect to the* $L^\infty(\mathbb{B}_R^d)$*-distance* $\| \cdot \|_\infty$ *satisfies*

$$\log N(t, \mathrm{V}_C(\mathbb{B}_R^d), \| \cdot \|_\infty) \lesssim_d \left( \frac{C}{t} \right)^{\frac{2d}{d+3}}. \tag{53}$$

*where* $\lesssim_d$ *hides constants (which could depend on d) and logarithmic factors.*

### F.3 Generalization Gap of Unweighted Variation Function Class

As a middle step towards bounding the generalization gap of the weighted variation function class, we first bound the generalization gap of the unweighted variation function class according to a metric entropy analysis.

**Lemma F.5.** *Let* $\mathcal{F}_{M,C} = \{f \in \mathrm{V}_C(\mathbb{B}_R^d) \mid \|f\|_\infty \le M\}$ *with* $M \ge D$ *where* $D$ *refers to Theorem 3.5. Then with probability at least* $1 - \delta$:

$$\sup_{f \in \mathcal{F}_{M,C}} \left| R(f) - \widehat{R}_n(f) \right| \lesssim_d C^{\frac{d}{2d+3}} M^{\frac{3(d+2)}{2d+3}} n^{-\frac{d+3}{4d+6}}. \tag{54}$$

*Proof.* According to Proposition F.4, one just needs $N(t)$ balls to cover $\mathcal{F}$ in $\| \cdot \|_\infty$ with radius $t > 0$ such that where

$$\log N(t) \lesssim_d \left( \frac{C}{t} \right)^{\frac{2d}{d+3}}.$$

Then for any $f, g \in \mathcal{F}_{M,C}$ and any $(\boldsymbol{x}, y)$,

$$\left| (f(\boldsymbol{x}) - y)^2 - (g(\boldsymbol{x}) - y)^2 \right| = |f(\boldsymbol{x}) - g(\boldsymbol{x})|\, |f(\boldsymbol{x}) + g(\boldsymbol{x}) - 2y| \le 4M \|f - g\|_\infty.$$

Hence replacing $f$ by a centre $f_i$ within $t$ changes both the empirical and true risks by at most $4Mt$.

For any fixed centre $\bar{f}$ in the covering, Hoeffding's inequality implies that with probability at least $\ge 1 - \delta$, we have

$$|R(\bar{f}) - \hat{R}(\bar{f})| \le 4M^2 \sqrt{\frac{\log(2/\delta)}{n}} \tag{55}$$

because each squared error lies in $[0, 4M^2]$. Then we take all the centers with union bound to deduce that with probability at least $1 - \delta/2$, for any center $\bar{f}$ in the set of covering index, we have

$$\begin{aligned}
|R(\bar{f}) - \hat{R}(\bar{f})| &\le 4M^2 \sqrt{\frac{\log(4N(t)/\delta)}{n}} \\
&\lesssim_d M^2 \cdot \left( \frac{C}{t} \right)^{\frac{d}{d+3}} n^{-\frac{1}{2}}.
\end{aligned} \tag{56}$$

According to the definition of covering sets, for any $f \in \mathcal{F}_{M,C}$, we have that $\|f - \bar{f}\|_\infty \le t$ for some center $\bar{f}$. Then we have

$$\begin{aligned}
&|R(f) - \hat{R}(f)| \\
&\le |R(\bar{f}) - \hat{R}(\bar{f})| + O(Mt) \\
&\le M^2 \cdot \left( \frac{C}{t} \right)^{\frac{d}{d+3}} n^{-\frac{1}{2}} + O(Mt).
\end{aligned} \tag{57}$$

After tuning $t$ to be the optimal choice, we deduce that (54). $\qquad \square$

### F.4 Concentration Property on the Ball: Uniform Distribution

In the following analysis, we will handle the interior and boundary of the unit ball separately. In this part, we define the annulus of a ball rigorously and provide a high-probability bound on the number of samples falling in the annulus.

**Definition F.6.** Let $\mathbb{B}_1^d$ be the unit ball. The $\varepsilon$-*annulus* is a subset of $\mathbb{B}_1^d$ defined as

$$\mathbb{A}_\varepsilon^d := \{\boldsymbol{x} \in \mathbb{B}_1^d \mid \|\boldsymbol{x}\|_2 \geq 1 - \varepsilon\}$$

and the closure of its complement is called $\varepsilon$-*strict interior* and denoted by $\mathbb{I}_\varepsilon^d$.

**Lemma F.7** (High-Probability Upper Bound on Annulus). *Let $d \in \mathbb{N}$ and $\varepsilon \in (0, 1)$. Let*

$$\boldsymbol{x}_1, \ldots, \boldsymbol{x}_n \sim \mathrm{Uniform}\big(\mathbb{B}_1^d\big).$$

*Define $n_A := |\{\, i \mid \boldsymbol{x}_i \in \mathbb{A}_\varepsilon^d\}|$ and $p = \mathbb{P}\big(\boldsymbol{X} \in \mathbb{A}_\varepsilon^d\big) = 1 - (1 - \varepsilon)^d = \Theta(\varepsilon)$. Then for any $\delta \in (0,1)$, with probability at least $1 - \delta$,*

$$\frac{n_A}{n} \leq p + \sqrt{\frac{3\, p\, \log\big(1/\delta\big)}{n}}. \tag{58}$$

*Proof.* For each $i = 1, \ldots, n$, consider a Bernoulli random variable

$$U = \mathbb{1}\{\boldsymbol{X} \in \mathbb{A}_\varepsilon^d\},$$

so that $\mathbb{E}[U] = p$ and regards $U_i$ as a sample. Then we may take $n_A = \sum_{i=1}^n U_i$. By the multiplicative Chernoff bound for the upper tail of a sum of independent Bernoulli variables,

$$\mathbb{P}\big(n_A > (1 + \gamma)\, n\, p\big) \leq \exp\Big(-\tfrac{\gamma^2}{3}\, n\, p\Big), \qquad \forall\, \gamma > 0.$$

Set the right-hand side equal to $\delta$ and solve for $\gamma$:

$$\exp\Big(-\tfrac{\gamma^2}{3}\, n\, p\Big) = \delta \quad \Longrightarrow \quad -\tfrac{\gamma^2}{3}\, n\, p = \ln \delta \quad \Longrightarrow \quad \gamma = \sqrt{\frac{3\, \ln(1/\delta)}{n\, p}}.$$

If $\gamma > 1$, note that trivially $n_A/n \leq 1 \leq p + \sqrt{\frac{3\, p\, \ln(1/\delta)}{n}}$, so the claimed bound holds in all cases. Otherwise, plugging this choice of $\gamma$ into the Chernoff bound gives

$$\mathbb{P}\Big(n_A \leq n\, p\, (1 + \gamma)\Big) \geq 1 - \delta,$$

i.e. with probability at least $1 - \delta$,

$$n_A \leq n\, p + \sqrt{3\, n\, p\, \ln(1/\delta)},$$

and dividing by $n$ yields the stated inequality. $\qquad\square$

### F.5 Upper Bound of Generalization Gap of Stable Minima

Let $f = f_{\boldsymbol{\theta}}$ be a stable solution of the loss function $\mathcal{L}(\boldsymbol{\theta})$, trained by gradient descent with learning rate $\eta$. Then we have

$$\frac{2}{\eta} \geq \lambda_{\max}(\nabla_{\boldsymbol{\theta}}^2 \mathcal{L}(\boldsymbol{\theta})) \geq \boldsymbol{v}^\mathsf{T} \nabla_{\boldsymbol{\theta}}^2 \mathcal{L}(\boldsymbol{\theta}) \boldsymbol{v}$$

$$= \underbrace{\lambda_{\max}\left(\frac{1}{n} \sum_{i=1}^n (\nabla_{\boldsymbol{\theta}} f_{\boldsymbol{\theta}}(\boldsymbol{x}_i))(\nabla_{\boldsymbol{\theta}} f_{\boldsymbol{\theta}}(\boldsymbol{x}_i))^\mathsf{T}\right)}_{\text{(Term A)}} \tag{59}$$

$$+ \underbrace{\frac{1}{n} \sum_{i=1}^n (f_{\boldsymbol{\theta}}(\boldsymbol{x}_i) - y_i) \boldsymbol{v}^\mathsf{T} \nabla_{\boldsymbol{\theta}}^2 f_{\boldsymbol{\theta}}(\boldsymbol{x}_i) \boldsymbol{v}}_{\text{(Term B)}}.$$

For (Term A), we have

$$\lambda_{\max}\left(\frac{1}{n}\sum_{i=1}^{n}(\nabla_{\boldsymbol{\theta}}f_{\boldsymbol{\theta}}(\boldsymbol{x}_i))(\nabla_{\boldsymbol{\theta}}f_{\boldsymbol{\theta}}(\boldsymbol{x}_i))^{\mathsf{T}}\right) \geq 1 + 2|f_{\boldsymbol{\theta}}|_{\mathrm{V}_g}. \tag{60}$$

For (Term B), we have

$$|(\text{Term B})| \leq \sqrt{\frac{1}{n}\sum_{i=1}^{n}(f_{\boldsymbol{\theta}}(\boldsymbol{x}_i) - y_i)^2} \cdot \sqrt{\frac{1}{n}\sum_{i=1}^{n}(\boldsymbol{v}^{\mathsf{T}}\nabla_{\boldsymbol{\theta}}^2 f_{\boldsymbol{\theta}}(\boldsymbol{x}_i)\boldsymbol{v})^2} \leq 4\sqrt{2\mathcal{L}(\boldsymbol{\theta})}. \tag{61}$$

Let $M = \max\{\|f\|_{\infty}, D, 1\}$. Then we have

$$\sqrt{2\mathcal{L}(\boldsymbol{\theta})} = \sqrt{\frac{1}{n}\sum_{i=1}^{n}(f_{\boldsymbol{\theta}}(\boldsymbol{x}_i) - y_i)^2} \leq 2M.$$

Combining these inequalities together, we may deduce that

$$|f_{\boldsymbol{\theta}}|_{\mathrm{V}_g} \leq \frac{1}{\eta} - \frac{1}{2} + 4M. \tag{62}$$

With all the preparations, we are ready to prove the generalization gap upper bound for stable minima.

**Theorem F.8.** *(First part of Theorem 3.5) Let $\mathcal{P}$ denote the joint distribution of $(\boldsymbol{x}, y)$. Assume that $\mathcal{P}$ is supported on $\mathbb{B}_1^d \times [-D, D]$ for some $D > 0$ and that the marginal distribution of $\boldsymbol{x}$ is* Uniform($\mathbb{B}_1^d$). *Fix a data set $\mathcal{D} = \{(\boldsymbol{x}_i, y_i)\}_{i=1}^n$, where each $(\boldsymbol{x}_i, y_i)$ is drawn i.i.d. from $\mathcal{P}$, and $\mathcal{D}$ yields the empirical weight function $g$ defined in (6). Then, with probability at least $1 - \delta$, we have that for the plug-in risk estimator $\hat{R}(f) := \frac{1}{n}\sum_{i=1}^n (f(\boldsymbol{x}_i) - y_i)^2$,*

$$\sup_{\substack{f \in \mathrm{V}_g(\mathbb{B}_1^d) \\ |f|_{\mathrm{V}_g} \leq A, \|f\|_{L^\infty} \leq B}} \text{GeneralizationGap}(f; \hat{R}) := |R(f) - \hat{R}(f)| \lesssim_d A^{\frac{d}{d^2+4d+3}} B^2 n^{-\frac{1}{2d+4}},$$

*where $B$ is assumed $> 1$ and $\lesssim_d$ hides constants (which could depend on $d$) and logarithmic factors in $n$ and $(1/\delta)$. In particular, Theorem 3.2 and (62) imply that that*

$$f_{\boldsymbol{\theta}} \in \left\{ f \in \mathrm{V}_g(\mathbb{B}_1^d) \,\middle|\, |f|_{\mathrm{V}_g} \leq \frac{1}{\eta} - \frac{1}{2} + 4M, \|f\|_{L^\infty(\mathbb{B}_1^d)} \leq M \right\} \tag{63}$$

*for every*

$$\boldsymbol{\theta} \in \left\{ \boldsymbol{\theta} \in \Theta_{\mathrm{flat}}(\eta; \mathcal{D}) \,\middle|\, \|f\|_{L^\infty(\mathbb{B}_1^d)} \leq M \right\}. \tag{64}$$

*Therefore, we may conclude that*

$$\sup_{\boldsymbol{\theta} \in \Theta_{\mathrm{flat}}(\eta; \mathcal{D})} \text{GeneralizationGap}(f_{\boldsymbol{\theta}}; \hat{R}) := |R(f_{\boldsymbol{\theta}}) - \hat{R}(f_{\boldsymbol{\theta}})|$$

$$\lesssim_d \left(\frac{1}{\eta} - \frac{1}{2} + 4M\right)^{\frac{d}{d^2+4d+3}} M^2 n^{-\frac{1}{2d+4}}, \tag{65}$$

*where $M := \max\left\{ D, \|f_{\boldsymbol{\theta}}\|_{L^\infty(\mathbb{B}_1^d)}, 1 \right\}$.*

*Proof.* For any fixed $\varepsilon < 1/4$, we may decompose $\mathbb{B}_1^d$ into $\varepsilon$-annulus and $\varepsilon$-strict interior

$$\mathbb{B}_1^d = \mathbb{A}_\varepsilon^d \cup \mathbb{I}_\varepsilon^d.$$

According to the law of total expectation, the population risk is decomposed into

$$\mathbb{E}_{(\boldsymbol{x}, y) \sim \mathcal{P}}\left[(f(\boldsymbol{x}) - y)^2\right] = \mathbb{P}(\boldsymbol{x} \in \mathbb{A}_\varepsilon^d) \cdot \mathbb{E}_{\mathbb{A}}\left[(f(\boldsymbol{x}) - y)^2\right] + \mathbb{P}(\boldsymbol{x} \in \mathbb{I}_\varepsilon^d) \cdot \mathbb{E}_{\mathbb{I}}\left[(f(\boldsymbol{x}) - y)^2\right], \tag{66}$$

where $\mathbb{E}_{\mathbb{A}}$ means that $\{\boldsymbol{x}, y\}$ is a new sample from the data distribution conditioned on $\boldsymbol{x} \in \mathbb{A}_\varepsilon^d$ and $\mathbb{E}_{\mathbb{I}}$ means that $(\boldsymbol{x}, y)$ is a new sample from the data distribution conditioned on $\boldsymbol{x} \in \mathbb{I}_\varepsilon^d$.

Similarly, we also have this decomposition for empirical risk

$$
\begin{aligned}
\frac{1}{n} \sum_{i=1}^{n} (f(\boldsymbol{x}_i) - y_i)^2 &= \frac{1}{n} \left( \sum_{i \in I} (f(\boldsymbol{x}_i) - y_i)^2 + \sum_{j \in A} (f(\boldsymbol{x}_j) - y_j)^2 \right) \\
&= \frac{n_I}{n} \frac{1}{n_I} \sum_{i \in I} (f(\boldsymbol{x}_i) - y_i)^2 + \frac{n_A}{n} \frac{1}{n_A} \sum_{j \in A} (f(\boldsymbol{x}_j) - y_j)^2,
\end{aligned} \tag{67}
$$

where $I$ is the set of data points with $\boldsymbol{x}_i \in \mathbb{I}_\varepsilon^d$ and $A$ is the set of data points with $\boldsymbol{x}_i \in \mathbb{A}_\varepsilon^d$. Then the generalization gap can be decomposed into

$$
|R(f) - \hat{R}(f)| \leq \mathbb{P}(\boldsymbol{x} \in \mathbb{A}_\varepsilon^d) \cdot \mathbb{E}_{\mathbb{A}} \left[ (f_{\boldsymbol{\theta}}(\boldsymbol{x}) - y)^2 \right] + \frac{n_A}{n} \frac{1}{n_A} \sum_{j \in A} (f(\boldsymbol{x}_j) - y_j)^2 + \tag{68}
$$

$$
+ \left| \mathbb{P}(\boldsymbol{x} \in \mathbb{I}_\varepsilon^d) - \frac{n_I}{n} \right| \frac{1}{n_I} \sum_{i \in I} (f(\boldsymbol{x}_i) - y_i)^2 \tag{69}
$$

$$
+ \mathbb{P}(\boldsymbol{x} \in \mathbb{I}_\varepsilon^d) \cdot \left| \mathbb{E}_{\mathbb{I}} \left[ (f(\boldsymbol{x}) - y)^2 \right] - \frac{1}{n_I} \sum_{i \in I} (f(\boldsymbol{x}_i) - y_i)^2 \right|. \tag{70}
$$

Using the property that the marginal distribution of $\boldsymbol{x}$ is $\mathrm{Uniform}(\mathbb{B}_1^d)$ and its concentration property (see Lemma F.7), with probability at least $1 - \delta/2$:

$$
(68) \lesssim_d O(B^2 \varepsilon), \tag{71}
$$

where $\lesssim_d$ hides the constants that could depend on $d$ and logarithmic factors of $1/\delta$.

For the term (69), with probability $1 - \delta/3$

$$
\begin{cases}
\left| \mathbb{P}(\boldsymbol{x} \in \mathbb{I}_\varepsilon^d) - \frac{n_I}{n} \right| & \lesssim \sqrt{\frac{\varepsilon \log(3/\delta)}{n}}, \quad \text{(Lemma F.7)} \\
\frac{1}{n_I} \sum_{i \in I} (f(\boldsymbol{x}_i) - y_i)^2 & \leq 4B^2
\end{cases} \tag{72}
$$

so we may also conclude that

$$
(69) \lesssim M^2 \sqrt{\frac{\varepsilon \log(3/\delta)}{n}} \tag{73}
$$

For the part of the interior (70), the scalar $\mathbb{P}(\boldsymbol{x} \in \mathbb{I}_\varepsilon^d)$ is less than 1 with high-probability. Therefore, we just need to deal with the term

$$
\mathbb{E}_{\mathbb{I}} \left[ (f(\boldsymbol{x}) - y)^2 \right] - \frac{1}{n_I} \sum_{i \in I} (f(\boldsymbol{x}_i) - y_i)^2. \tag{74}
$$

Since both the distribution and sample points only support in $\mathbb{I}_\varepsilon^d$, we may consider $f$ by its restrictions in $\mathbb{I}_\varepsilon^d$, which are denoted by $f^\varepsilon$. Furthermore, according to the definition, we have

$$
\begin{aligned}
f(\boldsymbol{x}) &= \int_{\mathbb{S}^{d-1} \times [-1,1]} \phi(\boldsymbol{u}^{\mathsf{T}} \boldsymbol{x} - t) \, \mathrm{d}\nu(\boldsymbol{u}, t) + \boldsymbol{c}^{\mathsf{T}} \boldsymbol{x} + b \\
&= \int_{\mathbb{S}^{d-1} \times [-1+\varepsilon, 1-\varepsilon]} \phi(\boldsymbol{u}^{\mathsf{T}} \boldsymbol{x} - t) \, \mathrm{d}\nu(\boldsymbol{u}, t) + \underbrace{\int_{\mathbb{S}^{d-1} \times [-1, -1+\varepsilon) \cup (1-\varepsilon, 1]} \phi(\boldsymbol{u}^{\mathsf{T}} \boldsymbol{x} - t) \, \mathrm{d}\nu(\boldsymbol{u}, t)}_{\text{Annulus ReLU}} \\
&+ \boldsymbol{c}^{\mathsf{T}} \boldsymbol{x} + b
\end{aligned} \tag{75}
$$

where the Annulus ReLU term is totally linear in the strictly interior i.e. there exists $\boldsymbol{c}', b'$ such that

$$
\boldsymbol{c}'^{\mathsf{T}} \boldsymbol{x} + b' = \int_{\mathbb{S}^{d-1} \times [-1, -1+\varepsilon) \cup (1-\varepsilon, 1]} \phi(\boldsymbol{u}^{\mathsf{T}} \boldsymbol{x} - t) \, \mathrm{d}\nu(\boldsymbol{u}, t), \quad \forall \boldsymbol{x} \in \mathbb{I}_\varepsilon^d. \tag{76}
$$

Therefore, we may write

$$f(\boldsymbol{x}) = f^\varepsilon(\boldsymbol{x}) = \int_{\mathbb{S}^{d-1}\times[-1+\varepsilon,1-\varepsilon]} \phi(\boldsymbol{u}^\mathsf{T}\boldsymbol{x} - t)\, \mathrm{d}\nu(\boldsymbol{u}, t) + (\boldsymbol{c} + \boldsymbol{c}')^\mathsf{T}\boldsymbol{x} + b + b', \quad \boldsymbol{x} \in \mathbb{I}_\varepsilon^d. \quad (77)$$

The core of the argument is to rigorously bound the interior generalization gap. Recall that a stable minima $\boldsymbol{\theta} \in \Theta_{\mathrm{flat}}(\eta; \mathcal{D})$ satisfies $|f|_{V_g} \leq A$ with respect to the empirical weight function $g$. To analyze the complexity of its restriction $f^\varepsilon$ on the core $\mathbb{I}_\varepsilon^d$, we need a lower bound on $g_{\min}^\varepsilon := \inf_{|t|\leq 1-\varepsilon} g(\boldsymbol{u}, t)$. This quantity is a random variable.

From empirical process we discussed in Section E.2, especially Theorem E.5, we know that with probability at least $1 - \delta/3$,

$$\sup_{\boldsymbol{u},t} |g(\boldsymbol{u},t) - g_P(\boldsymbol{u},t)| \lesssim_d \sqrt{\frac{d + \log(6/\delta)}{n}} =: \epsilon_n. \quad (78)$$

This implies a lower bound on the empirical minimum weight in the core with probability at least $1 - \delta/3$,

$$g_{\min}^\varepsilon = \inf_{|t|\leq 1-\varepsilon} g(\boldsymbol{u}, t) \geq \underbrace{\inf_{|t|\leq 1-\varepsilon} g_P(\boldsymbol{u}, t)}_{g_{P,\min}^\varepsilon} - \epsilon_n = g_{P,\min}^\varepsilon - \epsilon_n. \quad (79)$$

Here, $g_{P,\min}^\varepsilon \asymp \varepsilon^{d+2}$ is the minimum of the population weight function in the core.

For the bound $|f^\varepsilon|_V \leq A/g_{\min}^\varepsilon \leq A/(g_{P,\min} - \epsilon_n)$ to be meaningful with high probability, we must operate in a regime where $g_{\min} \geq \epsilon_n$. We enforce a stricter **validity condition** for our proof

$$g_{\min} \geq 2\epsilon_n \quad \Longrightarrow \quad \varepsilon^{d+2} \gtrsim_d n^{-\frac{1}{2}}. \quad (80)$$

Therefore, we may choose

$$\varepsilon \asymp \left( A^{\frac{d}{d^2+4d+3}} \cdot \sqrt{\frac{d + \log(6/\delta)}{n}} \right)^{\frac{1}{d+2}} \quad (81)$$

Under this condition, we have $g_{\min}^\varepsilon \geq g_{P,\min}^\varepsilon - \epsilon_n \geq g_{P,\min}^\varepsilon/2 \asymp \varepsilon^{d+2}$. Thus, for any stable solution $f$, its restriction $f^\varepsilon$ has a controlled unweighted variation norm with high probability:

$$|f^\varepsilon|_{V(\mathbb{B}_{1-\varepsilon}^d)} \leq \frac{A}{g_{\min}^\varepsilon} \leq \frac{A}{g_{P,\min}^\varepsilon/2} \asymp \frac{A}{\varepsilon^{d+2}} =: C_\varepsilon.$$

We can now apply the generalization bound from Lemma F.5 to the class $V_{C_\varepsilon}(\mathbb{B}_{1-\varepsilon}^d)$ by plugging in (81), with probability $1 - \delta/3$,

$$\text{Interior Gap (70)} \lesssim_d (C_\varepsilon)^{\frac{d}{2d+3}} B^{\frac{3(d+2)}{2d+3}} n^{-\frac{d+3}{4d+6}} \quad (82)$$

$$= \left( A^{1-\frac{d}{d^2+4d+3}} \sqrt{\frac{n}{d + \log(6/\delta)}} \right)^{\frac{d}{2d+3}} B^{\frac{3(d+2)}{2d+3}} n^{-\frac{d+3}{4d+6}} \quad (83)$$

$$\lesssim_d A^{\frac{d}{d^2+4d+3}} B^{\frac{3(d+2)}{2d+3}} n^{-\frac{3}{4d+6}} \quad (84)$$

where $\lesssim_d$ hides the constants that could depend on $d$ and logarithmic factors of $1/\delta$.

Now we combine the upper bounds (71), (73) and (84) to deduce an upper bound of the generalization gap. With probability $1 - \delta$, we have

$$|R(f) - \hat{R}(f)| \lesssim_d A^{\frac{d}{d^2+4d+3}} B^2 n^{-\frac{1}{2d+4}} + A^{\frac{d}{d^2+4d+3}} B^{\frac{3(d+2)}{2d+3}} n^{-\frac{3}{4d+6}}. \quad (85)$$

Since $n^{-\frac{1}{2d+4}} > n^{-\frac{3}{4d+6}}$ and $B^2 > B^{\frac{3(d+2)}{2d+3}}$ with the assumption $M \geq 1$, we conclude that

$$|R(f) - \hat{R}(f)| \lesssim_d \left( \frac{1}{\eta} - \frac{1}{2} + 4B \right)^{\frac{d}{d^2+4d+3}} M^2 n^{-\frac{1}{2d+4}}, \quad (86)$$

which finishes the proof. $\qquad\square$

**Remark F.9.** For the generalization gap lower bound (second part of Theorem 3.5), we defer the proof to Appendix I as it relies on a construction that is used to prove Theorem 3.7 from Appendix H.

# G  Proof of Theorem 3.6: Estimation Error Rate for Stable Minima

## G.1  Computation of Local Gaussian Complexity

It is known from Wainwright 2019 that a tight analysis of MSE results from *local gaussian complexity*. We begin with the following proposition that connects the local gaussian complexity to the critical radius.

**Proposition G.1** (Wainwright 2019, Chapter 13). *Let $\mathcal{F}$ be a* convex *model class that contains the constant function* $1$. *Fix design points $\boldsymbol{x}_1, \ldots, \boldsymbol{x}_n$ in the region of interest and denote the empirical norm*

$$\|f\|_n^2 := \frac{1}{n} \sum_{i=1}^n f(\boldsymbol{x}_i)^2.$$

*For any radius $r > 0$ write*

$$\mathcal{F}(r) := \big\{ f \in \mathcal{F} : \|f\|_n \leq r \big\}, \quad \widehat{\mathcal{G}}_n(r, \mathcal{F}) := \sup_{f \in \mathcal{F}(r)} \frac{1}{n} \sum_{i=1}^n \varepsilon_i f(\boldsymbol{x}_i),$$

*where $\varepsilon_1, \ldots, \varepsilon_n \overset{i.i.d.}{\sim} \mathcal{N}(0, \sigma^2)$ and $\mathcal{G}_n(r, \mathcal{F}) := \mathbb{E}\, \widehat{\mathcal{G}}_n(r, \mathcal{F})$.*

*If $\delta$ satisfies the integral inequality*

$$\frac{16}{\sqrt{n}} \int_0^r \sqrt{\log N\big(t,\, \partial \mathcal{F},\, \|\cdot\|_n\big)}\, \mathrm{d}t \;\leq\; \frac{r}{4}, \tag{87}$$

*where $\partial \mathcal{F} := \big\{ f_1 - f_2 : f_1, f_2 \in \mathcal{F} \big\}$, then the* local empirical Gaussian complexity *obeys*

$$\frac{\mathcal{G}_n(r, \mathcal{F})}{r} \;\leq\; \frac{r}{2\sigma}. \tag{88}$$

*Moreover, with probability at least $1 - \delta$ one has*

$$\widehat{\mathcal{G}}_n(r, \mathcal{F}) \;\leq\; \frac{r^2}{2\sigma} + r\, \frac{\sqrt{\log(1/\delta)}}{\sqrt{n}} \qquad (\delta > 0). \tag{89}$$

As a result, we can derive an upper bound for the local empirical Gaussian complexity of the variation function class through a careful analysis of the critical radius.

**Lemma G.2.** *Let $\mathcal{F}_{B,C}(\mathbb{B}_R^d) = \{ f \in \mathrm{V}_C(\mathbb{B}_R^d) \mid \|f\|_{L^\infty(\mathbb{B}_R^d)} \leq B \}$. Then with probability at least $1 - \delta$, we have*

$$\frac{1}{n} \sum_{i=1}^n \varepsilon_i(f_1(\boldsymbol{x}_i) - f_2(\boldsymbol{x}_i)) \lesssim_d C^{\frac{2d}{2d+3}}\, n^{-\frac{d+3}{2d+3}} + C^{\frac{d}{2d+3}}\, n^{-\frac{3d+6}{4d+6}}\, \sqrt{\log(1/\delta)}, \tag{90}$$

*for any two $f_1, f_2 \in \mathcal{F}_{B,C}$.*

*Proof.* As $\partial \mathcal{F}_{B,C} = 2\mathcal{F}_{B,C} \subset \mathcal{F}_{2B,2C}$, bounding the entropy of $\mathcal{F}_{2B,2C}$ suffices. Using $\|f\|_n \leq \|f\|_{L^\infty(\mathbb{B}_1^d)}$ and referring to Proposition F.4, we have, up to logarithmic factors,

$$\log N\big(t, \mathcal{F}_{2B,2C}, \|\cdot\|_n\big) \;\lesssim_d\; \left(\frac{C}{t}\right)^{\frac{2d}{d+3}}.$$

Plugging this entropy bound into the left side of (87) and integrating,

$$\frac{16}{\sqrt{n}} \int_0^r \left(\frac{C}{t}\right)^{\frac{d}{d+3}} \mathrm{d}t \;\lesssim_d\; \frac{C^{\frac{d}{d+3}}}{\sqrt{n}} \int_0^r t^{-\frac{d}{d+3}}\, \mathrm{d}t \;=\; \frac{C^{\frac{d}{d+3}}\, r^{\frac{3}{d+3}}}{\sqrt{n}}.$$

Hence inequality (87) is met provided

$$\frac{C^{\frac{d}{d+3}}\, r^{\frac{3}{d+3}}}{\sqrt{n}} \;\lesssim_d\; \frac{r}{4}, \quad \Longleftrightarrow \quad r^{\frac{d}{d+3}} \;\gtrsim_d\; C^{\frac{d}{d+3}}\, n^{-1/2}.$$

Solving for $r^2$ (and keeping only dominant terms) yields

$$r_n^2 \;\asymp_d\; C^{\frac{2d}{2d+3}}\, n^{-\frac{d+3}{2d+3}}.$$

With this choice of $r_n$, Proposition G.1 guarantees

$$\mathcal{G}_n\big(\mathcal{F}_{B,C}(r_n)\big) \;\lesssim_d\; r_n,$$

and the high-probability version (89) holds verbatim. $\qquad\qquad\square$

## G.2 Proof of the Estimation Error Upper Bound

Given the local gaussian complexity upper bound, together with the assumption of solutions being "optimized", we can prove the following MSE upper bound.

**Theorem G.3** (Restate Theorem 3.6). *Fix a step size $\eta > 0$ and noise level $\sigma > 0$. Given a ground truth function $f_0 \in V_g(\mathbb{B}_1^d)$ such that $\|f_0\|_{L^\infty} \leq B$ and $|f_0|_{V_g} \leq \widetilde{O}\left(\frac{1}{\eta} - \frac{1}{2} + 2\sigma\right)$, suppose that we are given a data set $y_i = f_0(\boldsymbol{x}_i) + \varepsilon_i$, where $\boldsymbol{x}_i$ are i.i.d. $\mathrm{Uniform}(\mathbb{B}_1^d)$ and $\varepsilon_i$ are i.i.d. $\mathcal{N}(0, \sigma^2)$. Then, with probability at least $1 - \delta$, we have that*

$$\frac{1}{n}\sum_{i=1}^n (f_{\boldsymbol{\theta}}(\boldsymbol{x}_i) - f_0(\boldsymbol{x}_i))^2 \lesssim_d \left(\frac{1}{\eta} - \frac{1}{2} + 2\sigma\right)^{\frac{d}{(2d^2+6d+3)(d+2)}} B^2 \left(\frac{\sigma^2}{n}\right)^{\frac{1}{2d+4}}, \tag{91}$$

*for any $\boldsymbol{\theta} \in \Theta_{\mathrm{flat}}(\eta; \mathcal{D})$ that is optimized, i.e., $(f_{\boldsymbol{\theta}}(\boldsymbol{x}_i) - y_i)^2 \leq (f_0(\boldsymbol{x}_i) - y_i)^2$, for $i = 1, \ldots, n$. Here, $\lesssim_d$ hides constants (that could depend on $d$) and logarithmic factors in $n$ and $(1/\delta)$.*

*Proof of Theorem 3.6.* The empirical Mean Squared Error (MSE) we want to bound is $\mathrm{MSE}(f_{\boldsymbol{\theta}}, f_0) = \frac{1}{n}\sum_{i=1}^n (f_{\boldsymbol{\theta}}(\boldsymbol{x}_i) - f_0(\boldsymbol{x}_i))^2$.

First, we establish bounds on the regularity of $f_{\boldsymbol{\theta}}(\boldsymbol{x}) - f_0(\boldsymbol{x})$. The condition that $f_{\boldsymbol{\theta}}$ is "optimized" means $(f_{\boldsymbol{\theta}}(\boldsymbol{x}_i) - y_i)^2 \leq (f_0(\boldsymbol{x}_i) - y_i)^2$ for all $i$. Summing over $i$ and dividing by $n$, we have $\frac{1}{n}\sum_{i=1}^n (f_{\boldsymbol{\theta}}(\boldsymbol{x}_i) - y_i)^2 \leq \frac{1}{n}\sum_{i=1}^n (f_0(\boldsymbol{x}_i) - y_i)^2 = \frac{1}{n}\sum_{i=1}^n \varepsilon_i^2$. Since $\varepsilon_i \sim \mathcal{N}(0, \sigma^2)$, $\mathbb{E}[\frac{1}{n}\sum_{i=1}^n \varepsilon_i^2] = \sigma^2$. Standard concentration inequalities (e.g., for sums of $\chi^2(1)$ scaled variables) show that $\frac{1}{n}\sum_{i=1}^n \varepsilon_i^2 \lesssim \sigma^2$ with high probability (hiding logarithmic factors in $1/\delta$, which are absorbed into $\lesssim_d$). Thus, $2\mathcal{L}(\boldsymbol{\theta}) \lesssim \sigma^2$. For $\boldsymbol{\theta} \in \Theta_{\mathrm{flat}}(\eta; \mathcal{D})$, by Corollary 3.3 (with $R = 1$ for $\mathbb{B}_1^d$, so $R + 1 = 2$), we have

$$|f_{\boldsymbol{\theta}}|_{V_g} \leq \frac{1}{\eta} - \frac{1}{2} + 2\sqrt{2\mathcal{L}(\boldsymbol{\theta})} \leq \frac{1}{\eta} - \frac{1}{2} + 2\sigma. \tag{92}$$

Let $C := \frac{1}{\eta} - \frac{1}{2} + 2\sigma$. The theorem assumes $|f_0|_{V_g} \leq C$. Thus, we have $|f_0|_{V_g} \lesssim C$. The difference $f_{\boldsymbol{\theta}}(\boldsymbol{x}) - f_0(\boldsymbol{x})$ then satisfies

$$|f_{\boldsymbol{\theta}} - f_0|_{V_g} \leq |f_{\boldsymbol{\theta}}|_{V_g} + |f_0|_{V_g} \leq 2C. \tag{93}$$

Also, $\|f_{\boldsymbol{\theta}} - f_0\|_{L^\infty(\mathbb{B}_1^d)} \leq \|f_{\boldsymbol{\theta}}\|_{L^\infty(\mathbb{B}_1^d)} + \|f_0\|_{L^\infty(\mathbb{B}_1^d)} \leq B + B = 2B$.

The optimized condition $(f_{\boldsymbol{\theta}}(\boldsymbol{x}_i) - y_i)^2 \leq (f_0(\boldsymbol{x}_i) - y_i)^2$ implies $((f_{\boldsymbol{\theta}}(\boldsymbol{x}_i) - f_0(\boldsymbol{x}_i)) - \varepsilon_i)^2 \leq \varepsilon_i^2$. Expanding this gives $(f_{\boldsymbol{\theta}}(\boldsymbol{x}_i) - f_0(\boldsymbol{x}_i))^2 - 2(f_{\boldsymbol{\theta}}(\boldsymbol{x}_i) - f_0(\boldsymbol{x}_i))\varepsilon_i + \varepsilon_i^2 \leq \varepsilon_i^2$, which simplifies to

$$(f_{\boldsymbol{\theta}}(\boldsymbol{x}_i) - f_0(\boldsymbol{x}_i))^2 \leq 2(f_{\boldsymbol{\theta}}(\boldsymbol{x}_i) - f_0(\boldsymbol{x}_i))\varepsilon_i. \tag{94}$$

This inequality is crucial and holds for each data point.

We decompose the MSE based on the location of data points. Let $\mathbb{A}_\varepsilon^d := \{\boldsymbol{x} \in \mathbb{B}_1^d : \|\boldsymbol{x}\|_2 \geq 1 - \varepsilon\}$ be the annulus and $\mathbb{B}_{1-\varepsilon}^d$ be the inner core. Let $S_A := \{i : \boldsymbol{x}_i \in \mathbb{A}_\varepsilon^d\}$ and $S_I := \{i : \boldsymbol{x}_i \in \mathbb{B}_{1-\varepsilon}^d\}$. The total empirical MSE is

$$\begin{aligned}
\mathrm{MSE}(f_{\boldsymbol{\theta}}, f_0) &= \frac{1}{n}\sum_{i \in S_A}(f_{\boldsymbol{\theta}}(\boldsymbol{x}_i) - f_0(\boldsymbol{x}_i))^2 + \frac{1}{n}\sum_{i \in S_I}(f_{\boldsymbol{\theta}}(\boldsymbol{x}_i) - f_0(\boldsymbol{x}_i))^2 \\
&\leq \frac{n_A}{n}\left(\frac{1}{n_A}\sum_{i \in S_A}(f_{\boldsymbol{\theta}}(\boldsymbol{x}_i) - f_0(\boldsymbol{x}_i))^2\right) + \frac{n_I}{n}\left(\frac{1}{n_I}\sum_{i \in S_I}(f_{\boldsymbol{\theta}}(\boldsymbol{x}_i) - f_0(\boldsymbol{x}_i))^2\right) \\
&\leq \frac{n_A}{n}\underbrace{\left(\frac{1}{n_A}\sum_{i \in S_A}(f_{\boldsymbol{\theta}}(\boldsymbol{x}_i) - f_0(\boldsymbol{x}_i))^2\right)}_{\mathrm{MSE}_{\mathcal{S}}} + \underbrace{\frac{1}{n_I}\sum_{i \in S_I}(f_{\boldsymbol{\theta}}(\boldsymbol{x}_i) - f_0(\boldsymbol{x}_i))^2}_{\mathrm{MSE}_{\mathcal{I}}}
\end{aligned} \tag{95}$$

The contribution from the shell, $\mathrm{MSE}_{\mathcal{S}}$, is bounded using the $L^\infty$ norm of $f_{\boldsymbol{\theta}} - f_0$ and the concentration of points in the shell. Let $n_A := |S_A|$. By Lemma F.7, $n_A/n \lesssim \varepsilon$ with high probability.

$$\mathrm{MSE}_{\mathcal{S}} \leq \frac{n_A}{n}\|f_{\boldsymbol{\theta}} - f_0\|_{L^\infty}^2 \leq \frac{n_A}{n}(2B)^2 \lesssim_d B^2\varepsilon. \tag{96}$$

For the inner core's contribution, $\text{MSE}_{\mathcal{I}}$, we use Equation (94):

$$\text{MSE}_{\mathcal{I}} = \frac{1}{n} \sum_{i \in S_I} (f_{\boldsymbol{\theta}}(\boldsymbol{x}_i) - f_0(\boldsymbol{x}_i))^2 \le \frac{2}{n} \sum_{i \in S_I} (f_{\boldsymbol{\theta}}(\boldsymbol{x}_i) - f_0(\boldsymbol{x}_i))\, \varepsilon_i. \tag{97}$$

Let $n_I := |S_I|$. The empirical process term is $2\frac{n_I}{n} \left( \frac{1}{n_I} \sum_{i \in S_I} (f_{\boldsymbol{\theta}}(\boldsymbol{x}_i) - f_0(\boldsymbol{x}_i))\, \varepsilon_i \right)$. The function $f_{\boldsymbol{\theta}} - f_0$ restricted to $\mathbb{B}^d_{1-\varepsilon}$ has an unweighted variation norm. As shown in Appendix E, for $\boldsymbol{x} \in \text{Uniform}(\mathbb{B}^d_1)$, the population weight function $g_P(\boldsymbol{u}, t) \asymp (1 - |t|)^{d+2}$. For activation hyperplanes relevant to $\mathbb{B}^d_{1-\varepsilon}$ (i.e., $|t| \le 1 - \varepsilon$), $g_P(\boldsymbol{u}, t) \gtrsim \varepsilon^{d+2}$. Thus, the unweighted variation of $f_{\boldsymbol{\theta}} - f_0$ on $\mathbb{B}^d_{1-\varepsilon}$ is

$$|f_{\boldsymbol{\theta}} - f_0|_{\text{V}(\mathbb{B}^d_{1-\varepsilon})} \lesssim |f_{\boldsymbol{\theta}} - f_0|_{\text{V}_{g_P}}/\varepsilon^{d+2} \lesssim C/\varepsilon^{d+2}. \tag{98}$$

We apply Lemma G.2 to bound $\frac{1}{n_I} \sum_{i \in S_I} (f_{\boldsymbol{\theta}}(\boldsymbol{x}_i) - f_0(\boldsymbol{x}_i))\, \varepsilon_i$. The function $h(\boldsymbol{x}) = f_{\boldsymbol{\theta}}(\boldsymbol{x}) - f_0(\boldsymbol{x})$ has unweighted variation $\lesssim C/\varepsilon^{d+2}$ and $L^\infty$ norm $\le 2B$. Therefore, we have that

$$\text{MSE}_{\mathcal{I}} \lesssim \left( \frac{C}{\varepsilon^{d+2}} \right)^{\frac{2d}{2d+3}} \left( \frac{\sigma^2}{n} \right)^{\frac{d+3}{2d+3}}. \tag{99}$$

Combining the bounds for $\text{MSE}_{\mathcal{S}}$ and $\text{MSE}_{\mathcal{I}}$:

$$\text{MSE}(f_{\boldsymbol{\theta}}, f_0) \lesssim B^2 \varepsilon + C^{\frac{2d}{2d+3}} \varepsilon^{-(d+2)\frac{2d}{2d+3}} \left( \frac{\sigma^2}{n} \right)^{\frac{d+3}{2d+3}}. \tag{100}$$

Similarly to the proof of Theorem 3.5 in Appendix F.5, we require that

$$\frac{1}{\inf_{|t| \le 1-\varepsilon} g_P(\boldsymbol{u}, t)} \asymp \varepsilon^{d+2} \gtrsim_d \sqrt{\frac{1}{n}}. \tag{101}$$

to filling the gap between the empirical weighted function $g$ and the population $g_P$ with high probability, becase with high probability,

$$\sup_{\boldsymbol{u}, t} |g(\boldsymbol{u}, t) - g_P(\boldsymbol{u}, t)| \lesssim_d \sqrt{\frac{1}{n}} \tag{102}$$

where $\lesssim_d$ hides constants (which could depend on $d$) and logarithmic factors, as stated by by Theorem E.5 in Section E.2. Therefore, we may choose

$$\varepsilon \asymp \left( C^{\frac{2d}{2d^2+6d+3}} \cdot \frac{\sigma^2}{n} \right)^{\frac{1}{2d+4}}, \tag{103}$$

and plug it into (100) to have

$$\text{MSE}(f_{\boldsymbol{\theta}}, f_0) \lesssim_d \left( \frac{1}{\eta} - \frac{1}{2} + 2\sigma \right)^{\frac{d}{(2d^2+6d+3)(d+2)}} \left( B^2 \left( \frac{\sigma^2}{n} \right)^{\frac{1}{2d+4}} + \left( \frac{\sigma^2}{n} \right)^{\frac{3}{2d+3}} \right). \tag{104}$$

Since $\frac{1}{2d+4} < \frac{3}{2d+3}$, we conclude that

$$\text{MSE}(f_{\boldsymbol{\theta}}, f_0) \lesssim_d \left( \frac{1}{\eta} - \frac{1}{2} + 2\sigma \right)^{\frac{d}{(2d^2+6d+3)(d+2)}} B^2 \left( \frac{\sigma^2}{n} \right)^{\frac{1}{2d+4}},$$

which completes the proof. $\qquad\qquad\qquad\qquad\qquad\qquad\qquad\qquad\qquad\qquad\qquad\square$

# H Proof of Theorem 3.7: Minimax Lower Bound

## H.1 The Multivariate Case

In this section, we assume that $d > 1$ and all the norms and semi-norms are restricted to the unit ball $\mathbb{B}^d_1$. Let $\boldsymbol{u} \in \mathbb{S}^{d-1}$ be a unit vector. Let $\varepsilon \in \mathbb{R}_+$ be a constant with $\varepsilon \le 1/2$. Consider the ReLU atom:

$$\varphi_{\boldsymbol{u}, \varepsilon^2}(\boldsymbol{x}) = \phi(\boldsymbol{u}^\mathsf{T} \boldsymbol{x} - (1 - \varepsilon^2)). \tag{105}$$

**Lemma H.1.** *The $L^2$-norm of $\varphi_{\boldsymbol{u},\varepsilon^2}$ over $\mathbb{B}_1^d$ is given by*

$$\|\varphi_{\boldsymbol{u},\varepsilon^2}\|_{L^2(\mathbb{B}_1^d)} \underset{c_7(d)}{\overset{c_8(d)}{\asymp}} \varepsilon^{\frac{d+5}{2}}, \tag{106}$$

*where $c_7(d)$ and $c_8(d)$ are constants depends on $d$ (the conconcrete definition is* (113)*). Recall that $\underset{c_7(d)}{\overset{c_8(d)}{\asymp}}$ means*

$$c_7(d)\,\varepsilon^{\frac{d+5}{2}} \leq \|\varphi_{\boldsymbol{u},\varepsilon^2}\|_{L^2(\mathbb{B}_1^d)} \leq c_8(d)\,\varepsilon^{\frac{d+5}{2}}.$$

*Proof.* The squared $L^2$ norm of $\varphi_{\boldsymbol{u},\varepsilon^2}$ over the unit ball $\mathbb{B}_1^d$ is defined as:

$$\|\varphi_{\boldsymbol{u},\varepsilon^2}\|_{L^2(\mathbb{B}_1^d)}^2 = \int_{\mathbb{B}_1^d} |\varphi_{\varepsilon^2}(\boldsymbol{x})|^2 \, d\boldsymbol{x}$$

Substituting the definition of $\varphi_{\varepsilon^2}(\boldsymbol{w}, \boldsymbol{x})$ and using the property of the ReLU function that $\phi(z) = z$ for $z > 0$ and $\phi(z) = 0$ for $z \leq 0$, we get:

$$\begin{aligned}
\|\varphi_{\boldsymbol{u},\varepsilon^2}\|_{L^2(\mathbb{B}_1^d)}^2 &= \int_{\mathbb{B}_1^d} [\phi(\boldsymbol{u}^\mathsf{T}\boldsymbol{x} - (1-\varepsilon^2))]^2 \, d\boldsymbol{x} \\
&= \int_{\{\boldsymbol{x}\in\mathbb{B}_1^d \,:\, \boldsymbol{u}^\mathsf{T}\boldsymbol{x}>1-\varepsilon^2\}} (\boldsymbol{u}^\mathsf{T}\boldsymbol{x} - (1-\varepsilon^2))^2 \, d\boldsymbol{x}
\end{aligned} \tag{107}$$

To simplify the integral, we perform a rotation of the coordinate system such that $\boldsymbol{w}$ aligns with the $d$-th standard basis vector $e_d = (0, \ldots, 0, 1)$. In these new coordinates, $\boldsymbol{u}^\mathsf{T}\boldsymbol{x} = X_d$. The unit ball remains the unit ball under rotation. The integral becomes:

$$I = \|\varphi_{\boldsymbol{u},\varepsilon^2}\|_{L^2(\mathbb{B}_1^d)}^2 = \int_{\{X\in\mathbb{B}_1^d \,:\, X_d>1-\varepsilon^2\}} (X_d - (1-\varepsilon^2))^2 \, dX$$

We can write the volume element $dX$ as $dX' \, dX_d$, where $X' \in \mathbb{R}^{d-1}$ represents the first $d-1$ coordinates. The condition $X \in \mathbb{B}_1^d$ translates to $\|X'\|_2^2 + X_d^2 \leq 1$. The integral can be written as an iterated integral:

$$I = \int_{1-\varepsilon^2}^{1} \left( \int_{\|X'\|_2^2 \leq 1-X_d^2} (X_d - (1-\varepsilon^2))^2 \, dX' \right) dX_d$$

The inner integral is over a $(d-1)$-dimensional ball in $\mathbb{R}^{d-1}$ with radius $R = \sqrt{1 - X_d^2}$. The integrand $(X_d - (1-\varepsilon^2))^2$ is constant with respect to $X'$. Therefore, the inner integral evaluates to:

$$(X_d - (1-\varepsilon^2))^2 \cdot \text{Vol}_{d-1}(R)$$

where $\text{Vol}_{d-1}(R)$ is the volume of the $(d-1)$-dimensional ball of radius $R$. This volume is given by $V_{d-1}R^{d-1}$, with $V_{d-1} = \frac{\pi^{(d-1)/2}}{\Gamma(\frac{d+1}{2})}$. So, the inner integral is $(X_d - (1-\varepsilon^2))^2 V_{d-1}(1 - X_d^2)^{(d-1)/2}$, and the outer integral becomes:

$$I = V_{d-1} \int_{1-\varepsilon^2}^{1} (X_d - (1-\varepsilon^2))^2(1 - X_d^2)^{\frac{d-1}{2}} \, dX_d \tag{108}$$

Let $X_d = 1 - \delta$ performing a change of variable. Then $dX_d = -d\delta$. The integration limits change

$$\begin{aligned}
I &= V_{d-1} \int_{\varepsilon^2}^{0} ((1-\delta) - (1-\varepsilon^2))^2(1 - (1-\delta)^2)^{\frac{d-1}{2}}(-d\delta) \\
&= V_{d-1} \int_{0}^{\varepsilon^2} (\varepsilon^2 - \delta)^2(1 - (1 - 2\delta + \delta^2))^{\frac{d-1}{2}} \, d\delta \\
&= V_{d-1} \int_{0}^{\varepsilon^2} (\varepsilon^2 - \delta)^2(2\delta - \delta^2)^{\frac{d-1}{2}} \, d\delta
\end{aligned} \tag{109}$$

Since we assumed $\varepsilon^2 < \frac{1}{4}$, for the integration range $[0, \varepsilon^2]$, we may write $2\delta - \delta^2 = (2 - \delta)\delta \asymp_{7/4}^2 2\delta$.

$$\left(\frac{7}{4}\right)^{\frac{d-1}{2}} \delta^{\frac{d-1}{2}} \leq (2\delta - \delta^2)^{\frac{d-1}{2}} \leq 2^{\frac{d-1}{2}} \delta^{\frac{d-1}{2}}$$

The integral is approximated by:

$$V_{d-1}\left(\frac{7}{4}\right)^{\frac{d-1}{2}} \int_0^{\varepsilon^2} (\varepsilon^2 - \delta)^2 \delta^{\frac{d-1}{2}} \, \mathrm{d}\delta \leq I \leq V_{d-1} 2^{\frac{d-1}{2}} \int_0^{\varepsilon^2} (\varepsilon^2 - \delta)^2 \delta^{\frac{d-1}{2}} \, \mathrm{d}\delta \qquad (110)$$

Consider another change of variable: $\delta = \varepsilon^2 s$. Then $\mathrm{d}\delta = \varepsilon^2 \, \mathrm{d}s$. The limits change

$$\begin{aligned}
\int_0^{\varepsilon^2} (\varepsilon^2 - \delta)^2 \delta^{\frac{d-1}{2}} \, \mathrm{d}\delta &= \int_0^1 (\varepsilon^2 - \varepsilon^2 s)^2 (\varepsilon^2 s)^{\frac{d-1}{2}} (\varepsilon^2 \, \mathrm{d}s) \\
&= \int_0^1 (\varepsilon^2)^2 (1-s)^2 (\varepsilon^2)^{(d-1)/2} s^{\frac{d-1}{2}} \varepsilon^2 \, \mathrm{d}s \\
&= (\varepsilon^2)^{2 + \frac{d-1}{2} + 1} \int_0^1 (1-s)^2 s^{\frac{d-1}{2}} \, \mathrm{d}s \\
&= \varepsilon^{d+5} \int_0^1 (1-s)^2 s^{\frac{d-1}{2}} \, \mathrm{d}s \\
&= \underbrace{\left(\int_0^1 (1-s)^2 s^{\frac{d-1}{2}} \, \mathrm{d}s\right)}_{\text{constant}} \varepsilon^{d+5}
\end{aligned} \qquad (111)$$

The $L^2$ norm is the square root of $I$ is given by

$$c_7(d) \, \varepsilon^{\frac{d+5}{2}} \leq \left\|\varphi_{\boldsymbol{u}, \varepsilon^2}\right\|_{L^2(\mathbb{B}_1^d)} = \sqrt{I} \leq c_8(d) \, \varepsilon^{\frac{d+5}{2}} \qquad (112)$$

where $c_7(d)$ and $c_8(d)$ are constants defined by

$$\begin{aligned}
c_7(d) &= \sqrt{V_{d-1}\left(\frac{7}{4}\right)^{\frac{d-1}{2}} \left(\int_0^1 (1-s)^2 s^{\frac{d-1}{2}} \, \mathrm{d}s\right)} \\
c_8(d) &= \sqrt{V_{d-1} 2^{\frac{d-1}{2}} \left(\int_0^1 (1-s)^2 s^{\frac{d-1}{2}} \, \mathrm{d}s\right)}
\end{aligned} \qquad (113)$$

This completes the proof. $\qquad\square$

**Lemma H.2.** *Let $\varphi_{\boldsymbol{u}, \varepsilon^2}$ be a ReLU atom defined in* (105). *Then*

$$|\varphi_{\boldsymbol{u}, \varepsilon^2}|_{\mathrm{V}_g} = \varepsilon^{2d+4}. \qquad (114)$$

*Proof.* We decode the definiton (see Example 3.1) and compute directly the weighted function $g(\boldsymbol{u}, 1 - \varepsilon^2) = (\varepsilon^2)^{d+2} = \varepsilon^{2d+4}$. $\qquad\square$

Let $\mathbb{S}^{d-1}$ be the unit sphere in $\mathbb{R}^d$. For $0 < \varepsilon < 1$ and $\boldsymbol{w} \in \mathbb{S}^{d-1}$, define the spherical cap $C(\boldsymbol{u}, \varepsilon^2)$ as

$$C(\boldsymbol{u}, \varepsilon^2) = \{\boldsymbol{x} \in \mathbb{S}^{d-1} : \boldsymbol{u}^\mathsf{T} \boldsymbol{x} \geq 1 - \varepsilon^2\}. \qquad (115)$$

**Lemma H.3.** *Let $N_{max}(\varepsilon, d)$ denote the maximum number of points $\boldsymbol{u}_1, \ldots, \boldsymbol{u}_N \in \mathbb{S}^{d-1}$ such that the caps $C(\boldsymbol{u}_i, \varepsilon^2)$ are mutually disjoint. Then, as $\varepsilon \to 0$,*

$$N_{max}(\varepsilon, d) \asymp \varepsilon^{-(d-1)}$$

*where the implicit constants depend only on the dimension d.*

*Proof.* The spherical cap $C(\boldsymbol{u}, \varepsilon^2)$ has an angular radius $\vartheta = \arccos(1 - \varepsilon^2)$, satisfying $\vartheta = \Theta(\varepsilon)$ for small $\varepsilon$. The condition that caps $C(\boldsymbol{u}_i, \varepsilon^2)$ and $C(\boldsymbol{u}_j, \varepsilon^2)$ are disjoint requires the angular separation $\phi_{ij}$ between their centers $\boldsymbol{w}_i$ and $\boldsymbol{w}_j$ to be at least $2\vartheta$. Thus, $N_{max}(\varepsilon, d)$ is the maximum size $M(\mathbb{S}^{d-1}, 2\vartheta)$ of a $2\vartheta$-separated set (packing number) on $\mathbb{S}^{d-1}$.

The upper bound $N_{max}(\varepsilon, d) = O(\varepsilon^{-(d-1)})$ follows from a surface area argument: $N$ disjoint caps $C(\boldsymbol{u}_i, \varepsilon^2)$, each with surface area $\Theta(\vartheta^{d-1}) = \Theta(\varepsilon^{d-1})$, must fit within the total surface area of $\mathbb{S}^{d-1}$.

For the lower bound, we relate the packing number $M(\mathbb{S}^{d-1}, \alpha)$ to the covering number $N(\mathbb{S}^{d-1}, \alpha)$, the minimum number of caps of angular radius $\alpha$ needed to cover $\mathbb{S}^{d-1}$. It is a standard result that these quantities are closely related, for instance, $M(\mathbb{S}^{d-1}, \alpha) \geq N(\mathbb{S}^{d-1}, \alpha)$ can be shown via a greedy packing argument [Vershynin, 2018, see discussions in Chapter 4]. Furthermore, the asymptotic behavior of the covering number is known to be $N(\mathbb{S}^{d-1}, \alpha) \asymp \alpha^{-(d-1)}$ for small $\alpha$ [Vershynin, 2018, Corollary 4.2.14]. Setting the minimum separation $\alpha = 2\vartheta = \Theta(\varepsilon)$, we obtain the lower bound:

$$N_{max}(\varepsilon, d) = M(\mathbb{S}^{d-1}, 2\vartheta) \geq N(\mathbb{S}^{d-1}, 2\vartheta) \asymp (2\vartheta)^{-(d-1)} = \Omega(\varepsilon^{-(d-1)}),$$

where the implicit constants depend only on the dimension $d$. Combining the upper and lower bounds, we conclude that $N_{max}(\varepsilon, d) \asymp \varepsilon^{-(d-1)}$. $\qquad\square$

**Construction H.4.** We construct a suitable packing set in $\mathcal{F} = \{f \in V_g(\mathbb{B}_1^d) : \|f\|_{L^\infty} \leq 1, |f|_{V_g} \leq 1\}$ based on a weighted ReLU atoms. Let $\varphi_{\boldsymbol{u}, \varepsilon^2}$ be the ReLU atom defined in (105), and according to Lemma H.1 and Lemma H.2:

$$\Phi_{\boldsymbol{u}, \varepsilon^2} := \varepsilon^{-2} \varphi_{\boldsymbol{u}, \varepsilon^2} \implies \begin{cases} \|\Phi_{\boldsymbol{u}, \varepsilon^2}\|_{L^\infty(\mathbb{B}_1^d)} = 1, \\ \|\Phi_{\boldsymbol{u}, \varepsilon^2}\|_{L^2(\mathbb{B}_1^d)} \asymp \varepsilon^{\frac{d+1}{2}}, \\ |\Phi_{\boldsymbol{u}, \varepsilon^2}|_{V_g} = \varepsilon^{2d+2}. \end{cases} \tag{116}$$

According to Lemma H.1, there exists $N = c_N(d)\varepsilon^{-d+1}$ spherical caps $\boldsymbol{u_1}, \cdots, \boldsymbol{u_N}$ such that the caps $C(\boldsymbol{u}_i, \varepsilon^2)$ are mutually disjoint, for some constant $c_N(d) \leq 1$ that may depend on the dimension $d$. For convenience, we simply denote $\Phi_i = \Phi_{\boldsymbol{u}_i, \varepsilon^2}$. Therefore, we have $|N\Phi_i|_{V_g} = c_N(d)\varepsilon^{d+3} < 1$ referring to (116). For each $\xi = (\xi_1, \ldots, \xi_N) \in \{-1, 1\}^N$, define

$$f_\xi(\boldsymbol{x}) = \sum_{i=1}^N \xi_i \Phi_i(\boldsymbol{x}).$$

According to the conventions, each $f_\xi$ belongs to $\mathcal{F}$. Since the supports of the ridge functions are disjoint, for any $\xi, \xi' \in \{-1, 1\}^N$ we have

$$\|f_\xi - f_{\xi'}\|_{L^2(\mathbb{B}_1^d)}^2 = \sum_{i \in S} \|2\Phi_i\|_{L^2(\mathbb{B}_1^d)}^2 \asymp \varepsilon^{\frac{d+1}{2}} d_H(\xi, \xi'),$$

where $d_H(\xi, \xi')$ denotes the Hamming distance between $\xi$ and $\xi'$, and $S$ is the set of indices where $\xi_i \neq \xi'_i$. By the Varshamov–Gilbert lemma, there exists a subset $\Xi \subset \{-1, 1\}^N$ with

$$\log|\Xi| = K \asymp \varepsilon^{-d+1}.$$

for some constant, and such that for any distinct $\xi, \xi' \in \Xi$, the Hamming distance $d_H$

$$d_H(\xi, \xi') \gtrsim K.$$

Thus, for any distinct $\xi, \xi' \in \Xi$, we obtain

$$\|f_\xi - f_{\xi'}\|_{L^2(\mathbb{B}_1^d)} \gtrsim \varepsilon^{\frac{d+1}{2}} \sqrt{K} \asymp \varepsilon^{\frac{d+1}{2}} \varepsilon^{\frac{-d+1}{2}} = \varepsilon.$$

**Proposition H.5** (Minimax Lower Bound via Fano's Lemma). *Consider the problem of estimating a function $f \in \mathcal{F} = \{f \in V_g(\mathbb{B}_1^d) : \|f\|_{L^\infty} \leq 1, |f|_{V_g} \leq 1\}$ with*

$$y_i = f(\boldsymbol{x}_i) + \varepsilon_i, \quad i = 1, \ldots, n$$

*where $\{\varepsilon_i\}_{i=1}^n$ are i.i.d. $\mathcal{N}(0, \sigma^2)$ random variables and $\{\boldsymbol{x}_i\}_{i=1}^n \subset \mathbb{B}_1^d$ are i.i.d. uniform random variables on $\mathbb{B}_1^d$. The lower bound of the minimax non-parametric risk is given by*

$$\inf_{\hat{f}} \sup_{f \in \mathcal{F}} \mathbb{E}\|\hat{f} - f\|_{L^2(\mathbb{B}_1^d)}^2 \gtrsim \left(\frac{\sigma^2}{n}\right)^{\frac{2}{d+1}}.$$

*Proof.* We use the standard Fano's lemma argument. By our Construction (105), we have a packing set $\{f_\xi : \xi \in \Xi\}$ in $\mathcal{F}$ with the following properties:

1. The $L^2$-distance between any two distinct functions is at least $\delta$, where $\delta \asymp \varepsilon$.

2. The size of the packing set satisfies $\log |\Xi| \gtrsim K \asymp \varepsilon^{-(d-1)}$.

For Gaussian noise with variance $\sigma^2$, the Kullback–Leibler divergence between the distributions induced by two functions $f_\xi$ and $f_{\xi'}$ is

$$\mathrm{KL}(P_\xi \| P_{\xi'}) = \frac{n}{2\sigma^2} \|f_\xi - f_{\xi'}\|^2_{L^2(\mathbb{B}^d_1)} = \frac{n\,\delta^2}{2\sigma^2}.$$

In order to use Fano's lemma J.1 effectively, we need to satisfy the requirement (141), where in this context is

$$\frac{n\,\delta^2}{2\sigma^2} \lesssim \log |\Theta|$$

for some small constant $\alpha > 0$, then the minimax risk is bounded from below by a constant multiple of $\delta^2$.

Substituting $\delta \asymp \varepsilon$ and $\log |\Xi| \gtrsim \varepsilon^{-(d-1)}$, the condition becomes

$$\frac{n\,\varepsilon^2}{\sigma^2} \lesssim \varepsilon^{-(d-1)},$$

or equivalently,

$$n \lesssim \frac{\sigma^2}{\varepsilon^{d+1}}.$$

Solving for $\varepsilon$, we have

$$\varepsilon^{d+1} \asymp \frac{\sigma^2}{n} \quad \implies \quad \varepsilon \asymp \left(\frac{\sigma^2}{n}\right)^{\frac{1}{d+1}}. \tag{117}$$

Then, the separation becomes

$$\delta \asymp \varepsilon \asymp \left(\frac{\sigma^2}{n}\right)^{\frac{1}{d+1}}.$$

Therefore, Fano's lemma J.1 (particularly (140)) yields

$$\inf_{\hat{f}} \sup_{f \in \mathcal{F}} \mathbb{E} \|\hat{f} - f\|^2_{L^2(\mathbb{B}^d_1)} \gtrsim \delta^2 \asymp \left(\frac{\sigma^2}{n}\right)^{\frac{2}{d+1}},$$

which is the desired result. $\qquad\square$

**Corollary H.6.** *Let* $\{f \in \mathrm{V}_g(\mathbb{B}^d_1) : \|f\|_{L^\infty} \leq B, |f|_{\mathrm{V}_g} \leq C\}$. *Then*

$$\inf_{\hat{f}} \sup_{f \in \mathcal{F}} \mathbb{E} \|\hat{f} - f\|^2_{L^2} \gtrsim \min(B, C)^2 \left(\frac{\sigma^2}{n}\right)^{\frac{2}{d+1}}.$$

*Proof.* We just need to replace $f_\xi$ in Construction 105 by $\min(B, C) f_\xi$ and adapt it to the the proof of Proposition H.5. $\qquad\square$

## H.2 Why Classical Bump-Type Constructions Are Ineffective

The minimax lower bound construction in this paper crucially hinges on exploiting the properties of the data-dependent weighted variation norm, denoted as $|\cdot|_{\mathrm{V}_g}$, where the weight function is $g(\boldsymbol{u}, t)$. A key characteristic of $g(\boldsymbol{u}, t)$ (when data is, for instance, uniform on the unit ball $\mathbb{B}^d_1$) is that $g(\boldsymbol{u}, t) \asymp (1 - |t|)^{d+2}$. This implies that $g(\boldsymbol{u}, t)$ becomes very small as $|t| \to 1$, i.e., for activations near the boundary of the domain. This property allows for the construction of functions with significant magnitudes near the boundary using a relatively small variation norm. Therefore, any effective construction for the lower bound must create functions that are highly localized near this boundary.

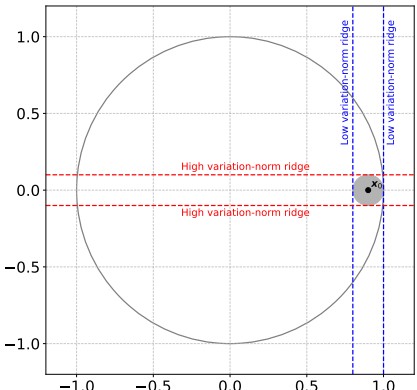

Figure 17: **Isotropic Locality is Costly:** An isotropic bump function, by definition, must be localized (decay rapidly) in all directions around its center. Suppose we place such a bump centered at a point $x_0$ near the boundary in direction $u_0$ (i.e., $u_0^\top x_0 \approx 1 - \varepsilon^2$, here $u_0 = (1,0)$). To achieve localization in directions *orthogonal* to $u_0$, one would need to combine ReLU activations whose ridges are oriented appropriately. More critically, to achieve localization in the direction *parallel* to $u_0$ (i.e., to ensure the bump decays as we move radially inward from $x_0$), we would need ReLU activations whose ridges $\{x : u_0^\top x = t\}$ have $t < 1 - \varepsilon^2$ and are potentially much closer to the origin (i.e., $t$ is significantly smaller than 1).

For these ReLU activations whose ridges are not very close to the boundary (i.e., $t$ is not close to 1), the weight function $g(u_0, t)$ will *not* be small. Consequently, constructing a sharply localized bump isotropically would require a substantial sum of weighted coefficients in the $V_g$ norm to cancel out the function in regions away from its intended support while maintaining a significant peak. This large variation norm would make such functions "too regular" or "too expensive" to serve as effective elements in a packing set for Fano's Lemma, especially when aiming to show a rate degradation due to dimensionality.

In essence, isotropic bump functions do not efficiently leverage the anisotropic nature of the ReLU activation and the specific properties of the $g(u, t)$ weighting. The construction used in this paper, which employs ReLU atoms active only on thin spherical caps near the boundary (an anisotropic construction), is far more effective. It allows for localization and significant function magnitude primarily by choosing the activation threshold $t$ to be very close to 1 (making $g(u, t)$ small), rather than by intricate cancellations of many neurons with large weighted coefficients. This is why such anisotropic, boundary-localized constructions are essential for revealing the curse of dimensionality in this setting.

### H.3   The Univariate Case

The minimax lower bound construction detailed above, which leverages a packing argument with boundary-localized ReLU neurons exploiting the multiplicity of available directions on $\mathbb{S}^{d-1}$, is particularly effective in establishing the curse of dimensionality for $d > 1$. However, the geometric foundation of this approach, specifically the ability to pack an exponential number of disjoint spherical caps, does not directly translate to the univariate case ($d = 1$) where the notion of distinct directional activation regions fundamentally changes. Consequently, the lower bound for $d = 1$ necessitates a separate construction or modification of the argument. Fortunately, in the one-dimensional setting, the distinction between isotropic and anisotropic function characteristics, which posed challenges for classical approaches in higher dimensions under the specific data-dependent weighted norm, becomes moot. This simplification allows us to directly employ classical bump function constructions, suitably adapted to the function class, to establish the minimax rates in one dimension.

According to Theorem 3.4, we have

$$|f|_{V_g} = \|g \cdot \mathscr{R}(-\Delta)^{\frac{d+1}{2}} f\|_{\mathcal{M}} \tag{118}$$

When $d = 1$ and $f$ is smooth, (118) is simplified to be

$$|f|_{V_g} = \|f'' \cdot g\|_{\mathcal{M}} = \int_{-1}^{1} |f''(x)|g(x)\,\mathrm{d}x = \int_{-1}^{1} |f''(x)|(1 - |x|)^3\,\mathrm{d}x \tag{119}$$

and so is the unweighted variation seminorm

$$|f|_V = \|f''\|_{\mathcal{M}} = \int_{-1}^{1} |f''(x)|\,\mathrm{d}x =: \mathrm{TV}^2(f) \tag{120}$$

which is also known as the second-order total variation seminorm. Therefore, the function class of stable minima in univariate case is characterized into

$$\mathcal{F}_{B,C} := \left\{ f : [-1,1] \to \mathbb{R} \,\big|\, \|f\|_{L^\infty} \le B,\ \|f'' \cdot (1 - |\cdot|)^3\|_{\mathcal{M}} \le C \right\}. \tag{121}$$

Using this characterization, it is more convenient to smooth bump functions to construct a minimax risk lower bound for stable minima class.

**Construction H.7.** Consider a smooth compact support function:

$$\Phi(x) = \begin{cases} c\exp\left(-\frac{1}{1-x^2}\right) & |x| < 1 \\ 0 & \text{otherwise} \end{cases}. \tag{122}$$

By adjusting the constant $c$, we may assume

$$\mathrm{TV}^2(\Phi) := \int_{-1}^{1} |\Phi''(t)|\,\mathrm{d}t = 1 \tag{123}$$

and let $D$ be a constant such that $\|\Phi(x)\|_{L^2} = \sqrt{2}D$. We can construct a translated and scaled version:

$$\Phi_{a,b}(x) = \Phi\left(\frac{2x - (a+b)}{b - a}\right) \quad \text{for } a < b. \tag{124}$$

and in particular, $\Phi_{a,b}$ has the following properties by directly computations:

$$\begin{cases} \mathrm{supp}(\Phi_{a,b}) &= (a,b), \\ \mathrm{TV}^2(\Phi_{a,b}) &= \frac{1}{b-a}, \\ \|\Phi_{a,b}\|_{L^2([-1,1])} &= \sqrt{b-a}\,D. \end{cases} \tag{125}$$

**Proposition H.8.** *Consider the problem of estimating a function $f \in \mathcal{F}_{1,1}$ with*

$$y_i = f(x_i) + \varepsilon_i,\ i = 1, \ldots, n$$

*where $\{\varepsilon_i\}_{i=1}^n$ are i.i.d. $\mathcal{N}(0, \sigma^2)$ random variables and $\{x_i\}_{i=1}^N \subset [-1,1]$ are i.i.d. uniform random variables on $[-1,1]$. The lower bound of the minimax non-parametric risk is given by*

$$\inf_{\hat{f}} \sup_{f \in \mathcal{F}_{1,1}} \mathbb{E}\|f - \hat{f}\|_{L^2([-1,1])}^2 \gtrsim \left(\frac{\sigma^2}{n}\right)^{\frac{1}{2}}$$

*Proof.* For any $\varepsilon > 0$, we may construct a family. Let $a_k = 1 - \varepsilon + k\varepsilon^2$, $k = 0, \ldots, \lfloor\frac{1}{\varepsilon}\rfloor$. We denote $K = \lfloor\frac{1}{\varepsilon}\rfloor$. For each $k = 1, \ldots, K$, we define $\Phi_k := \Phi_{a_{k-1}, a_k}$ Since $a_k - a_{k-1} = \varepsilon^2$, we have the following properties

- $\|\Phi_k\|_{L^2} = D \cdot \varepsilon$;

- $\mathrm{TV}^2(\Phi_k) \asymp \frac{1}{\varepsilon^2} \implies \int_{-1}^{1} |f''(t)|g(t)\,\mathrm{d}x \lesssim \varepsilon$ because $g(t) < \varepsilon^3$, $\quad \forall t \in [a_{k-1}, a_k]$.

Let $\{\Phi_1, \ldots, \Phi_K\}$, $K \asymp \lfloor\frac{1}{\varepsilon}\rfloor$ be such a family of function classes, and any $K$-terms combination $\{\Phi_1, \ldots, \Phi_K\}$ is in $\mathcal{F}_{1,1}$. Then we let

$$\phi : \{\pm 1\}^K \to \mathcal{F}_{1,1}, \quad \xi = (a_k)_{k=1}^K \mapsto \sum_{k=1}^{K} a_k \Phi_k =: f_\xi. \tag{126}$$

For any two indexes $\xi_1, \xi_2$ in $\{\pm 1\}^K$, we have that

$$\|f_{\xi_1} - f_{\xi_2}\|_{L^2} = \varepsilon\sqrt{d_H(\xi_1, \xi_2)}. \tag{127}$$

where $d_H$ is the Hamming distance. Then, using Varshamov-Gilbert's lemma (Lemma J.2), the pruned cube of $\{f_1, ..., f_M\}$ has a size $M \geq 2^{K/8}$, and each has the property that if $i \neq j$,

$$\|f_i - f_j\|_{L^2([-1,1])} \geq D \cdot \sqrt{\frac{K}{8}} \cdot \varepsilon \asymp \sqrt{\varepsilon},$$

and thus for any $i \neq j$

$$\mathrm{KL}(P_{f_i}\|P_{f_j}) = \frac{n\varepsilon}{2\sigma^2}$$

On the other hand, to satisfy the Fano inequality (141):

$$\frac{n\varepsilon}{2\sigma^2} = \mathrm{KL}(P_{f_i}\|P_{f_j}) \lesssim \log M \asymp \frac{1}{\varepsilon}$$

we let

$$\varepsilon \asymp \left(\frac{\sigma^2}{n}\right)^{\frac{1}{2}}.$$

and thus Fano's lemma (Lemma J.1, particularly (140)) implies that

$$\inf_{\hat{f}} \sup_{f \in \mathcal{F}_{1,1}} \mathbb{E}\|f - \hat{f}\|_{L^2([-1,1])}^2 \gtrsim \left(\frac{\sigma^2}{n}\right)^{\frac{1}{2}}.$$

$\square$

Note that by rescale the functions in the lower bound construction, we can deduce a more general result.

**Corollary H.9.** *For general case $\mathcal{F}_{B,C}$, we can scale the construction functions by $\min(B, C)$ to deduce the result:*

$$\inf_{\hat{f}} \sup_{f \in \mathcal{F}_{B,C}} \mathbb{E}\|f - \hat{f}\|_{L^2([-1,1])}^2 \gtrsim \min(B, C)^2 \left(\frac{\sigma^2}{n}\right)^{\frac{1}{2}} \tag{128}$$

# I  Lower Bound on Generalization Gap

In this section, we derive a lower bound for the generalization gap.

## I.1  The Lower Bound Construction Can be Realized by Stable Minima

Recall the notations in Construction H.4, for $\varepsilon \in (0, 1)$ and a unit vector $u \in \mathbb{S}^{d-1}$, define the (ball) cap

$$C(\boldsymbol{u}, \varepsilon) := \{x \in \mathbb{B}_1^d : \boldsymbol{u}^{\mathsf{T}}\boldsymbol{x} \geq 1 - \varepsilon\}.$$

Fix a dimension $d \geq 2$ and a fixed cap $C = C(\boldsymbol{u}, \varepsilon^2)$, the mass under $\mathrm{Uniform}(\mathbb{B}_1^d)$ satisfies the two–sided bound

$$\underbrace{\frac{2}{d+1}\frac{v_{d-1}}{v_d}}_{L_d^{cap}} \varepsilon^{d+1} \leq \mathbb{P}_{\boldsymbol{X} \sim \mathrm{Uniform}(\mathbb{B}_1^d)}(\boldsymbol{X} \in C) \leq \underbrace{\frac{2^{\frac{d+1}{2}}}{d+1}\frac{v_{d-1}}{v_d}}_{U_d^{cap}} \varepsilon^{d+1}. \tag{129}$$

where writing $v_k := \mathrm{Vol}(Bb_1^k)$.

Indeed, writing $h = \varepsilon^2$ and parameterizing $x = t\,u + \sqrt{1 - t^2}\,z$ with $t \in [1 - h, 1]$ and $z \in \mathbb{S}^{d-2}$, we have

$$\mathrm{Vol}(C) = v_{d-1} \int_{1-h}^1 (1 - t^2)^{\frac{d-1}{2}} \, dt = v_{d-1} \int_0^h \left(s(2 - s)\right)^{\frac{d-1}{2}} \, ds,$$

where $s = 1 - t$. Using $1 \leq 2 - s \leq 2$ on $[0, h]$ yields

$$v_{d-1} \int_0^h s^{\frac{d-1}{2}}\, ds \;\leq\; \mathrm{Vol}(C) \;\leq\; v_{d-1}\, 2^{\frac{d-1}{2}} \int_0^h s^{\frac{d-1}{2}}\, ds \;=\; \frac{2}{d+1} v_{d-1}\, h^{\frac{d+1}{2}} \;\leq\; \frac{2^{\frac{d+1}{2}}}{d+1} v_{d-1}\, h^{\frac{d+1}{2}}.$$

Dividing by $v_d = \mathrm{Vol}(\mathbb{B}_1^d)$ and recalling $h = \varepsilon^2$ gives the stated probability bounds.

**Proposition I.1** (Many caps does not have a sample point w.h.p. via Poissonization). *Fix $d \geq 2$ and $\varepsilon = \kappa\, n^{-1/(d+1)}$ with a constant $\kappa \in (0, 1]$. Let $\{C(\boldsymbol{u}_j, \varepsilon^2)\}_{j=1}^m$ be any family of pairwise-disjoint caps as in (129), and draw $\boldsymbol{X}_1, \ldots, \boldsymbol{X}_n \overset{\text{i.i.d.}}{\sim} \mathrm{Uniform}(\mathbb{B}_1^d)$. For each $j$, write $Z_j := \#\{i \leq n : \boldsymbol{X}_i \in C(\boldsymbol{u}_j, \varepsilon^2)\}$ and $p_j := \mathbb{P}(\boldsymbol{X} \in C(\boldsymbol{u}_j, \varepsilon^2))$. Then there exist absolute constants $c_*, C_* > 0$ (depending only on $d$ and $\kappa$) such that, for any $\delta \in (0, 1)$,*

$$\mathbb{P}\Big(\#\{1 \leq j \leq m : Z_j = 0\} \geq (q_* - \delta)\, m\Big) \;\geq\; 1 - \exp(-c_*\, \delta^2\, m) \;-\; C_*\, m\, \varepsilon^{2(d+1)},$$

*where $q_* = e^{-\lambda_+}$ and $\lambda_+ = U_d^{cap}\, \kappa^{d+1}$ is a constant independent of $n$. In particular, with probability at least $1 - \exp(-cm)$ (for some $c > 0$), there exists a subset $\Gamma \subset \{1, \ldots, m\}$ with $|\Gamma| \geq c_0\, m$ such that $Z_j \leq 1$ for every $j \in \Gamma$, where $c_0 \in (0, q_*)$ depends only on $d, \kappa$.*

*Proof.* Introduce a Poisson variable $N \sim \mathrm{Poi}(n)$ independent of the data. Conditionally on $N$, draw $\boldsymbol{X}_1, \ldots, \boldsymbol{X}_N \overset{\text{i.i.d.}}{\sim} \mathrm{Uniform}(\mathbb{B}_1^d)$. Let $\widetilde{Z}_j := \#\{i \leq N : \boldsymbol{X}_i \in C(\boldsymbol{u}_j, \varepsilon^2)\}$. By standard Poisson thinning, $\widetilde{Z}_1, \ldots, \widetilde{Z}_m$ are independent with $\widetilde{Z}_j \sim \mathrm{Poi}(\lambda_j)$ and

$$\lambda_j = \mathbb{E}[\widetilde{Z}_j] = n\, p_j, \qquad n\, L_d^{cap}\, \varepsilon^{d+1} \;\leq\; \lambda_j \;\leq\; n\, U_d^{cap}\, \varepsilon^{d+1}.$$

At the critical scaling $\varepsilon = \kappa\, n^{-1/(d+1)}$, we thus have constants

$$\underbrace{L_d^{cap}\, \kappa^{d+1}}_{\lambda_-} \;\leq\; \lambda_j \;\leq\; \underbrace{U_d^{cap}\, \kappa^{d+1}}_{\lambda_+}.$$

For each $j$, set $A_j := \mathbb{1}\{\widetilde{Z}_j = 0\}$. Then $A_1, \ldots, A_m$ are independent Bernoulli random variables with

$$\mathbb{E}[A_j] = \mathbb{P}(\mathrm{Poi}(\lambda_j) = 0) = e^{-\lambda_j} \geq \underbrace{e^{-\lambda_+}}_{q_*}.$$

Therefore, by Hoeffding's inequality for independent bounded variables,

$$\mathbb{P}\left(\frac{1}{m}\sum_{j=1}^m A_j \;<\; q_* - \delta\right) \;\leq\; \exp(-2\delta^2 m) \quad \text{for all } \delta \in (0, 1).$$

Equivalently,

$$\mathbb{P}\Big(\#\{j : \widetilde{Z}_j = 0\} \geq (q_* - \delta)\, m\Big) \;\geq\; 1 - \exp(-2\delta^2 m).$$

To proceed de-Poissonization, we compare $(Z_1, \ldots, Z_m)$ under the fixed-$n$ model (a multinomial random variable) to the corresponding joint Poisson variable $(\widetilde{Z}_1, \ldots, \widetilde{Z}_m)$. By Le Cam's inequality for Poisson approximation, the total variation distance between the joint law of the Bernoulli multi-variables $(Z_1, \ldots, Z_m)$ and that of independent $\mathrm{Poi}(\lambda_j)$ variables is bounded by

$$\sup_E \Big|\mathbb{P}_{\text{multi}}\Big((Z_1, \ldots, Z_m) \in E\Big) - \mathbb{P}_{\text{Poi}}\Big((\widetilde{Z}_1, \ldots, \widetilde{Z}_m) \in E\Big)\Big| \leq \sum_{j=1}^m p_j^2 \;\leq\; m\, (U_d^{cap}\, \varepsilon^{d+1})^2. \tag{130}$$

Applying this to the event $E = \{\#\{j : Z_j = 0\} \geq (q_* - \delta)m\}$ and combining with (130) yields

$$\mathbb{P}\Big(\#\{j : Z_j = 0\} \geq (q_* - \delta)\, m\Big) \;\geq\; 1 - \exp(-2\delta^2 m) \;-\; (U_d^{cap})^2\, m\, \varepsilon^{2(d+1)}.$$

Setting $c_* := 2$ and $C_* := (U_d^{cap})^2$ gives the stated bound. In particular, choosing any fixed $\delta \in (0, q_*)$ and defining $c_0 = q_* - \delta \in (0, q_*)$ proves that with probability at least $1 - \exp(-cm) - C_* m \varepsilon^{2(d+1)}$ there exists $\Gamma \subset \{1, \ldots, m\}$, $|\Gamma| \geq c_0 m$, such that $Z_j \leq 1$ for all $j \in \Gamma$. $\qquad\square$

Proposition I.1 ensures that, at the critical scaling $\varepsilon \asymp n^{-1/(d+1)}$, there exists (with overwhelmingly high probability) a *large* subcollection of caps, each containing no sample.

We now show that if a neural network does not have any activated datapoint, the operator norm of its Hessian is constantly 1.

**Proposition I.2.** *Let $f_{\boldsymbol{\theta}}(\boldsymbol{x}) = \sum_{k=1}^{K} v_k \phi(\boldsymbol{w}_k^{\mathsf{T}} \boldsymbol{x} - b_k) + \beta$ be network defined in* (1). *Let $\mathcal{D} = \{(\boldsymbol{x}_i, y_i)\}_{i=1}^{n}$ be a data set such that each neuron of $f_{\boldsymbol{\theta}}$ contains no activated datapoint, i.e for each $k$, $\sum_{i=1}^{n} \mathbb{1}\{\boldsymbol{w}_k^{\mathsf{T}} \boldsymbol{x}_i - b_k\} = 0$, and $f_{\boldsymbol{\theta}}$ interpolates $\mathcal{D}$ in the sense that $f_{\boldsymbol{\theta}}(\boldsymbol{x}_i) = y_i = 0$ for each $i$. Then $\lambda_{\max}\left(\nabla_{\boldsymbol{\theta}}^2 \mathcal{L}\right) = 1$.*

*Proof.* By direct computation, the Hessian $\nabla_{\boldsymbol{\theta}}^2 \mathcal{L}$ is given by

$$\nabla_{\boldsymbol{\theta}}^2 \mathcal{L} = \frac{1}{n} \sum_{i=1}^{n} \nabla_{\boldsymbol{\theta}} f_{\boldsymbol{\theta}}(\boldsymbol{x}_i) \nabla_{\boldsymbol{\theta}} f_{\boldsymbol{\theta}}(\boldsymbol{x}_i)^{\mathsf{T}} + \frac{1}{n} \sum_{i=1}^{n} (f_{\boldsymbol{\theta}}(\boldsymbol{x}_i) - y_i) \nabla_{\boldsymbol{\theta}}^2 f_{\boldsymbol{\theta}}(\boldsymbol{x}_i). \tag{131}$$

Since the model interpolates $f_{\boldsymbol{\theta}}(\boldsymbol{x}_i) = y_i$ for all $i$, we have

$$\nabla_{\boldsymbol{\theta}}^2 \mathcal{L} = \frac{1}{n} \sum_{i=1}^{n} \nabla_{\boldsymbol{\theta}} f_{\boldsymbol{\theta}}(\boldsymbol{x}_i) \nabla_{\boldsymbol{\theta}} f_{\boldsymbol{\theta}}(\boldsymbol{x}_i)^{\mathsf{T}}. \tag{132}$$

Consider the tangent features matrix that is defined by

$$\boldsymbol{\Phi} = \left[\nabla_{\boldsymbol{\theta}} f_{\boldsymbol{\theta}}(\boldsymbol{x}_1), \nabla_{\boldsymbol{\theta}} f_{\boldsymbol{\theta}}(\boldsymbol{x}_2), \cdots, \nabla_{\boldsymbol{\theta}} f_{\boldsymbol{\theta}}(\boldsymbol{x}_n)\right]. \tag{133}$$

Then we have $\nabla_{\boldsymbol{\theta}}^2 \mathcal{L} = \boldsymbol{\Phi}\boldsymbol{\Phi}^{\mathsf{T}}/n$, and the operator norm is computed by

$$\lambda_{\max}(\nabla_{\boldsymbol{\theta}}^2 \mathcal{L}) = \max_{\boldsymbol{\gamma} \in \mathbb{S}^{(d+2)K}} \frac{1}{n} \|\boldsymbol{\Phi}^{\mathsf{T}} \boldsymbol{\gamma}\|^2 = \max_{\boldsymbol{u} \in \mathbb{S}^{n-1}} \frac{1}{n} \|\boldsymbol{\Phi} \boldsymbol{u}\|^2 \tag{134}$$

Furthermore, we have

$$\nabla_{\boldsymbol{\theta}} f_{\boldsymbol{\theta}}(\boldsymbol{x}) = \begin{pmatrix} \nabla_{\boldsymbol{W}}(f_{\boldsymbol{\theta}}) \\ \nabla_{\boldsymbol{b}}(f_{\boldsymbol{\theta}}) \\ \nabla_{\boldsymbol{v}}(f_{\boldsymbol{\theta}}) \\ \nabla_{\beta}(f_{\boldsymbol{\theta}}) \end{pmatrix} \tag{135}$$

For the parameters $[\boldsymbol{w}_k, b_k, v_k]$ associated to the neuron of index $k$,

$$\frac{\partial f_{\boldsymbol{\theta}}(\boldsymbol{x})}{\partial v_k} = \mathbb{1}\{\boldsymbol{w}_k^{\mathsf{T}} \boldsymbol{x} > b_k\} \left(\boldsymbol{w}_k^{\mathsf{T}} \boldsymbol{x} - b_k\right), \qquad \frac{\partial f_{\boldsymbol{\theta}}(\boldsymbol{x})}{\partial \boldsymbol{w}_k} = \mathbb{1}\{\boldsymbol{w}_k^{\mathsf{T}} \boldsymbol{x} > b_k\} v_k \, \boldsymbol{x},$$

$$\frac{\partial f_{\boldsymbol{\theta}}(\boldsymbol{x})}{\partial b_k} = \mathbb{1}\{\boldsymbol{w}_k^{\mathsf{T}} \boldsymbol{x} > b_k\} v_k, \qquad \frac{\partial f_{\boldsymbol{\theta}}(\boldsymbol{x})}{\partial \beta} = 1.$$

Since there is no data point activating, we have that

$$\nabla_{(\boldsymbol{w}_k, b_k, v_k, \beta)} f_{\boldsymbol{\theta}}(\boldsymbol{x}_k) = \begin{pmatrix} \boldsymbol{0} \\ 1 \end{pmatrix}, \tag{136}$$

After subsistion by (136), (134) is of the form

$$\boldsymbol{\Phi} = \begin{pmatrix} \boldsymbol{0} & \boldsymbol{0} & \cdots & \boldsymbol{0} \\ \vdots & \vdots & \cdots & \vdots \\ \boldsymbol{0} & \boldsymbol{0} & \cdots & \boldsymbol{0} \\ 1 & 1 & \cdots & 1 \end{pmatrix}. \tag{137}$$

Let $\boldsymbol{u} = (u_1, \cdots, u_n) \in \mathbb{S}^{n-1}$ and plug (137) in (134) to have

$$\lambda_{\max}(\nabla_{\boldsymbol{\theta}}^2 \mathcal{L}) = \max_{\boldsymbol{u} \in \mathbb{S}^{n-1}} \frac{1}{n} \|\boldsymbol{\Phi} \boldsymbol{u}\|^2 = \max_{\boldsymbol{u} \in \mathbb{S}^{n-1}} \frac{\left(\sum_{i=1}^{n} u_i\right)^2}{n} = 1.$$

$\square$

We now establish that such a specially constructed interpolation solution is indeed stable. As shown in the proof, for an interpolation solution where none of the hidden neurons are active on the training data, the Hessian of the loss has an operator norm of exactly 1, i.e., $\lambda_{\max}(\nabla^2 \mathcal{L}(\boldsymbol{\theta})) = 1$. The primary contribution to this norm comes from the gradient of the output layer bias. According to the stability condition defined in Proposition 2.1 ($\lambda_{\max} \leq 2/\eta$), this solution is guaranteed to be in $\Theta_{\text{flat}}(\eta; \mathcal{D})$ so long as the step size satisfies $\eta \leq 2$. Since we assume that $\eta < 2$ in this paper (cf. Proposition 2.1 and the discussion below it), we have that this interpolating solution is indeed stable.

For brevity, we write $\mathcal{F}_{\text{flat}}(\eta; \mathcal{D}) := \{f_{\boldsymbol{\theta}} \mid \boldsymbol{\theta} \in \Theta_{\text{flat}}(\eta; \mathcal{D})\}$ in the sequel.

**Corollary I.3** (Stronger Version of Minimax Lower Bound). *Consider the problem of estimating a function $f \in \mathcal{F} = \{f \mid f \in \mathcal{F}_{\text{flat}}(\eta; \mathcal{D}), \|f\|_{L^\infty(\mathbb{B}_1^d)} \leq L\}$ with*

$$y_i = f(\boldsymbol{x}_i)$$

*where $\{\boldsymbol{x}_i\}_{i=1}^n \subset \mathbb{B}_1^d$ are i.i.d. uniform random variables on $\mathbb{B}_1^d$. The lower bound of the minimax nonparametric risk is given by*

$$\inf_{\hat{f}} \sup_{f \in \mathcal{F}} \mathop{\mathbb{E}}_{\mathcal{D} \sim \mathcal{P}^{\otimes n}} \|\hat{f}(\mathcal{D}) - f\|_{L^2(\mathbb{B}_1^d)}^2 \gtrsim_d L^2 \left(\frac{1}{n}\right)^{\frac{2}{d+1}}.$$

*where $\hat{f}$ refers to a estimator and $\hat{f}(\mathcal{D})$ means the estimation based on the data set $\mathcal{D}$.*

*Proof.* The core of the proof is to construct two functions, $f_1$ and $f_2$, which belong to the function class $\mathcal{F}$ but are far apart in $L^2$ norm. We will show that for a typical random data set $\{\boldsymbol{x}_i\}_{i=1}^n$, any estimator $\hat{f}(\mathcal{D})$ cannot distinguish between them, as they produce identical observations on $\mathcal{D}$. This implies a lower bound on the minimax risk.

We set the critical scaling for our construction to be:

$$\varepsilon = n^{-1/(d+1)}. \tag{138}$$

Following the geometric packing argument from Lemma H.3, we can find a set of $N \asymp \varepsilon^{-(d-1)}$ pairwise-disjoint spherical caps $\{C_j\}_{j=1}^N$, where each cap $C_j = C(\boldsymbol{u}_j, \varepsilon^2)$ is defined by a unique direction $\boldsymbol{u}_j \in \mathbb{S}^{d-1}$.

Let $\{\boldsymbol{x}_i\}_{i=1}^n$ be the randomly drawn data set's inputs. Let $\mathcal{E}$ be the event that there exists a subset of indices $\Gamma \subset \{1, \ldots, N\}$ such that:

(i) $|\Gamma| \geq c_0 N$ for some constant $c_0 > 0$.

(ii) For every $j \in \Gamma$, the cap $C_j$ is empty, i.e., $C_j \cap \{\boldsymbol{x}_i\}_{i=1}^n = \emptyset$.

According to Proposition I.1, this event $\mathcal{E}$ occurs with high probability, i.e., $\mathbb{P}(\mathcal{E}) \geq 1 - \exp(-c_1 N)$ for some constant $c_1 > 0$. From now on, we condition our entire analysis on this high-probability event $\mathcal{E}$ occurring.

Conditioned on the event $\mathcal{E}$, we now define two functions. Let $j \in \Gamma$ be an index corresponding to one of the empty caps, $C_j$.

1. Let the first function be the zero function:

$$f_1(\boldsymbol{x}) = 0.$$

   Clearly, $f_1 \in \mathcal{F}_{\text{flat}}(\eta; \mathcal{D})$ and $\|f_1\|_{L^\infty} = 0 \leq 1$.

2. Let the second function be a combination appropriately scaled ReLU atoms supported on the empty cap $C_j$:

$$f_2(\boldsymbol{x}) = L \sum_{j \in \Gamma} \varepsilon^{-2} \phi(\boldsymbol{u}_j^\top \boldsymbol{x} - (1 - \varepsilon^2)).$$

On the event $\mathcal{E}$, both functions produce the exact same observations. For any $\boldsymbol{x}_i$:

$$y_{i,1} = f_1(\boldsymbol{x}_i) = 0 \quad \text{and} \quad y_{i,2} = f_2(\boldsymbol{x}_i) = 0.$$

Therefore, the data set generated by both functions is identical: $\mathcal{D} = \{(\boldsymbol{x}_1, 0), \ldots, (\boldsymbol{x}_n, 0)\}$.

Since the data set $\mathcal{D}$ is fixed by the condition $\mathcal{E}$ and the cap $C_j$ is empty for $j \in \Gamma$, the neuron implementing $f_\Gamma$ is never active on any data point $\boldsymbol{x}_i \in \{\boldsymbol{x}_i\}_{i=1}^n$. This implies $f_2(\boldsymbol{x}_i) = 0$ for all $\boldsymbol{x}_i$. The corresponding labels are $y_i = 0$. Therefore, $f_2$ perfectly interpolates the data $(\boldsymbol{x}_i, 0)$. According to Proposition I.2, any such network that interpolates the data and has no active neurons on the data set has $\lambda_{\max}(\nabla^2 \mathcal{L}(\boldsymbol{\theta}_2)) = 1$, where $\boldsymbol{\theta}_2$ is the parameter vector that implements $f_2$. Since $\eta < 2$, this solution is stable. Thus, $f_2 \in \mathcal{F}_{\text{flat}}(\eta; \mathcal{D})$. Moreover, $\|f_2\|_{L^\infty} = L$, since the spherical caps $\{C_j\}$ are disjoint, at any point $\boldsymbol{x}$ at most one of the scaled ReLU atoms is non-zero. In summary, both $f_1$ and $f_2$ are valid functions in the class $\mathcal{F}$.

An estimator $\hat{f}$ takes as input the data set $\mathcal{D}$, which is identical for both potential ground-truth functions $f_1$ and $f_2$, and produces an estimate function, which we denote by $\hat{f}(\mathcal{D})$. The performance of this estimator is measured by its population risk. The estimator's objective is to minimize this risk under a worst-case choice of the ground truth from the set $\{f_1, f_2\}$.

For a given estimate $\hat{f}(\mathcal{D})$, the worst-case risk over this set is

$$\text{Risk}(\hat{f}(\mathcal{D})) = \max\left\{\left\|\hat{f}(\mathcal{D}) - f_1\right\|_{L^2(\mathbb{B}_1^d)}^2, \left\|\hat{f}(\mathcal{D}) - f_2\right\|_{L^2(\mathbb{B}_1^d)}^2\right\},$$

The minimax risk for this problem is the minimal possible worst-case risk achievable by any estimator. It is lower-bounded by considering the optimal decision rule conditioned on the event $\mathcal{E}$:

$$\inf_{\hat{f}} \sup_{f \in \{f_1, f_2\}} \mathbb{E}_{\mathcal{D}}\left[\left\|\hat{f}(\mathcal{D}) - f\right\|_{L^2(\mathbb{B}_1^d)}^2\right] \geq \inf_{\hat{f}} \text{Risk}(\hat{f}(\mathcal{D})) \cdot \mathbb{P}(\mathcal{E}).$$

The function $\hat{f}(\mathcal{D})^*$ that minimizes $\max\left\{\left\|\hat{f}(\mathcal{D}) - f_1\right\|_{L^2(\mathbb{B}_1^d)}^2, \left\|\hat{f}(\mathcal{D}) - f_2\right\|_{L^2(\mathbb{B}_1^d)}^2\right\}$ is the average of $f_1$ and $f_2$ in the Hilbert space $L^2(\mathbb{B}_1^d)$. This optimal estimate is $\hat{f}(\mathcal{D})^* = (f_1 + f_2)/2$. The minimal possible worst-case risk is thus achieved at this midpoint:

$$\inf_{\hat{f}(\mathcal{D}) \in \mathcal{F}} \text{Risk}(\hat{f}(\mathcal{D})) = \left\|\hat{f}(\mathcal{D})^* - f_1\right\|_{L^2(\mathbb{B}_1^d)}^2 = \left\|\frac{f_1 + f_2}{2} - f_1\right\|_{L^2(\mathbb{B}_1^d)}^2 = \left\|\frac{f_2 - f_1}{2}\right\|_{L^2(\mathbb{B}_1^d)}^2.$$

According to the computation in Construction H.4, we may conclude that

$$\|f_2 - f_1\|_{L^2(\mathbb{B}_1^d)}^2 = \left\|L \sum_{j \in \Gamma} \varepsilon^{-2} \phi(\boldsymbol{u}_j^\mathsf{T} \boldsymbol{x} - (1 - \varepsilon^2))\right\|_{L^2(\mathbb{B}_1^d)}^2 \asymp L^2 \varepsilon^2 \asymp L^2 \left(\frac{1}{n}\right)^{\frac{2}{d+1}}$$

This completes the proof. $\qquad\square$

**Theorem I.4.** *Let $\mathcal{P}$ denote any joint distribution of $(\boldsymbol{x}, y)$ where the marginal distribution of $\boldsymbol{x}$ is $\text{Uniform}(\mathbb{B}_1^d)$ and $y$ satisfies the $\mathbb{P}_\mathcal{P}[-D \leq y \leq D] = 1$.*

*Let $\mathcal{D} = \{(\boldsymbol{x}_j, y_j)\}_{j=1}^n$ be a data set of $n$ i.i.d. samples from $\mathcal{P}$, and that $\tilde{R}$ is any risk estimator that takes any $f$ and $\mathcal{D}$ as input, then outputs a scalar that aims at estimating the risk $R_\mathcal{P}(f) := \mathbb{E}_{(\boldsymbol{x},y) \sim \mathcal{P}}\left[(f(\boldsymbol{x}) - y)^2\right]$. Moreover, let $\mathcal{F}$ be the function class we defined in Corollary I.3.*

*Then*

$$\inf_{\tilde{R}} \sup_{\mathcal{P}} \mathbb{E}\left[\sup_{\substack{f \in \mathcal{F}_{\text{flat}}(\eta; \mathcal{D}) \\ \|f\|_{L^\infty(\mathbb{B}_1^d)} \leq L}} \left|R_\mathcal{P}(f) - \tilde{R}(f; \mathcal{D})\right|\right] \gtrsim_d L^2 n^{-\frac{2}{d+1}}. \tag{139}$$

*where we assume that $L \geq D$.*

*Proof.* Let the $\mathbb{E}[\cdot]$ be the short-hand for the expectation over the random training data set $\mathcal{D}$.

$$\inf_{\tilde{R}} \sup_{\mathcal{P}} \mathbb{E} \left[ \sup_{f \in \mathcal{F}_{\text{flat}}(\eta; \mathcal{D})} \left| R_{\mathcal{P}}(f) - \tilde{R}(f; \mathcal{D}) \right| \right]$$

$$\geq \inf_{\tilde{R}} \sup_{\substack{\mathcal{P} = \text{Unif}(\mathbb{B}_1^d) \times f_0 \\ f_0 \in \mathcal{F}_{\text{flat}}(\eta; \mathcal{D})}} \mathbb{E} \left[ \sup_{f \in \mathcal{F}_{\text{flat}}(\eta; \mathcal{D})} \left| R_{\mathcal{P}}(f) - \tilde{R}(f; \mathcal{D}) \right| \right]$$

$$\geq \inf_{\tilde{R}} \sup_{\substack{\mathcal{P} = \text{Unif}(\mathbb{B}_1^d) \times f_0 \\ f_0 \in \mathcal{F}_{\text{flat}}(\eta; \mathcal{D})}} \frac{1}{2} \mathbb{E} \left[ R_{\mathcal{P}}(\hat{f}_{\text{ERM}}(\tilde{R}(\cdot, \mathcal{D}))) - R_{\mathcal{P}}(f_0) \right]$$

$$\geq \inf_{\hat{f}} \sup_{\substack{\mathcal{P} = \text{Unif}(\mathbb{B}_1^d) \times f_0 \\ f_0 \in \mathcal{F}_{\text{flat}}(\eta; \mathcal{D})}} \frac{1}{2} \mathbb{E} \left[ R_{\mathcal{P}}(\hat{f}(\mathcal{D})) - R_{\mathcal{P}}(f_0) \right]$$

$$= \frac{1}{2} \inf_{\hat{f}} \sup_{\substack{\mathcal{P} = \text{Unif}(\mathbb{B}_1^d) \times f_0 \\ f_0 \in \mathcal{F}_{\text{flat}}(\eta; \mathcal{D})}} \mathbb{E} \left[ \| \hat{f}(\mathcal{D}) - f_0 \|_{L^2(\mathbb{B}_1^d)}^2 \right]$$

(Corollary I.3) $\longrightarrow \gtrsim_d L^2 n^{-\frac{2}{d+1}}$.

The first inequality restricts $\mathcal{P}$ further to deterministic labels with labeling functions in $\mathcal{F}$. Check that any function in $\mathcal{F}$ is bounded between $[-M, M]$. The second inequality uses the fact that $f_0 \in \mathcal{F}_{\text{flat}}(\eta; \mathcal{D})$, and the following decomposition

$$R_{\mathcal{P}}(\hat{f}_{\text{ERM}}(\tilde{R})) - R_{\mathcal{P}}(f_0)$$
$$= R_{\mathcal{P}}(\hat{f}_{\text{ERM}}(\tilde{R})) - \tilde{R}(\hat{f}_{\text{ERM}}; \mathcal{D}) + \tilde{R}(\hat{f}_{\text{ERM}}; \mathcal{D}) - \tilde{R}(f_0; \mathcal{D}) + \tilde{R}(f_0; \mathcal{D}) - R_{\mathcal{P}}(f_0)$$
$$\leq \left| R_{\mathcal{P}}(\hat{f}_{\text{ERM}}(\tilde{R})) - \tilde{R}(\hat{f}_{\text{ERM}}; \mathcal{D}) \right| + \left| \tilde{R}(f_0; \mathcal{D}) - R_{\mathcal{P}}(f_0) \right|$$
$$\leq 2 \sup_{f \in \mathcal{F}_{\text{flat}}(\eta; \mathcal{D})} \left| R_{\mathcal{P}}(f) - \tilde{R}(f; \mathcal{D}) \right|,$$

where we used $\tilde{R}(\hat{f}_{\text{ERM}}; \mathcal{D}) - \tilde{R}(f_0; \mathcal{D}) \leq 0$ from the definition of $\hat{f}_{\text{ERM}}$

$$\hat{f}_{\text{ERM}}(\tilde{R}(\cdot, \mathcal{D})) := \operatorname*{argmin}_{f \in \mathcal{F}_{\text{flat}}(\eta; \mathcal{D})} \tilde{R}(f; \mathcal{D}).$$

The third inequality enlarges the set of ERM estimators to any function of the data $\hat{f}$ that output. The subsequent identity uses the fact that $R_{\mathcal{P}}(f_0) = 0$. $\qquad \square$

This completes the proof for the lower bound on generalization gap stated in Theorem 3.5.

# J Technical Lemmas

## J.1 Information-Theoretic tools

Fano's Lemma provides a powerful method for establishing such minimax lower bounds by relating the estimation problem to a hypothesis testing problem. It leverages information-theoretic concepts, particularly the Kullback-Leibler (KL) divergence.

**Lemma J.1** (Fano's Lemma (Statistical Estimation Context)). *Consider a finite set of functions (or parameters) $\{f_1, f_2, \ldots, f_M\} \subset \mathcal{F}$, with $N \geq 2$. Let $P_{f_j}$ denote the probability distribution of the observed data $\mathcal{D}$ when the true underlying function is $f_j$. Suppose that for any estimator $\hat{f}$, the loss function $L(f_j, \hat{f})$ satisfies $L(f_j, \hat{f}) \geq s^2/2 > 0$ if $\hat{f}$ is not close to $f_j$ (e.g., if we make a wrong decision in a multi-hypothesis test where closeness is defined by a metric $d(f_j, f_k) \geq s$). More specifically, for function estimation with squared $L^2$-norm loss, if we have a packing set $\{f_1, \ldots, f_M\} \subset \mathcal{F}$ such that $\|f_j - f_k\|_{L^2}^2 \geq s^2$ for all $j \neq k$, then the minimax risk is bounded as:*

$$\inf_{\hat{f}} \sup_{f \in \mathcal{F}} \mathbb{E} \| \hat{f} - f \|_{L^2}^2 \geq \frac{s^2}{4} \left( 1 - \frac{\max_{j \neq k} \text{KL}(P_{f_j} \| P_{f_k}) + \log 2}{\log M} \right), \tag{140}$$

*provided the term in the parenthesis is positive.* $\mathrm{KL}(P_{f_j}\|P_{f_k})$ *denotes the Kullback-Leibler divergence between the distributions $P_{f_j}$ and $P_{f_k}$. For this bound to be non-trivial (e.g., $\gtrsim s^2$), we typically require that the number of well-separated functions $M$ is large enough such that*

$$\log \frac{M}{2} > \max_{j \neq k} \mathrm{KL}(P_{f_j}\|P_{f_k}). \tag{141}$$

One can refer to Wasserman [2020, Theorem 12, Corollary 13] or Tsybakov [2009, Chapter 2] for more details.

Our application of Fano's Lemma (for proving Proposition H.5) involves:

1. Constructing a suitable finite subset of functions $\{f_1, \ldots, f_M\}$ within the class $\mathcal{F}$ such that they are well-separated in the metric defined by the loss function (e.g., pairwise $L^2$-distance $s$). This is often achieved using techniques like the Varshamov-Gilbert lemma (Lemma J.2) for constructing packings.
2. Bounding the KL divergence (or another information measure like $\chi^2$-divergence) between the probability distributions generated by pairs of these functions. For $n$ i.i.d. observations with additive Gaussian noise $\mathcal{N}(0, \sigma^2)$, and if using the empirical $L^2$ norm $\|\cdot\|_{L^2(\mathbb{P}_n)}$ based on fixed data points $\boldsymbol{x}_i$, this divergence is often related to $\frac{1}{2\sigma^2}\sum_{i=1}^n (f_j(\boldsymbol{x}_i) - f_k(\boldsymbol{x}_i))^2$. More generally, for population norms, it's often $\frac{n\|f_j - f_k\|_{L^2}^2}{2\sigma^2}$.
3. Choosing $M$ and $s$ (or the parameters defining the packing) to maximize the lower bound, typically by ensuring that the KL divergence term does not dominate $\log M$.

**Lemma J.2** (Varshamov–Gilbert Lemma). *Let*

$$\Xi = \big\{\xi = (\xi_1, \ldots, \xi_N) \colon \xi_j \in \{0, 1\}\big\}.$$

*Suppose $N \geq 8$. Then there exist*

$$\xi^0, \xi^1, \ldots, \xi^M \in \Xi$$

*such that*

1. *$\xi_0 = (0, \ldots, 0)$,*

2. *$M \geq 2^{N/8}$,*

3. *for all $0 \leq j < k \leq M$, the Hamming distance satisfies*

$$d_H(\xi^j, \xi^k) \geq \frac{N}{8}.$$

*We call $\{\xi^0, \xi^1, \ldots, \xi^M\}$ a* pruned hypercube.

One can refer to Tsybakov [2009, Lemma 2.9] and Wasserman [2020, Lemma 15] for more details.

## J.2 Poissonization and Le Cam's Inequality

For a random variable $S$ on a probability space $(\Omega, \mathcal{F}, \mathbb{P})$ with values in a measurable space $(E, \mathcal{E})$, the law is $\mathcal{L}(S) := \mathbb{P} \circ S^{-1}$. For two laws $\mu, \nu$ on the same space, define the total variation distance

$$d_{\mathrm{TV}}(\mu, \nu) := \sup_{A \in \mathcal{E}} |\mu(A) - \nu(A)|.$$

When $E$ is countable, $d_{\mathrm{TV}}(\mu, \nu) = \frac{1}{2}\sum_{x \in E} |\mu(\{x\}) - \nu(\{x\})|$.

**Lemma J.3** (Poissonization [Barbour et al., 1992, Ch. 1]). *Let $N \sim \mathrm{Poi}(\lambda)$ and, conditional on $N$, let $W_1, \ldots, W_N$ be i.i.d. taking values in $\{0, 1, \ldots, m\}$ with $\mathbb{P}\{W = j\} = p_j$ for $j = 0, 1, \ldots, m$, where $p_0 := 1 - \sum_{j=1}^m p_j \geq 0$. Define $\widetilde{Z}_j := \#\{1 \leq i \leq N : W_i = j\}$ for $j = 1, \ldots, m$. Then $\widetilde{Z}_1, \ldots, \widetilde{Z}_m$ are independent and $\widetilde{Z}_j \sim \mathrm{Poi}(\lambda p_j)$.*

**Remark J.4.** This standard Poissonization trick replaces the fixed sample size $n$ by a Poisson random size $N \sim \mathrm{Poi}(n)$, making the cell counts independent. See Barbour, Holst and Janson [Barbour et al., 1992, Ch. 1] for a general treatment and applications in occupancy problems.

**Lemma J.5** (Le Cam's inequality for Poisson approximation [Cam, 1960, Arratia et al., 1989, Barbour et al., 1992]). *Let $(Z_1, \ldots, Z_m) \sim \mathrm{Mult}(n; p_1, \ldots, p_m, p_0)$ with $p_0 = 1 - \sum_{j=1}^m p_j$. Let $Y_1, \ldots, Y_m$ be independent with $Y_j \sim \mathrm{Poi}(np_j)$. Then there exists a universal constant $C > 0$ such that*

$$d_{\mathrm{TV}}\big(\mathcal{L}(Z_1, \ldots, Z_m), \ \mathcal{L}(Y_1) \otimes \cdots \otimes \mathcal{L}(Y_m)\big) \ \leq \ C \sum_{j=1}^m p_j^2.$$

**Remark J.6.** Lemma J.5 is a classical result of Le Cam [Cam, 1960], with modern proofs and refinements given by [Arratia et al., 1989] and by [Barbour et al., 1992, Sec. 1.3]. It provides a quantitative control of the total variation distance between the multinomial occupancy vector and the independent Poisson approximation. We use this bound to justify the de-Poissonization step in the proof of Proposition I.1.

