# OpenReview forum: "Stable Minima of ReLU Neural Networks Suffer from the Curse of Dimensionality: The Neural Shattering Phenomenon"
_NeurIPS.cc/2025/Conference — NeurIPS 2025 spotlight_

### Official Review · Reviewer_WMiE · 2025-06-16

**Clarity:** 2
**Significance:** 3
**Originality:** 3
**Rating:** 5
**Confidence:** 2

**Summary:**

The flatness of a parameter at a minima of the loss is typically correlated with the generalization, even though obstacles like reparametrization affect this claim.
This paper shows that dimensionality weakens the correlation by providing mostly theoretical results for two layers ReLU networks.

More precisely, after laying out a definition of flat or *stable minima* based on linear stability the paper first shows that flatness in parameter space translates to smoothness of the implemented function in the sense of a weighted variation norm.
This variation norm is defined for a function $f$ by finding a corresponding minimal representative $f_\nu$ of $f$ inside the class of infinite width shallow ReLU networks and computing the total variation of the measure $\nu$ defining $f_\nu$ multiplied by a weight function that depends on data.
Then, 4 bounds specific to stable minima are given, first on the generalization then on robustness to noise.
Upper bounds show that stable minima can generalize and converge towards the noiseless function (i.e. infinite data leads to perfect results).
Lower bounds reveal an exponential dependency on the dimension.

**Questions:**

1. **Intuition on the weight function** I interpret the weight function as a multiplication between (a) the expected activations of a neuron and (b) the probability of a neuron being activated times a data norm in the case the neuron is activated. In particular, what is the role of the (b) term ? And why can't the weight function $g(u,t)$ be simply $\tilde g(u,t)$ ?
1. **Sharpness-Aware Minimization** (SAM) (e.g. "Sharpness-Aware Minimization for Efficiently Improving Generalization" by Foret et al, "ASAM: Adaptive Sharpness-Aware Minimization for Scale-Invariant Learning of Deep Neural Networks" by Kwon et al) seem to work on vision datasets like CIFAR, which are very high dimensional. They use other models but do you have comments on the relation with your work and in particular on the fact that dimensionality didn't seem to be blocking in SAM ?
1. **Pseudo sparsity** I am a bit sceptical on the pseudo-sparsity phenomenon being specific to flatness. When trying to fit perfectly a function without regularization, the parameter might diverge to endlessly try to minimize the loss. This has been theoretically studied in "Best k-Layer Neural Network  Approximations" by Lim et al. If you think the pseudo-sparsity phenomenon is fundamentally different and specific to flat minima, could you contrast the 2 phenomena ?

**Ethical Concerns:**

["NO or VERY MINOR ethics concerns only"]

**Final Justification:**

In their rebuttal the authors clarified a misconception I had which was that the lower bound in theorem 3.7 could be achieved only by very specific functions and was therefore not very informative for typical parameters obtained via gradient descent.
Moreover they also offered helpful intuitive explanations that allowed me to better grasp part of their work, in particular regarding how pseudo-sparsity enables even stable minima to overfit.
Additionally, the authors clarified the scope of their contributions with respect to SAM and addressed my concerns about Corollary 3.3.
Overall, I find the paper presents an interesting and motivated negative result: that flatness alone is insufficient to bypass the curse of dimensionality.
The theoretical analysis is well detailed, and some empirical validation is provided.
Therefore, I see no reason to recommend against acceptance.
Notably, the authors also convincingly addressed other reviewers’ comments on the role of network width.
I still chose a low confidence score because I am not an expert on the mathematical tools used in the paper and found parts of the paper challenging to follow.

**Limitations:**

yes

**Paper Formatting Concerns:**

I didn't notice issues

**Quality:**

3

**Strengths And Weaknesses:**

## Quality
The paper seems technically sound. The theoretical claims are supported with a detailed technical appendix.
Experimental results are complementary to the theoretical claims, but are more of a sanity check than an experimental validation.

However I have some doubts on the upper bound of Corollary 3.3:
- with a low loss and a high learning rate it seems the norm could become negative.
- in the paragraph below Corollary 3.3: in case of high loss, scaling $\eta \to \infty$ would not be enough to get an affine function which has 0 seminorm. The upperbound does not necessarily converge to 0 if the loss is high so I don't understand this claim.

## Clarity

The exposition is good overall except for the beginning of section 3.
The different steps leading to the comprehensive view of the weighted variation seminorm are difficult to follow.
I think adding motivation or intuition along the way (on the choice of the weight function and its role w.r.t. to the measure, $\infty$ width NN, Radon measure, total variation of a measure) could help the reader, like it is done in the previous section.\
Less critically in theorem 3.7 it is not clear exactly over which set $\hat{f}$ can vary.


**Minors**
- line 187: in the definition of $\mathcal F_{\text{stable}}(\eta)$, there is no condition on the $\theta$ being a minimum but the text after refers to stable minima.
- line 148: "sharp minima can also be generalized" -> "can generalize".
- line 218: "any ReLUs ... <del> is </del> can be".
- Figure 1: missing x/y axis labels.
- Figure 2: in the caption, orange instead of red.
- Notation in theorem F.8 in appendix: $\hat{R}$ vs $\hat{R_n}$, $f_\theta$ vs $f$.
- $\phi$ is never defined as ReLU


## Significance

This paper reveals that dimensionality is another important component to take into account when studying the relation between flatness and generalization.
A theoretical understanding of this phenomenon could be of great use to the community as it might help distinguishing when flatness is relevant and when it is not (however, see my question on SAM below).
Moreover the paper shows a methodology on how to link the parameter and function spaces, which is an important step when thinking about identifiability for example.

One weak point is that the lower bounds, which are the most significant results since the upper bounds are basically feasibility statement, are formulated for parameters that are not the outcome of a training process.
For example, this means that the lower bound of theorem 3.5 could be achieved for exotic configurations inside $\mathcal F_{\text{stable}}(\eta)$ that are not the solutions found by a typical optimization algorithm (as for theorem 3.7 see "Clarity" above).
This fact limits the insights provided by the paper.

The experiments reported in Figure 1, though limited in scope, address the gap empirically: it is reported that solutions found with high learning rate (stable minima) suffer when increasing the dimensionality.

## Originality

There were already results on the smoothness of function implemented by stable minima, corollary 3.3 is a similar result for another definition of function smoothness.

While smoothness results have also been obtained for the multivariate case, the part of this paper focusing explicitly on the quantifying the role of dimension on generalization and overfitting noise is to the best of my knowledge new (and critically, the bounds).

---

> ### Author Rebuttal · Authors · 2025-07-30
>
> Thanks for your support and detailed feedback! We will check and revise the minors. Responses to your questions below:
>
> >   For example, this means that the lower bound of theorem 3.5 could be achieved for exotic configurations inside $\mathcal{F}_{\text{stable}}(\eta) $ that are not the solutions found by a typical optimization algorithm (as for theorem 3.7 see "Clarity" above). This fact limits the insights provided by the paper.
>
> Thank you for mentioning this critical point. We argue that the configurations used in our lower bound are not exotic because they are precisely the structures that emerge from a typical training process, which our experiments empirically confirm.
>
> Our work uses the set of stable minima as a prism to understand the dynamics of GD's representation leanring. Within this paradigm, our lower bound construction (see Append B.4 Figure 9 for the visualization of our lower bound construction) provides a intuition: it proves the existence of a "hard-to-learn" solution built from pseudo-sparse neurons, hinting at a dynamic where neurons pathologically shatter the dataset and get 'stuck' fitting local points. Moreover, the higher the dimension is, the easier for these neuron to shatter the dataset.
>
> Furthermore, our experiments then provide direct empirical evidence for this theoretical construction. As shown in (Figure 1 right panel, Appendix A.1, A.2), vanilla Gradient Descent converges to a solution containing many neurons that exhibit these exact characteristics: a very low activation rate but a high weight norm. This demonstrates that the fundamental components of our lower bound represents a tangible failure mode produced by standard optimization algorithms in high dimensions.
>
>
> >Intuition on the weight function I interpret the weight function as a multiplication between (a) the expected activations of a neuron and (b) the probability of a neuron being activated times a data norm in the case the neuron is activated. In particular, what is the role of the (b) term ?
>
> This is a great question! As derived in [Nacson et al. 2023, Appendix F], the form of $\tilde{g}(  {u},t)$ is derived by brute-force computation of the loss function's Hessian matrix for a two-layer ReLU network:$$
> \nabla_{\theta}^2\mathcal{L}=\frac{1}{n}\sum_{i=1}^n\nabla_{  {\theta}}f_{\theta}( x_i)\nabla_{\theta}f_{\theta}(x_i)^{\top} + \sum_{i=1}^n(f_{\theta}(x_i)-y_i)\nabla^2_{\theta}f_{  \theta}(x_i)
> $$
>
> Intuitively, a neuron defined by the hyperplane $  {w}^{\top}  {x} - b = 0$ contributes to the Hessian based on how many data points are in its activation region. Hyperplanes that are near the boundary of the data support (i.e., activate on very few data points) contribute less to the Hessian. Consequently, the weight $g(  {u},t)$ is small in these regions. This means that the stability condition imposes a weaker regularization on neurons that are rarely active.
>
>
> Ref: [Nacson et al. 2023]: Nacson et al. The Implicit Bias of Minima Stability in Multivariate Shallow ReLU Networks, ICLR 2023.
>
>
>
> >  why can't the weight function $g(  {u},t)$ be simply $\tilde{g}(  {u},t)$?
>
> Note that any function $f$ implemented by a ReLU neural network, can be displayed by different sets of parameters. Our goal is to find a lower bound on the top eigen vector of the Hessian for any implementation of $f$. However, choosing between $\tilde{g}(  {u},t)$ and $\tilde{g}(-  {u},-t)$ directly according to the sign of the hidden weight vector $  {w}$ is implementation dependent. For example, a linear function can be implemented by $  {w}^{\top}x=\phi(  {w}^{\top}x-t)-\phi(-  {w}^{\top}x+t)$ and by shifting $t$, we can get infinitely many implementations. Therefore, the choice of $\min(g(  {u},t),g(-  {u},-t))$ guarantees that the bound applies to all possible implementations of the function $f$.
>
>
> >  However I have some doubts on the upper bound of Corollary 3.3...
>
> Corollary 3.3 is deduced from Theorem 3.2 in our paper. The right handside of the inequality in Theorem is deduced from
>
>
> $$1 + 2\sum_{k=1}^Ka_k \tilde{g}(u_k,t_k)\leq \lambda_{max}(\frac{1}{n}\sum_{i=1}^n \nabla_{\theta} f(x_i) \nabla_{\theta} f(x_i)^{T})$$
>
>
> where $a_k$ is the path norm of the $k$-th neuron, see [Nacson et al. 2023, Appendix F. eq 47, 48, 50, 52-54]. Therefore, the term $\lambda_{max}(\frac{1}{n}\sum_{i=1}^n\nabla_{\theta}f(x_i) {\nabla}_{\theta} f(x_i)^{\top}) $ is at least 1. In other words, if $\eta >2$, then the set of stable solutions is an empty set. In practice, when $\eta\geq 1$, gradient descent cannot converge even for a linear ground true functions with mild noise. Therefore, it is almost impossible to use a very large learning rate to achieve a low empirical risk.
>
> >Sharpness-Aware Minimization (SAM) (e.g. "Sharpness-Aware Minimization for Efficiently Improving Generalization" by Foret et al, "ASAM: Adaptive Sharpness-Aware Minimization for Scale-Invariant Learning of Deep Neural Networks" by Kwon et al) seem to work on vision datasets like CIFAR, which are very high dimensional. They use other models but do you have comments on the relation with your work and in particular on the fact that dimensionality didn't seem to be blocking in SAM?
>
> SAM modifies the loss function, and therefore has a different loss landscape that comes with no sharp minima.  We do not modify the loss function and focus on what gradient descent can stably find. Our results thus do not apply to the solution of SAM. For CIFAR-type datasets, we believe there are other effects that might prevent the curse-of-dimensionality, such as the intrinsic low-dimensional data manifold that the neural networks might be able to adapt to. The takeaway from the investigation of four paper is that stability / flatness is not the reasons that neural networks avoid the curse of dimensionality.
>
>
>
>
>
> > Pseudo sparsity I am a bit sceptical on the pseudo-sparsity phenomenon being specific to flatness. When trying to fit perfectly a function without regularization, the parameter might diverge to endlessly try to minimize the loss. This has been theoretically studied in "Best k-Layer Neural Network Approximations" by Lim et al. If you think the pseudo-sparsity phenomenon is fundamentally different and specific to flat minima, could you contrast the 2 phenomena ?
>
> It is not about pseudo-sparsity being specific to flatness, but rather such pseudo-sparse (and overfit) solutions can be found *even if* we impose the flatness constraint.
>
> Compared to the work of [Lim et al.], which studies a large-weight solution that interpolates (and overfits) the noisy labels, our study aims at understanding whether flatness is sufficient in ruling out such bad solutions. Our paper shows that while the flatness constraint is strong enough to rule out the large weights on neurons that activate frequently, the neurons that activate rarely can still be permitted to have large weights. This was a very surprising observation to us. Our result also highlights that we should not solely depend on the implicit bias of GD to prevent overfitting.
>
> Therefore, we believe the two phenomena are fundamentally different, and our work is novel.

---

> > ### Comment · Reviewer_WMiE · 2025-08-03
> >
> > Thank you for responding to my questions and clarifying the points I raised.
> > Besides, the intuition you provided in your answers could help convey the core message of your work more clearly, as noted in my earlier comment on clarity.
> > I have no further questions, as all the issues I brought up have been addressed convincingly and I will improve my score.

---

### Official Review · Reviewer_UZNs · 2025-06-29

**Clarity:** 3
**Significance:** 3
**Originality:** 3
**Rating:** 5
**Confidence:** 3

**Summary:**

This work considers studies generalization bounds for a subset of functions which can be implemented by a two layer ReLU network. More specifically, the authors consider functions corresponding to local minima of a sample least squares loss which are $\epsilon$-stable,  such functions are shown to be regular with respect to a certain variation norm. In addition, uniform lower and upper generalization bounds are provided which decay to zero zero as the number of samples grows, the input dimension $d$ is assumed constant. For these bounds to be useful the sample complexity needs to grow exponentially in the data dimension.

**Questions:**

- Proposition 2.2: I don't see how this can be true for any $\epsilon$, if this where true then wouldn't this mean that you get convergence to the minimizer from any initial condition? Perhaps another way of putting it is that the linearized dynamics can't be a good approximation for any $\theta$ right unless unless the loss is a quadratic form in the parameters?

- A small comment, you also say on line 29 that ReLU networks are thought not to benignly overfit, but there are quite a number of papers now showing this exact phenomenon for $d$ growing like $n^2$ (or even $n$ for leaky ReLU networks) instead of being fixed. In addition, the specialization of neurons (i.e., sparse activations) is often how one shows benign overfitting.

- Can you provide some intuition as to the origin of the weighting function given in (4), how should one think about this?

- Does your class of $\epsilon$-stable functions only include those minimizers for which the loss is twice differentiable? What happens to minimizers which are not differentiable / are on the boundary between cells?

- What is special about ReLU here? Would similar results also hold for Leaky-ReLU, what is stopping you extend your techniques to other activation functions? Also what is the main technical barrier moving to 3 layers as opposed to 2?

- To confirm / clarify the main takeaway, is it the implicit bias of GD (i.e., minima that are stable for gradient descent) do not in general guarantee good generalization, unless you have an exponentially large amount of data in the input dimension $d$ (which then perhaps is not too surprising for data arising on a compact domain), and this is in contrast to weight decay?

**Ethical Concerns:**

["NO or VERY MINOR ethics concerns only"]

**Final Justification:**

This paper has limitations in terms of its practical relevance with regard to setup and assumptions. However, I think the theoretical analysis and observations are interesting: in particular, this work shows that flatness alone as measured by the largest eigenvalue of the Hessian does not guarantee good generalization / an escape from the curse of dimensionality and empirically illustrates a setting in which this pseudo sparsity phenomenon occurs.

**Limitations:**

Not applicable.

**Paper Formatting Concerns:**

None observed.

**Quality:**

3

**Strengths And Weaknesses:**

Understanding the role of flat / stable minima is, at least in my opinion, a topic of significant interest, the paper is well written and clear and the technical contributions seem good. Overall I think this paper should be accepted.

---

> ### Author Rebuttal · Authors · 2025-07-30
>
> Thank you for your support and insightful comments! Responses to your questions below:
>
> >  Proposition 2.2: I don't see how this can be true for any $\epsilon$, if this where true then wouldn't this mean that you get convergence to the minimizer from any initial condition? Perhaps another way of putting it is that the linearized dynamics can't be a good approximation for any $\theta$ right unless unless the loss is a quadratic form in the parameters?
>
> Our proposition is not a claim about global convergence from an arbitrary starting point. Instead, it is about a local-stability condition. The purpose is to use this local analysis to derive a static condition that characterizes the stable minima. The "initial condition" in our stability definition refers to the moment an iterate enters the local quadratic region of a local minimum, $\theta^\star$. The proposition then shows that the curvature condition, $\lambda_{max}(\nabla^2\mathcal{L}(\theta^\star))\leq 2/\eta$, is necessary and sufficient for the linearized dynamics to remain trapped near $\theta^\star$. A rigorous proof can be found in [Qiao et al 24, Appendix C].
>
> It is crucial to note that while this mathematical proposition about the *linearized system* holds for any $\epsilon$, we agree with the reviewer that this linearization is only a faithful approximation of the true GD dynamics for small $\epsilon$. This local regime is precisely where the concept of stability is meaningful, and it allows us to define the entire set of stable minima, which is the true object of our paper's generalization analysis. The key idea is that we are not analyzing the convergence from an arbitrary initialization, but rather characterizing the intrinsic properties of the entire set of stable minima that GD could possibly find.
>
> Ref: [Qiao et al 24] Qiao et al Stable Minima Cannot Overfit in Univariate ReLU Networks: Generalization by Large Step Sizes, Neurips 2024
>
> >  A small comment, you also say on line 29 that ReLU networks are thought not to benignly overfit, but there are quite a number of papers now showing this exact phenomenon for $d$ growing like $n^2$ (or even $n$ for leaky ReLU networks) instead of being fixed. In addition, the specialization of neurons (i.e., sparse activations) is often how one shows benign overfitting.
>
> Our work is motivated by the observation that standard ReLU networks typically do not benignly overfit in fixed-dimensional, noisy regression. The recent work by [Haas et al. 2023] supports this, showing that benign overfitting in this setting requires modifying the ReLU activation with special high-frequency functions. In their paper, Theorem 4 explicitly confirms that standard ReLU NTKs are inconsistent when overfitting and Figure 1c empirically shows that the standard ReLU network exhibits harmful overfitting. This suggests the phenomenon is not intrinsic to standard ReLU. We would be grateful if the reviewer could share specific citations that show otherwise.
>
> Ref: [Haas et al. 2023]: Haas et al, Mind the spikes: benign overfitting of kernels and dneural networks in fixed dimension. Neurips 2023
>
> >  Can you provide some intuition as to the origin of the weighting function given in (4)..." & "What is special about ReLU here? Would similar results also hold for Leaky-ReLU, what is stopping you extend your techniques to other activation functions?
>
> These are great questions! We would answer these two questions by explaining their common mechanism organically.
>
> The weight function $g$ arises from two complementary perspectives: a static analysis of the loss landscape's curvature and a dynamic view of representation learning.
>
> From a static viewpoint, as derived in [Nacson et al. 2023, Appendix F], the form of $g(  {u},t)$ emerges directly from a careful analysis of the loss function's Hessian matrix for a two-layer ReLU network. Intuitively, a neuron defined by the hyperplane $  {w}^{\top}  {x} - b = 0$ contributes to the Hessian based on how many data points it activates. Hyperplanes that are "on the boundary" of the data support (i.e., activate very few data points) contribute less to the Hessian. Consequently, the weight $g(  {u},t)$ is small in these regions, meaning the stability condition imposes a weaker regularization on neurons that are rarely active.  ReLU is analytically special because its hard-sparsity property leads to a sparse loss Hessian, which allows for a clean derivation of the weighting function $g(  {u},t)$ that is central to our analysis.
>
>
> From a dynamic learning perspective, one can form an intuitive picture of how these pseudo-sparse neurons might emerge. High-dimensional geometry may make it easier for neurons' activation boundary to drift to shatter the dataset and isolate a very small subset of data points for each of them. Such a neuron only receives gradients from a few nearby points and if those points are already well-fit, the local gradient on this neuron's parameters can vanish, causing it to get "stuck" nearby the boundary. The small value of $g(  {u},t)$ near the data boundary can be seen as the mathematical manifestation of this hypothesized dynamic: it creates "space" within the class of stable functions for these trapped, high-magnitude neurons to exist, a possibility our lower bound construction then formalizes.
>
>
> Informally speaking, **our work uses the set of stable minima as a prism to understand the dynamics of GD's representation leanring**. The insight gained from ReLU is a general one that should be reasonably extrapolated to other activations like Leaky-ReLU, where weak negative gradients also fail to solve the underlying problem. Our intuition is that a small leaky coefficient would yield similar results (some light experiments confirmed that), as the weak gradient in the negative domain is likely insufficient to overcome this scattered signal problem. A large coefficient (e.g., an absolute value activation) might behave differently, but a rigorous analysis is challenging. We will conduct more experiments to investigate this behavior.
>
> Ref: [Nacson et al. 2023]: Nacson et al. The Implicit Bias of Minima Stability in Multivariate Shallow ReLU Networks, ICLR 2023.
>
> >  Does your class of $\epsilon$-stable functions only include those minimizers for which the loss is twice differentiable...
>
> We thank the reviewer for this very insightful question that touches upon a key technical aspect of our work. We'd like to highlight three points. First, even though we are motivated by stable minima, the set of interest (below Prop 2.2) covers all NNs with Hessian's largest-eigenvalue being small. This set includes not just twice-differentiable stable local / global minima, but also many other low-curvature regions that are not local / global minima. Second, non-differentiable points in two-layer ReLU networks correspond to points where the hyperplane of a ReLU unit coincides with a data point. This is a measure 0 set. If we initialize the network randomly from a density and optimize using gradient descent (with constant stepsize), with probability 1, the whole trajectory will not run into such points. (see Footnote 2 in the paper).
> Third, analyzing twice-differentiable solutions is, in some sense, without loss of generality. This is because we can perturb the a non-differentiable solution infinitesimally to get a differentiable solution.  Thus, by the Lipschitzness of the loss, it suffices that there is a differentiable solution with low-curvature near the non-differentiable solution of interest.
>
>
>
>
> >  Also what is the main technical barrier moving to 3 layers as opposed to 2?
>
> Moving from a two-layer to a three-layer network introduces two primary technical barriers that would require highly non-trivial extensions of our current analytical framework.
>
> First, our analysis of the generalization gap and estimation error relies on calculating the metric entropy of some function class. For a two-layer ReLU network, the function space can be related to dictionary learning and ridge spline approximations, allowing us to leverage existing tools to bound its complexity [Parhi & Nowak 2023]. For a three-layer network, is is not even known what is the ``correct'' function class to consider. Even if the class were known, estimating the complexity of the corresponding function class would be significantly more involved and would require substantial new theoretical work.
>
> Second, the derivation of the data-dependent weighting function is tied to a precise analysis of the loss function's Hessian matrix. For a three-layer network, the Hessian becomes much more complex.
>
> Ref: [Parhi & Nowak 2023]: Rahul Parhi & Robert Nowak Near-minimax optimal estimation with shallow ReLU neural networks. IEEE Transcations on Information Theory, 2023
>
> >   To confirm / clarify the main takeaway...
>
> Yes, the reviewer's summary is an accurate takeaway within the setting of our paper. The implicit bias of gradient descent towards stable minima does not, on its own, prevent the curse of dimensionality, leading to a sample complexity that grows exponentially with the input dimension $d$. This is in sharp contrast to explicit regularization like weight decay.
>
> However, we wish to add a crucial clarification: the driving force behind this phenomenon is the data distribution's interaction with the high-dimensional geometry, not just the ambient dimension in isolation. Our work highlights this "data-dependent regularity" and particularly examines the case of the uniform distribution on a ball. As mentioned in the discussion part of the paper, the induced weighted function $g$ inherits the full geometry of the data and becomes harder to describe and interpret for arbitary distributions, so we leave such an investivgation as future work.

---

> > ### Comment · Reviewer_UZNs · 2025-08-04
> > **Thanks for answering my questions**
> >
> > I appreciate your detailed responses and will keep my score.

---

### Official Review · Reviewer_f8m7 · 2025-06-30

**Clarity:** 3
**Significance:** 3
**Originality:** 4
**Rating:** 5
**Confidence:** 4

**Summary:**

The paper investigates stable minima of two-layer ReLU networks using the MSE loss. It shows that although flatness, measured as the largest EV of the loss Hessian, implies generalization, this guarantee degrades in high-dimensional settings. Specifically, the dependence on dataset size deteriorates exponentially with the input dimension. Notably, these results hold in the non-interpolating regime, where networks do not fit the training data exactly but still seek stable minima.

**Questions:**

Questions:
- How do the theoretical results relate to deep learning practice? There, input dimensions have grown enormously (LLMS and LVLMs), but while datasests have grown enormously as well, it seems their growth was not exponential in the input dimension.
- The theoretical insight is interesting when compared to the results from relative flatness [1]. There, the bound also deteriorates exponentially, but in the dimension of the feature extractor, i.e., with roughly $n^\frac{-2}{4+m}$, where $m$ is the dimension of the feature space. In this paper, $m$ would be the number of hidden neurons. Now if $m$ was kept fixed when increasing $d$ the results from relative flatness would imply that generalization does not deteriorate, which would contradict this paper. I suspect, however, that this is ruled out in the construction of the proof: since the pseudo-sparse functions need to have a number of localized "bumps" that grows with $d$, $m$ would grow with $d$. If that is the case, it seems that indeed it is not the input dimension that ultimately governs the generalization ability, but the dimension of the feature extractor. If my argument is correct, this should be discussed openly.
- The discussion on low-norm solutions vs flat solutions is very interesting. I wonder whether this is particular to the MSE, since for the cross-entropy loss larger weights (at least in the penultimate layer) imply higher confidence, which in turn implies a flatter minimum [2]. So one could use this result to support flatness measures that take weight norms into account [1,3].

[1] Petzka, Henning, et al. "Relative flatness and generalization." Advances in neural information processing systems 34 (2021): 18420-18432.

[2] Walter, Nils Philipp, et al. "The uncanny valley: Exploring adversarial robustness from a flatness perspective." arXiv preprint arXiv:2405.16918 (2024).

[3] Liang, Tengyuan, et al. "Fisher-rao metric, geometry, and complexity of neural networks." The 22nd international conference on artificial intelligence and statistics. PMLR, 2019.

**Ethical Concerns:**

["NO or VERY MINOR ethics concerns only"]

**Final Justification:**

I remain convinced that this is a very solid paper and therefore maintain my score.

**Limitations:**

Limitations are adequately addressed and assumptions are explicitly mentioned.

**Paper Formatting Concerns:**

There are no major formatting issues.

**Quality:**

4

**Strengths And Weaknesses:**

**Strengths:**
- Interesting minimax lower bound using pseudo-sparse functions, where neurons have large weights but rarely activate.
- The analysis of flatness and generalization remains an important subject. The investigation using hard-to-learn functions is conceptually interesting.
- Although experiments are synthetic, they are well-designed and support the theory.

**Weaknesses:**
- It would have been great to test some assumptions in more realistic scenarios.

---

> ### Author Rebuttal · Authors · 2025-07-30
>
> Thank you for your support and insightful comments! Responses to your questions below:
>
> > The theoretical insight is interesting when compared to the results from relative flatness [1].
>
> Thanks for bringing up the Relative Flatness paper [1]. We will cite and appropriately discuss that work. However, while both [1] and our paper study how flatness is connected to generalization, the settings of the two papers seem very different, and consequently, the nature of the results is different.
>
> First and foremost, the definition of flatness in [1] seems to *condition on* the feature map $\phi$, i.e., the loss function is not a function of the parameter of $\phi$ and the first layer weights and biases are not learned. On the otherhand, in our paper, the loss takes *both* the first and second layer weights and biases as input. Our motivation is to study feature learning. Thus, in our setting, we do not consider a fixed feature map $\phi$.
>
> Second, we focus on the overparameterized ($m>n$) regime with the hope of understanding whether "flatness/EoS/MinimaStability" can be used to explain the generalization of neural networks in this regime. In the generalization bound (Theorem 6) in [1], it seems that $m$ (this is $K$ in our paper) cannot be too large before the $n^{-2/(m+4)}$ bound becomes vacuous. Our upper bound is **independent** of $m$. Thus, when $m$ is sufficiently large, our bound is tighter than that of Theorem 6 of [1]. Furthermore, there was not a lower bound established in [1] that requires an exponential dependence on $m$.
>
> > Now if $m$ was kept fixed when $d$ increasing, the results from relative flatness would imply that generalization does not deteriorate, which would contradict this paper.
>
> The reviewer is absolutely right here! Our lower bound does have an implicit assumption that $m$ is sufficiently large.  If $m$ is fixed and $n\rightarrow \infty$, then we are in the classical regime and one can obtain an $O(\sqrt{dm/n})$-type uniform generalization gap bound (for all solutions and thus applicable to flat solutions as well). This bound does not have $\exp(d)$ dependence. In this under-parameterized regime, our upper bound (that depends exponentially on $d$) is still valid, but vacuous. Our exponential lower bound requires, naturally, that $m$ be sufficiently large to avoid the contradiction. A sufficient condition for our bound to kick in is that $m = \Omega(n)$.
>
> In particular, we highlight that our Theorem 3.7 does not have this condition on $m$. This is because it deals with the weighted variation class directly, which can be viewed as a space of neural networks with arbitrary (even infinite) width and so the neural networks parameterization is abstracted away.
>
> We will add the above discussion to the paper to increase readability.
>
> >  How do the theoretical results relate to deep learning practice? There, input dimensions have grown enormously (LLMS and LVLMs), but while datasets have grown enormously as well, it seems their growth was not exponential in the input dimension.
>
> This is a good point. We thought about this too. If the input data are truly somewhat uniformly distributed on a ball in $\mathbb{R}^d$ then our lower bound would apply directly there. However, images and texts are very structured, and we believe they are embedded in a much lower-dimensional manifold.  The key to bridge the theory-practice gap could be to study how neural networks adapt to such low-dimensional structures.
>
>
> > Discussion on low-norm vs flat solutions is interesting. Does it only apply to MSE?
>
> The function-space characterization of low-norm (i.e., weight decay) solutions is independent of the loss function of interest, thus the resulting generalization bounds apply to other losses. On the other hand, the function-space characterization of "flatness" in terms of the largest-eigenvalue of Hessian is sensitive to the choice of loss functions.
>
> More specifically, we believe that the key features of low-norm vs flat solutions that we discovered in this paper do apply beyond MSE although there might be other factors to consider for other loss functions.
>
> In particular, we studied MSE as a first step. Some parts of the techniques work beyond MSE but this would require a thorough study, e.g., if the Hessian decomposition of other twice-differentiable loss functions has a similar structure to that of the square loss. The papers that you brought up are interesting (thanks!), and we will look into them more closely to understand the difficulty of going beyond MSE as well as the merits of other weight-norm sensitive flatness definitions for future work.

---

> > ### Comment · Reviewer_f8m7 · 2025-08-01
> > **Reply**
> >
> > Dear authors,
> >
> > Thank you for the great reply. Just a quick clarification question: Am I right in my understanding that $m$ (or your $K$) depends on the number of localized bumps of the pseudo-sparse functions?

---

> ### Author Response · Authors · 2025-08-01
>
> Thank you for your feedback and the following question! The short answer is yes! More specifically, one localized bump functions is implemented by one ReLU atom, so the required number of bumps to satisfy Fano's lemma for the lower-bound risk is the required width of the NN, which is $K\asymp n$  (see Appendix B.4 lines 483-492 for more detail, see also Figure 9 for the visualization of the localized bump functions).

---

> > ### Comment · Reviewer_f8m7 · 2025-08-03
> > **Reply**
> >
> > But then indeed both results are compatible. Thank you. I of course will maintain my positive score.

---

### Official Review · Reviewer_fmSs · 2025-07-02

**Clarity:** 3
**Significance:** 2
**Originality:** 3
**Rating:** 5
**Confidence:** 3

**Summary:**

The paper conducts a theoretical study of two-layer ReLU networks, focusing on the interplay between generalization ability of a minimum, and the flatness of the objective landscape around it, as measured by the maximum eigenvalue of the Hessian; with flatness also being equivalent to stability of gradient descent in the neighborhood of the minimum. The main theoretical result shows that while the network trained to reach an empirical minimum has vanishing generalization gap, the required number of training samples $n$ required for the gap to be small grows exponentially with the input dimension $d$. Authors provide both an upper and a lower bound on the generalization gap for stable/flat minima, and show a construction of a stable minimum that generalizes poorly, and consists of pseudo-sparse activations, with few ReLUs firing for any specific input. The theoretical results are validated experimentally.

**Questions:**

- Does network width affect the theoretical findings?

- How much are the empirical results influenced by width, e.g. what would happen for K=d, or K=4d as in Transformer MLP? At the extreme, a single ReLU hidden layer (K=1) should have no problem learning a noisy d-dimensional linear function, by shifting the ReLU bias to be outside of the data domain and effectively becoming a two-layer linear network in the data domain. More generally, lower width should, hypothetically, make pseudo-sparse solution less attractive.

**Ethical Concerns:**

["NO or VERY MINOR ethics concerns only"]

**Final Justification:**

The additional experiments provided in the authors' rebuttal have resolved my concerns about the impact of width.

**Limitations:**

yes

**Quality:**

3

**Strengths And Weaknesses:**

The paper extends our knowledge about the connections between flatness of the objective function around a minimum and the generalization capability of the minimum. Prior work shows varying levels of relationship, depending on the specific scenarios. Most closely, prior work with univariate ReLU network showed that stable minima, those with flat objective around a minimum, generalize well. Authors show that this is not the case in the multivariate case. Previous studies of 1-hidden-layer ReLU showed that it can be trained (with large step size) to arrive at stable/flat minima, resulting in smooth predictor function. The current paper shows that this does not translate to efficient (low sample complexity as a function of input dimensionality) generalization capabilities. On the other hand, they also show that networks trained with weight decay do generalize in the same scenarios.

The experimental part provides support for the theoretical findings, yet the spectrum of scenarios / synthetic datasets is very limited. The dimensionality of the datasets is very small (d<=5), while the networks are relatively very wide (e.g. 2048 ReLU neurons in Fig. 2). While this setup serves the purpose of demonstrating in practice the existence of pseudo-sparse solutions, it does so at the expense of demonstrating the applicability of the bounds across a broad rande of widths.  Also, only linear ground truth is considered (it would be interesting to see at least one simple non-linear function, e.g. absolute value).

Overall, while the above advances are novel, there are somewhat narrow in scope and in its relationship to practice. The connection between flatness and generalization has been questioned before, with recent work showing that training towards a flat minimum is beneficial in some tasks and/or architectures, but not in others (e.g. [1]). One wonders to what extent is the result influenced by not limiting the width of the network (if width plays some role in theorems and proofs, authors should make it more explicit) while at the same time not limiting weight magnitudes via e.g. weight decay. The study of very wide ReLU networks with no weight decay has limited connection to practice, where weight decay is typically used, width is limited, and, in recent years, GELU is increasingly used instead of ReLU.

[1] Kaddour et al., When do flat minima optimizers work? NeurIPS’22

---

> ### Author Rebuttal · Authors · 2025-07-30
>
> We thank the reviewer for their time and for providing the opportunity to clarify our contributions.
>
> **Clarification of the set-up of the experiment**
>
> The setup of our experiments was designed to test the precise claims of our generalization bounds. Our theory investigates the inductive bias of vanilla GD in the overparameterized regime, where generalization occurs even as network width scales with the sample size (e.g the scaling law). This is the reason why we use wide neural networks.
>
> Furthermore, the decision to limit our experiments to $d\leq5$ is a direct and practical consequence of our theory. Our bounds predict that the required sample complexity for effective generalization scales exponentially with dimension $d$. The primary goal of our log-log plots is therefore not to achieve a low error in high dimensions, but to empirically demonstrate the trend that the generalization slope flattens as d increases. Our results show this unequivocally: the slope for $d=5$ is already exceptionally flat for a simple linear function, indicating severe sample inefficiency. Extending the experiments to higher dimensions would require a computationally infeasible number of samples merely to confirm an already established trend, providing no additional scientific insight for the immense cost involved.
>
> >Also, only linear ground truth is considered (it would be interesting to see at least one simple non-linear function, e.g. absolute value).
>
> We have conducted experiments on non-linear functions (e.g., quadratic) and confirmed the same generalization trends, and we will add these experiments to the appendix in the final version.
>
>
>
> > Does network width affect the theoretical findings?
>
> We focus on the overparameterized regime because the underparameterized regime does not require implicit bias for generalization. Our upper bound does apply to small $K$, but the bound could be bigger than the standard parametric rate $\sqrt{Kd/n}$. Our lower bound requires $K$ to be large to be relevant for neural networks. We will make the assumption that $K>n$ explicit in the final version.
>
>
> >a single ReLU hidden layer ($K=1$) should have no problem learning a noisy d-dimensional linear function, by shifting the ReLU bias to be outside of the data domain and effectively becoming a two-layer linear network in the data domain.
>
> We agree that a 1-unit ReLU network is capable of learning a linear function. We ran new experiments and found that GD with sufficiently small learning rate does indeed finds good solutions.
>
> Our experiments focused on the overparameterized setting (when $K$ is large) for the reason we described earlier (generalization bound via standard uniform convergence is readily available even without flatness or EoS).  Our aim is not to find the best model / algorithm for fitting a linear function, but to illustrate how the modern regime of overparameterize + GD can be fragile even in such a simple setting.
>
> Interestingly, we find that when $K = 16, d = 64$ and $n=1024$, the "pseudo-sparsity" phenomenon is still very prominent. We suspect that even a mild overparameterization (over the number of parameters to describe the target function) is rather harmful, if we solely rely on "flatness / implicit bias of GD stability".
>
>
> This finding allows us to reiterate our core research paradigm. Our work uses the set of stable minima as a prism to understand the dynamics of GD's representation leanring, since directly analyzing large-step-size gradient dynamics is notoriously difficult. Within this paradigm, our lower bound construction (see Append B.4 Figure 9 for the visualization of our lower bound construction) provides a intuition: it proves the existence of a "hard-to-learn" solution built from pseudo-sparse neurons, hinting at a dynamic where neurons pathologically shatter the dataset and get 'stuck' fitting local points. For example, neurons whose activation boundary is near the edge of the data support only receives gradients from a few nearby points and if those points are already well-fit, the local gradient on this neuron's parameters can vanish, causing it to get "stuck" nearby the boundary. The higher the dimension is, the easier for these neuron to shatter the dataset.
>
> We agree that the width of an NN plays a role on the probability of the occurance of this dynamic. As the reviewer astutely points out, such a "stuck" state is unlikely with just a single neuron. However, once representational redundancy is introduced (e.g K=16, d=64), the learning task may become distributed on various neurons and thus create the possibility for a pseudo-sparse neuron to become stable, as other neurons can compensate for the global loss, effectively shielding the "stuck" neurons from further updates. Moreover, our empirical results support this perspective (even with a narrow NN). We thank the reviewer for their insight and will clarify this perspective in the final version.
>
>
> >in recent years, GELU is increasingly used instead of ReLU.
>
>
> We acknowledge the reviewer's point on GELU. Following the gradient analysis perspective, we believe this problem is not unique to ReLU. While GELU provides a non-zero gradient for negative inputs, this signal is weak and decays quickly. It is therefore plausible that this weak gradient is insufficient to pull a "stuck" neuron back from the data boundary. To test this, we conducted new experiments, and our empirical results show that the pseudo-sparsity and poor generalization phenomena indeed persist for GELU networks. We will add these supporting experiments to the appendix in the final version.

---

> > ### Comment · Reviewer_fmSs · 2025-08-05
> >
> > Thank you for the responses, especially for the additional experiments exploring the small-width scenario. This addresses my main concern, and I will raise the score.

---

### Comment · Area_Chair_JFmg · 2025-08-04

Dear reviewers,

Thank you for your valuable time and your expertise in reviewing. Engaging with authors is really important, and allows both authors and reviewers to gain deeper understanding of cutting edge topics. This is a unique opportunity of interaction in our community.

The author rebuttal phase is about to close, and we kindly request your prompt attention to ensure a thorough discussion.

The discussion period **ends in less than 3 days** (on Aug. 6, 11:59pm AOE ). To maintain the review timeline, we ask that you:

- Review the rebuttals,

- Engage in any ongoing discussion with fellow reviewers/authors (if applicable),

- Finalize your assessment.

If you have already completed this step, we sincerely appreciate your efforts.

Thank you for your collaboration!

Best regards,

AC

---

### Decision · Program_Chairs · 2025-09-17

**Decision:**

Accept (spotlight)

**Comment:**

This paper presents an in-depth study of the generalization performance and landscape properties of ReLU networks beyond the interpolation regime. The style is formal, pleasant, precise; the results are interesting, well supported numerically, and of great interest in the community of theoretical deep learning.

All reviewers agree the paper deserved publication, and I 100% agree with this. I congratulate the authors for focusing on such fundamental properties of simple models, and I find reading this extremely refreshing.